# FANTASTIC GENERALIZATION MEASURES ARE NOWHERE TO BE FOUND

**Michael Gastpar**
EPFL
michael.gastpar@epfl.ch

**Ido Nachum**
University of Haifa
inachum@univ.haifa.ac.il

**Jonathan Shafer**
MIT
shaferjo@mit.edu

**Thomas Weinberger**
EPFL
thomas.weinberger@epfl.ch

## ABSTRACT

We study the notion of a generalization bound being *uniformly tight*, meaning that the difference between the bound and the population loss is small for all learning algorithms and all population distributions. Numerous generalization bounds have been proposed in the literature as potential explanations for the ability of neural networks to generalize in the overparameterized setting. However, in their paper "Fantastic Generalization Measures and Where to Find Them," Jiang et al. (2020) examine more than a dozen generalization bounds, and show empirically that none of them are uniformly tight. This raises the question of whether uniformly-tight generalization bounds are at all possible in the overparameterized setting. We consider two types of generalization bounds: (1) bounds that may depend on the training set and the learned hypothesis (e.g., margin bounds). We prove mathematically that no such bound can be uniformly tight in the overparameterized setting; (2) bounds that may in addition also depend on the learning algorithm (e.g., stability bounds). For these bounds, we show a trade-off between the algorithm's performance and the bound's tightness. Namely, if the algorithm achieves good accuracy on certain distributions, then no generalization bound can be uniformly tight for it in the overparameterized setting. We explain how these formal results can, in our view, inform research on generalization bounds for neural networks, while stressing that other interpretations of these results are also possible.

## 1 INTRODUCTION

There has been extensive research in recent years aiming to understand generalization in neural networks. Principled mathematical approaches often focus on proving *generalization bounds*, which bound the population risk from above by quantities depending on the training set and the trained model. Unfortunately, many known bounds of this type are often very weak, or even vacuous[1], and they do not imply performance guarantees that could explain the strong real-world generalization of neural networks. Incidentally, there might be a good reason for this: in this paper we show that it is mathematically impossible for certain types of generalization bounds to be tight in a specific sense.

Generalization bounds in the literature often take the following form:

$$L_{\mathcal{D}}(\mathcal{A}(S)) < L_S(\mathcal{A}(S)) + C(\mathcal{A}(S), S), \tag{1}$$

where $S$ is the training set, $\mathcal{A}(S)$ is the hypothesis selected by the learning algorithm $\mathcal{A}$, $L_{\mathcal{D}}$ and $L_S$ denote the population and empirical risk respectively, $C$ is some measure of complexity, and the inequality holds with high probability over the choice of $S$. For example, in their paper "Fantastic Generalization Measures and Where to Find Them," Jiang et al. (2020) examine more than a dozen generalization bounds of this form that have been suggested in the literature.

---

[1]A generalization bound is *vacuous* if it implies a population loss no better than guessing random labels.

For a generalization bound to be useful, it should ideally be *tight*, meaning that the difference between the two sides of Eq. (1) is small with high probability. Moreover, we shall call a bound *uniformly tight* (Definition 7) if it is tight for every (distribution, algorithm)-pair.

In order to explain generalization in deep neural networks, it is necessary that the bound be tight in the *overparameterized* setting, which roughly means that the number of parameters in the networks is much larger than the number of examples in the training set (Definition 2; further definitions appear in Sections 4 to 6). Given that essentially all known generalization bounds do not satisfy these two criteria, it is natural to ask:

> **Question 1.** *Does there exist a generalization bound of the form of Eq. (1) that is uniformly tight in the overparameterized setting?*

Obviously, one can always bound the population loss using a *validation set*. However, the upper bound in Eq. (1) depends only on the hypothesis $\mathcal{A}(S)$ and the training set $S$, so using an additional validation set does not technically satisfy the requirement of Question 1. Beyond technicalities, using a validation set is conceptually very different from a generalization bound. Using a validation set is a post hoc measurement that provides little insight as to why a certain algorithm does or does not generalize. In contrast, a meaningful generalization bound (like the VC bound[2] for example) provides a scientific theory that predicts the behavior of learning algorithms in a wide range of conditions, and can inform the design of novel learning systems.

One might imagine that the generalization bounds for neural networks surveyed by Jiang et al. (2020) are not uniformly tight simply because the analyses in the proofs of these bounds are not optimal, and that a more careful proof might establish a tighter bound with better constants. Or perhaps, none of these measures yield uniformly-tight bounds for large neural networks, but in the future researchers might devise better complexity measures of the form of Eq. (1) that do. We show that obtaining a bound of the form of Eq. (1) that is uniformly tight requires more assumptions than are typically found in the current literature.

## 2 OUR CONTRIBUTIONS

Following is an overview of our contributions, which are also summarized in Table 1. Our results are stated using a notion of *estimability*, which is presented informally in Eq. (2) below (for formal definitions, see Definitions 3 and 4). **All proofs appear in the appendices.**

### 2.1 DISTRIBUTION- AND ALGORITHM-INDEPENDENT GENERALIZATION BOUNDS

One central message of this paper is that the answer to Question 1 is negative. The conclusion we draw from this and further analysis is that generalization bounds can be uniformly tight in the overparameterized setting, but only under suitable assumptions on the population distribution or the learning algorithm. Arguably, many bounds in the literature are presented without assumptions of the type we show are necessary for uniform tightness — so their tightness for any specific use case is not guaranteed.

To reason about Question 1, we introduce the notion of estimability. Informally, a hypothesis class $\mathcal{H}$ is *estimable* with accuracy $\varepsilon$ if there exists an *estimator* $\mathcal{E}$ such that for every algorithm $\mathcal{A}$ and every $\mathcal{H}$-realizable distribution $\mathcal{D}$, the inequality

$$\left| L_{\mathcal{D}}\big(\mathcal{A}(S)\big) - \mathcal{E}\big(\mathcal{A}(S), S\big) \right| < \varepsilon \tag{2}$$

holds with high probability over the choice of $S$ (see Definition 3). An immediate (but conceptually important) observation is that, in the realizable case, a uniformly tight generalization bound like Eq. (1) exists if and only if $\mathcal{H}$ is estimable (Claim 1). Furthermore, if $\mathcal{H}$ is not estimable then there exists no uniformly tight bound as in Eq. (1) also for learning in the agnostic (non-realizable) setting.

---

[2]The VC bound does not satisfy Question 1 because it is vacuous in the overparameterized setting.

Our negative results for the realizable setting are stronger than (i.e., they imply) negative results for the agnostic setting.[3]

| Algorithm-Independent Distribution-Independent | Algorithm-Dependent Distribution-Independent | Algorithm-Dependent Distribution-Dependent |
| --- | --- | --- |
| ✗

Bounds not uniformly tight | Estimability-learnability trade-off (see Figure 1) | ✓

Bounds can be tight |
| Theorem 1: In the overparameterized setting, any bound is not tight for a large fraction of (algorithm, distribution) combinations. | Theorem 3: In the overparameterized setting, for any specific algorithm there is a trade-off between population loss and estimability. | We suggest that future work focus on generalization bounds for specific combinations of algorithms and distributions. |
| Theorem 2 (quantitative result): For sample size $n \leq d/2$, classes with VC dimension $d$ are not estimable for at least $49\%$ of (algorithm, distribution) combinations. | Theorems 4 and 5 (quantitative results): A learning algorithm cannot simultaneously perform well over the class of linear functions *and* be estimable. | |

Table 1: An overview: when can generalization bounds be tight in the overparameterized setting?

Our first result shows that no hypothesis class $\mathcal{H}$ is estimable in the overparameterized setting:

**Theorem** (**Informal Version of Theorem 2**). *Let $\mathcal{H}$ be a hypothesis class. If $\mathcal{H}$ has VC dimension $d$ and the size of the training set is at most $d/2$, then every estimator $\mathcal{E}$ satisfying Eq. (2) has $\varepsilon \geq 1/8 - o(1)$.*

We emphasize that the lower bound $\varepsilon \geq 1/8 - o(1)$ in Theorem 2 does not hold merely for a single 'pathological' hard distribution. Rather, it holds for many ERMs over a *sizable fraction* of all $\mathcal{H}$-realizable distributions.

We believe Theorem 2 is worthy of attention because it precludes uniform tightness in the overparameterized setting for any generalization bound that depends solely on the training set, the learned hypothesis, and the hypothesis class. Determining which bounds in the literature on neural networks fall within this category is a matter of some debate. This category may arguably include some subset of the following bounds: VC bounds (Bartlett et al., 2019), Rademacher bounds (Bartlett & Mendelson, 2002), bounds based on the spectral norm (Pitas et al., 2017), Frobenius norm (Neyshabur et al., 2015b), path-norm (Neyshabur et al., 2015b), Fisher-Rao norm (Liang et al., 2019), as well as PAC-Bayes-flatness and sharpness-flatness measures (e.g., see the appendices of Jiang et al. (2020) and Dziugaite et al. (2020)), and some compression bounds like Arora et al. (2018). For further details on the aforementioned bounds, see Appendix B.1. We take an expansive position, arguing that the abovementioned bounds, when applied to large neural networks, fall within the framework of Theorem 2, and therefore are not uniformly tight; however, we acknowledge that other scholarly views (mentioned in Section 7) also have merit, and the reader is encouraged to form an independent opinion on this matter.

## 2.2 ALGORITHM-DEPENDENT GENERALIZATION BOUNDS

An important facet of generalization not addressed by the formalism of Eq. (1) involves the choice of the training algorithm. The bound in Eq. (1) depends only on the training set $S$ and the selected hypothesis $\mathcal{A}(S)$, and therefore it cannot capture certain beneficial aspects of the training algorithm. As a simple example, consider the case of a constant algorithm, that ignores the input $S$ and

---

[3]If a bound cannot be uniformly tight even just with respect to realizable distributions, then it definitely cannot be uniformly tight with respect to all distributions (both realizable and not) in the more general agnostic setting. See Appendix E for further details.

always outputs a specific fixed hypothesis $h_0 \in \mathcal{H}$. For such an algorithm, choosing $\mathcal{E}$ such that $\mathcal{E}(\mathcal{A}(S), S) = L_S(h_0)$ yields an excellent estimator of the population loss.[4]

Similarly, in the context of neural networks, it is possible that certain training algorithms like SGD perform 'implicit regularization', or satisfy various stability properties, etc. Therefore there might exist generalization bounds that are tight specifically for these algorithms. The work of Hardt et al. (2016) is a prominent example. We formalize this notion by considering generalization bounds of the form

$$L_{\mathcal{D}}(\mathcal{A}(S)) < L_S(\mathcal{A}(S)) + C(\mathcal{A}, S), \tag{3}$$

where the complexity $C$ depends also on the algorithm $\mathcal{A}$.[5] Eq. (3) leads to the following question:

> **Question 2.** *For which algorithms does there exist a generalization bound of the form of Eq. (3) that is tight for all population distributions in every overparameterized setting?*

Question 2 prompts us to define *algorithm-dependent estimability* (Definition 4) which is analogous to estimability (Eq. (2) and Definition 3), but involves an estimator $\mathcal{E}(\mathcal{A}, S)$ that depends also on the learning algorithm. Our second result establishes a trade-off between learning performance and estimability:

**Theorem (Informal Version of Theorem 3).** *Let $\mathcal{H} \subseteq \mathcal{Y}^{\mathcal{X}}$ be a hypothesis class that is rich enough in a certain technical sense.[6] Then, for any learning algorithm $\mathcal{A}$, at least one of the following conditions does **not** hold:*

1. *$\mathcal{A}$ learns a subset $\mathcal{H}_0 \subseteq \mathcal{H}$ with certain properties.[6]*

2. *$\mathcal{A}$ is algorithm-dependent estimable.*

Theorem 3 states that if an algorithm learns well enough in the sense of Item 1, then the algorithm is not estimable, and this implies that there exists no generalization bound for that algorithm that is tight across all population distributions.

Theorem 3 hinges on the algorithm satisfying Item 1 for a suitable choice of $\mathcal{H}$ and $\mathcal{H}_0$. We emphasize that it is known that this assumption can indeed be satisfied for some large neural network architectures when trained with SGD. For instance, the class of parity functions[7] is one suitable choice for $\mathcal{H}$. It is well known that even a simple fully-connected neural network is expressive enough to represent this class (e.g., Lemma 2 in Nachum & Yehudayoff, 2020). Furthermore, there exist specific network architectures that can provably learn a suitable subset $\mathcal{H}_0$ of the class of parities using SGD (e.g., Theorem 1 in Abbe & Sandon, 2020).

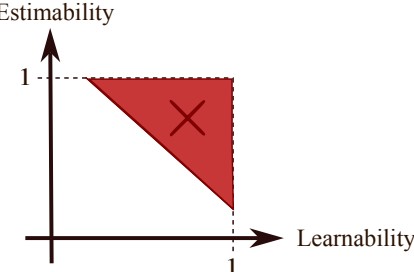

Figure 1: Illustration of the trade-off implied by Theorem 3. '1' represents perfect learning of $\mathcal{H}_0$ and perfect estimation of $\mathcal{A}$'s accuracy. An algorithm cannot simultaneously perform well and be certain that it does so.

To illustrate the utility of Theorem 3, in Section 6.1 we conduct a detailed mathematical study of classes $\mathcal{H}$ that are suitable for use in the theorem, and of the quantitative limitations on the tightness of generalization bounds that these classes entail. Specifically, we show that an algorithm that can learn a suitable subset of the class of parity functions is not estimable with $\varepsilon = 1/4$ (see Theorem 5). More generally, we consider the class of linear functions over finite fields, which is a generalization of parities, and show even stronger results. For example, for a field of size 11 (which corresponds

---

[4]This estimator works only for the constant algorithm that outputs $h_0$, so its existence does not contradict Theorem 2.

[5]In Eq. (3) $C$ receives a complete description of the algorithm $\mathcal{A}$, whereas in Eq. (1) it received $\mathcal{A}(S)$, which is the algorithm's output. Specifically, in Eq. (3), if $\mathcal{A}$ is deterministic then $C$ can compute the value $\mathcal{A}(S)$ using the inputs $\mathcal{A}$ and $S$.

[6]The precise details are specified in the formal version of Theorem 3.

[7]Namely, the class of functions that are an XOR of a fixed subset of the input bits.

to a multiclass classification task with 11 labels), we show that an algorithm that learns a suitable subclass is not estimable with $\varepsilon = 0.45$.

## 3 Related Work

Here we address two works that also study cases where generalization bounds are vacuous. For further related works, see Appendix B.

The main theorems in Nagarajan & Kolter (2019c) (Theorem 3.1) and in Bartlett & Long (2021) (Theorem 1) preclude the existence of tight algorithm-dependent generalization bounds in the overparameterized setting. They show this only for bounds based on uniform convergence and for a linear classifier (a single neuron). Also, Theorem 3.1 and Theorem 1 in Nagarajan & Kolter (2019c); Bartlett & Long (2021) consider the failure over a specific kind of distribution (Gaussian) and specific type of SGD algorithm, and in the proof of Theorem 1, the authors use a different distribution for every sample.

While those results are technically incomparable to ours, our results are more general and stronger in important ways: we show limitations for any kind of algorithm-dependent generalization bound, for many algorithms, and for any architecture in the overparameterized setting while using the same distribution across all sample sizes, with concrete quantitative implications (see Section 6.1).

## 4 Preliminaries

For our standard learning theory notation, see Appendix A.

**Definition 1** (learnability). *Let $\mathcal{H}$ be a hypothesis class and let $\mathbb{D} = \{\mathcal{D}_i\}_{i=1}^T$ a set of $\mathcal{H}$-realizable distributions. $(\mathcal{H}, \mathbb{D})$ is $(\alpha, \beta, n)$-learnable if there exists a (possibly randomized) algorithm $\mathcal{A}$ such that $L_{\mathcal{D}_I}(\mathcal{A}(S)) < \alpha$ with probability at least $1 - \beta$ over $I \sim \mathcal{U}([T])$ and $S \sim \mathcal{D}_I^n$. We say that such an algorithm $(\alpha, \beta, n)$-learns $(\mathcal{H}, \mathbb{D})$.*

Learning with neural networks is often considered an example of an overparameterized setting: in most practical scenarios, the number of parameters of the network greatly exceeds the number of data points $n$ in the training set. Hence, for any given dataset of size $n$, there exist many different sets of weights (or hypotheses) that fit the data. This implies that in the absence of assumptions on the population distribution, the network will overfit in many scenarios. Since we study a general learning setting (the hypothesis class is not necessarily parametrized), we take the above implication as our definition of an overparameterized setting:

**Definition 2** (overparameterized setting). *Let $\mathcal{H}$ be a hypothesis class, let $n, T \in \mathbb{N}$, let $\alpha, \beta \geq 0$, and let $\mathbb{D} = \{\mathcal{D}_i\}_{i=1}^T$ be a finite collection of $\mathcal{H}$-realizable distributions. We say that $(\mathcal{H}, \mathbb{D})$ is an $(\alpha, \beta, n)$-overparameterized setting if $(\mathcal{H}, \mathbb{D})$ is not $(\alpha, \beta, n)$-learnable.*

For a detailed discussion of how this definition compares to some common notions of overparameterization, see Appendix D.

## 5 Bounds that are Algorithm- and Distribution-Independent Cannot be Uniformly Tight

To answer Question 1, we introduce the following framework:

**Definition 3** (estimability). *A hypothesis class $\mathcal{H} \subseteq \mathcal{Y}^{\mathcal{X}}$ is $(\epsilon, \delta, n)$-estimable with respect to a loss function $\ell$ if there exists a function $\mathcal{E} : \mathcal{H} \times S \to \mathbb{R}$ such that for all algorithms $\mathcal{A} : (\mathcal{X} \times \mathcal{Y})^n \to \mathcal{H}$ and all $\mathcal{H}$-realizable distributions $\mathcal{D}$ over $\mathcal{X} \times \mathcal{Y}$ it holds that*

$$\left| \mathcal{E}(\mathcal{A}(S), S) - L_{\mathcal{D}}(\mathcal{A}(S)) \right| < \epsilon$$

*with probability at least $1 - \delta$ over $S \sim \mathcal{D}^n$. We call such $\mathcal{E}$ an $(\epsilon, \delta, n)$-estimator of $\mathcal{H}$.*

Note that given an algorithm-independent and distribution-independent bound $C$, it is not hard to construct a *single* (algorithm, distribution) pair that makes it vacuous. To illustrate this, consider

the ERM defined as $\mathcal{A}_C(S) \coloneqq \arg\min_{h \in \mathcal{H}: L_S(h)=0} C(h,S)$. Assume for simplicity that $C$ is a generalization bound such that Eq. (1) holds with probability 1 for every $\mathcal{H}$-realizable distribution $\mathcal{D}$ with labeling function $h_{\mathcal{D}}$. Then, with probability 1 over $S \sim \mathcal{D}^n$, $L_{\mathcal{D}}(\mathcal{A}_C(S)) < L_S(\mathcal{A}_C(S)) + C(\mathcal{A}_C(S), S) \le C(h_{\mathcal{D}}, S)$, where the first inequality is Eq. (1) and the second follows from the construction of $\mathcal{A}_C$.

Now assume that the setting is overparameterized (say with large $\alpha$ and $\beta$ for a fixed $n$), i.e., no algorithm can learn (with error $\alpha$) the ground-truth labeling with probability at least $1-\beta$ jointly over the uniform choice of the distributions, and $n$ samples from the chosen distribution. This implies that for every algorithm $\mathcal{A}$, there exists at least one distribution $\mathcal{D}'$ such that with probability at least $\beta$, $\mathcal{A}$ fails to learn when $S \sim (D')^n$. Hence there exists a realizable distribution $\mathcal{D}'$ with deterministic labeling function $h_{\mathcal{D}'}$ such that, w.h.p., $L_{\mathcal{D}'}(\mathcal{A}_C(S)) > \alpha$, which implies by the previous inequality that $C(h_{\mathcal{D}'}, S) > \alpha$ w.h.p. But then it holds w.h.p. that $C(h_{\mathcal{D}'}, S) > \alpha \gg 0 = L_{\mathcal{D}'}(h_{\mathcal{D}'}) - L_S(h_{\mathcal{D}'})$. Hence the bound is w.h.p. not $\alpha$-tight for the pair $(\mathcal{A}_{h_{\mathcal{D}'}}, \mathcal{D}')$, where $\mathcal{A}_{h_{\mathcal{D}'}}$ is the constant algorithm that always outputs $h_{\mathcal{D}'}$.

The argument above considers the binary classification setting with the $0-1$ loss and shows lack of tightness for a *single* (algorithm, distribution) pair. In the next theorem, we consider arbitrary hypotheses classes (not necessarily binary) and show that any estimator is not tight for a *large fraction* of possible (algorithm, distribution) pairs.

**Theorem 1.** *Let $\mathcal{H}$ be a hypothesis class, $\ell$ be the $0-1$ loss, $\mathbb{D} = \{\mathcal{D}_i\}_{i=1}^T$ be a finite collection of $\mathcal{H}$-realizable distributions each associated with a hypothesis $h_i$, and $(\mathcal{H}, \mathbb{D})$ an $(\alpha, \beta, n)$ over-parameterized setting. For any $h \in \mathcal{H}$, let $\mathcal{A}_h$ be an ERM algorithm that outputs $h$ for any input sample $S$ consistent with $h$. Then, there exists a distribution $\mathcal{D}_{ERM}$ over $ERM_{\mathcal{H}}$ (the set of all deterministic ERM algorithms over $\mathcal{H}$) such that for any estimator $\mathcal{E}$ of $\mathcal{H}$ and for any $\epsilon, \gamma \in [0,1]$ at least one of the following conditions does **not** hold:*

1. *With probability at least $1 - \gamma$ over $I \sim \mathcal{U}([T])$ and $S \sim \mathcal{D}_I^n$,*

$$|\mathcal{E}(\mathcal{A}_{h_I}(S), S) - L_{\mathcal{D}_I}(\mathcal{A}_{h_I}(S))| < \epsilon.$$

2. *With probability at least $1 - \beta + \gamma$ over $I \sim \mathcal{U}([T])$, $S \sim \mathcal{D}_I^n$, and $\mathcal{A}_{ERM} \sim \mathcal{D}_{ERM}$,*

$$|\mathcal{E}(\mathcal{A}_{ERM}(S), S) - L_{\mathcal{D}_I}(\mathcal{A}_{ERM}(S))| < \alpha - \epsilon.$$

*In particular, $\mathcal{H}$ is not $(\alpha/2, \beta/2, n)$-estimable.*

Theorem 2 below is an application of Theorem 1 for VC classes. Theorem 2 shows that when Theorem 1 is applied, we get substantial numerical values which highlight Theorem 1 prevalence.

**Theorem 2.** *Let $\mathcal{H}$ be a hypothesis class of VC dimension $d \gg 1$, and $\ell$ be the $0-1$ loss. Let $X \subset \mathcal{X}$ be a set of size $d$ shattered by $\mathcal{H}_X = \{h_i\}_{i=1}^{2^d} \subset \mathcal{H}$ and let $\{\mathcal{D}_i\}_{i=1}^{2^d}$ be the set of realizable distributions that correspond to $\mathcal{H}_X$, where for all $i$ the marginal of $\mathcal{D}_i$ on $\mathcal{X}$ is uniform over $X$. Let $\mathrm{ERM}_{\mathcal{H}_X}$ be the set of all deterministic ERM algorithms for $\mathcal{H}_X$. For any $h \in \mathcal{H}_X$, let $\mathcal{A}_h$ be an ERM algorithm that outputs $h$ for any input sample $S$ consistent with $h$. Then, for any estimator $\mathcal{E}$ of $\mathcal{H}$, at least one of the following conditions does **not** hold:*

1. *With probability at least $1/2$ over $I \sim \mathcal{U}([2^d])$ and $S \sim \mathcal{D}_I^n$,*

$$|\mathcal{E}(\mathcal{A}_{h_I}(S), S) - L_{\mathcal{D}_I}(\mathcal{A}_{h_I}(S))| < \frac{d-n}{4d}.$$

2. *With probability at least $1/2 - o(1)$ over $I \sim \mathcal{U}([2^d])$, $S \sim \mathcal{D}_I^n$, and $\mathcal{A}_{ERM} \sim \mathcal{U}(\mathrm{ERM}_{\mathcal{H}_X})$,*

$$|\mathcal{E}(\mathcal{A}_{ERM}(S), S) - L_{\mathcal{D}_I}(\mathcal{A}_{ERM}(S))| < \frac{d-n}{4d} - o(1),$$

   *where $\mathcal{U}(\mathrm{ERM}_{\mathcal{H}_X})$ denotes the uniform distribution over $\mathrm{ERM}_{\mathcal{H}_X}$.*

*In particular, $\mathcal{H}$ is not $\left(\frac{d-n}{4d} - o(1), 1/2 - o(1), n\right)$-estimable for any $n \le d/2$. The notation $o(1)$ denotes quantities that vanish as $d$ goes to infinity.*

Theorem 2 states that any estimator $\mathcal{E}$ fails to predict the performance of many ERM algorithms over many scenarios. If Item 1 does not hold, then $\mathcal{E}$ fails to estimate the success of algorithms $\mathcal{A}_h$ in a situation where they perform very well (since by definition $L_{\mathcal{D}_I}(\mathcal{A}_{h_I}(S)) = 0$), despite the fact that these are very simple algorithms. If Item 2 does not hold, then $\mathcal{E}$ fails to estimate the performance of many ERM algorithms across many distributions, namely, on roughly 50% of (ERM, distribution) pairs.

## 5.1    Discussion of Theorem 2

Theorem 2 shows that bounds that depend solely on the training set, the learned hypothesis, and the hypothesis class cannot be uniformly tight in the overparameterized setting. More broadly, Jiang et al. (2020) showed empirically that many published generalization bounds are in fact not tight. This is an empirical fact that calls for an explanation. Theorem 2 implies that algorithm-independent bounds being not-tight must be a very common phenomenon, in the sense that any algorithm-independent bound that is tight on (a specific set of) simple cases, must be far from tight for many natural algorithms and distributions. Thus, Theorem 2 can at least partially explain the empirical findings of Jiang et al. (2020). To clarify, Theorem 2 does not imply that any specific bound is not tight for a specific (algorithm, distribution) pair, but it qualitatively matches the observation that lack of tightness is ubiquitous among algorithm-independent bounds.

Many published generalization bounds are stated and proved without explicit restrictions on the set of distributions or algorithms. Hence, these bounds are valid upper bounds on the population loss in all scenarios. Due to the limitations on estimability shown in Theorem 2, these bounds cannot fully distinguish between distributions for which the algorithm performs well and distributions for which it does not. Therefore, such bounds have no other option but to predict a large population error (making them loose for some cases in which the population error is not as large).

We emphasize that Theorem 2 shows that VC classes are not estimable, and hence do not admit tight generalization bounds, not merely over some pathological distribution, but in a fairly simple setting. Let us elaborate: take a dataset of natural images $\mathcal{N}$ (e.g., MNIST, CIFAR, etc.) and consider the uniform distribution over this set with a sample size $n = |\mathcal{N}|/2$. This is a simple setting; we merely consider a small set of natural images. Still, Theorem 2 shows that without any assumption on the relation between the images and their labels, any estimator would not perform better than a random guess. That is, any estimator will fail over many ERMs (Item 1 and Item 2 in the theorem) with probability $1/2$ over $I \sim \mathcal{U}([2^d])$ and $S \sim \mathcal{D}_I^n$ to produce an estimate of the true error with an accuracy of $1/8$. Hence, distribution-independent bounds can be loose even in a simple setting.

## 6    Algorithm-Dependent Bounds are Limited by a Learnability-Estimability Trade-Off

To prove Theorem 2, we used constant algorithms, that is, algorithms that output the same hypotheses regardless of the input $S$. The true error of such algorithms is easy to estimate using the empirical error (by Hoeffding's inequality). Thus, it might be that adding the algorithm we use as a parameter for the estimator $\mathcal{E}$ might help. For example, Theorem 2 implies that sharpness-based measures cannot precisely estimate the accuracy of a neural network for all algorithms. Yet, these measures might be a good estimator when specifically considering SGD from random initialization.

**Definition 4** (algorithm-dependent estimability). *A hypotheses class and a collection of algorithms* $(\mathcal{H}, \mathbb{A})$ *is* $(\epsilon, \delta, n)$*-estimable with respect to a loss function* $\ell$ *if there exists a function* $\mathcal{E} : \mathbb{A} \times \mathcal{S} \to \mathbb{R}$ *such that for all algorithms* $\mathcal{A} \in \mathbb{A}$ *and all realizable distributions* $\mathcal{D}$ *over* $\mathcal{X} \times \mathcal{Y}$ *it holds*

$$|\mathcal{E}(\mathcal{A}, S) - L_{\mathcal{D}}(\mathcal{A}(S))| < \epsilon$$

*with probability at least* $1 - \delta$ *over* $S \sim \mathcal{D}^n$. *We call such* $\mathcal{E}$ *an algorithm-dependent estimator of* $\mathcal{H}$. *If* $(\mathcal{H}, \{\mathcal{A}\})$ *is* $(\epsilon, \delta, n)$*-estimable, we say* $\mathcal{A}$ *is* $(\epsilon, \delta, n)$*-estimable with respect to* $\mathcal{H}$ *and loss* $\ell$.

The function $\mathcal{E}$ is now provided with a complete description of the algorithm used to generate a hypothesis. Yet, the following theorem provides a more subtle negative answer to Question 2 compared to Theorem 2. Learnability and estimability are mutually exclusive.

**Theorem 3.** *Let* $\mathcal{H} = \mathcal{H}_0 \cup \mathcal{H}_1$ *be a hypothesis class, let* $\ell$ *be a loss function, and let* $T = T_0 + T_1$ *be integers. Let* $\mathbb{D} = \{\mathcal{D}_i\}_{i=1}^{T}$ *be a set of* $\mathcal{H}$*-realizable distributions such that* $\mathbb{D}_0 = \{\mathcal{D}_i\}_{i=1}^{T_0}$ *is*

*realizable over $\mathcal{H}_0$, and $\mathbb{D}_1 = \{\mathcal{D}_i\}_{i=T_0+1}^{T}$ is realizable over $\mathcal{H}_1$. Assume that $(\mathcal{H}, \mathbb{D})$ is an $(\alpha, \beta, n)$-overparameterized setting, and furthermore assume:*

$$d_{TV}(S_0, S_1) \le 1 - \gamma, \text{ where } S_0 \sim \mathcal{D}_{I_0}^n, \text{ and } S_1 \sim \mathcal{D}_{I_1}^n, I_0 \sim \mathcal{U}([T_0]) \text{ and } I_1 \sim T_0 + \mathcal{U}([T_1]).$$

*Let $\eta = \frac{\gamma}{2} - \frac{1 - \beta \frac{T}{T_1} + \delta\left(1 + \frac{T_0}{T_1}\right)}{2}$. Then, for any learning algorithm $\mathcal{A}$ (possibly randomized), at least one of the following conditions does **not** hold:*

1. *$\mathcal{A}$ $(\epsilon, \delta, n)$-learns $\mathcal{H}_0$.*

2. *$\mathcal{A}$ is $\left(\frac{\alpha - \epsilon}{2}, \eta, n\right)$-estimable with respect to $\mathcal{H}$ and loss $\ell$.*

*In particular, for any estimator $\mathcal{E}$ it holds $|\mathcal{E}(\mathcal{A}, S) - L_{\mathcal{D}_I}(\mathcal{A}(S))| > \frac{\alpha - \epsilon}{2}$ with probability of at least $\eta$ over $I \sim \mathcal{U}([T])$ and $S \sim \mathcal{D}_I^n$.*

As a concrete realization of Theorem 3 with a simple set of parameters, we refer the reader to Theorem 4 in Section 6.1 where we consider multiclass classification with $q$ labels and the $0 - 1$ loss. We show that many algorithms are not $(1/2 - o(1), 1/2 - o(1), n)$-estimable where $o(1)$ is with respect to $q$, and already for $q = 11$, we get $(0.45, 0.4, n)$. Note that $(0.5, 0.5, n)$ is the performance of a random estimator that outputs a random estimation uniformly at random from $[0, 1]$.

In the overparameterized setting, no algorithm can achieve low loss for all distributions $\mathcal{D}$. In this scenario, Theorem 3 shows that if a learning algorithm has a bias towards some part of the hypothesis class, that is, it learns the distributions in $\mathbb{D}_0$ well (a bias towards $\mathcal{H}_0$), then it is necessarily not estimable, even when using complete knowledge of the algorithm being used. So the theorem shows a trade-off between learnability and estimabilty. To see more carefully why, consider the parameters $\alpha, \beta, \gamma, T, T_0, T_1$ to be fixed, as they are not part of the algorithm $\mathcal{A}$ in question but parameters of the overparameterized setting at hand. Then, we observe an affine relation between the accuracy parameter $\epsilon$ and the accuracy parameter for estimating $\mathcal{A}$. An affine relation also holds between the confidence parameter $\delta$ and the confidence parameter for estimating $\mathcal{A}$. This is depicted in Figure 1. In conclusion, an algorithm cannot perform well and be certain of it when it does.

In the following section, we illustrate Theorem 3 and show that it applies to a natural setting that includes neural networks.

## 6.1 QUANTITATIVE LIMITATIONS FOR ALGORITHM-DEPENDENT BOUNDS

To illustrate our results, we conclude our paper with a case study of the hypothesis class of linear functionals $\mathbf{Lin}_q(d)$ over the vector space $\mathbb{F}_q^d$ where $\mathbb{F}_q$ is the finite field with $q$ elements, with $q$ prime. For example, $\mathbf{Lin}_2(d)$ consists of all parity functions with input size $d$.

$$\mathbf{Lin}_q(d) \equiv \left(\mathbb{F}_q^d\right)^* := \left\{f_a : \mathbb{F}_q^d \to \mathbb{F}_q : a \in \mathbb{F}_q^d, \ f_a(x) = \sum_{i=1}^d a_i \cdot x_i \mod q\right\}$$

An important property of this class is presented in the following lemma: each two distinct functions in the class differ exactly on a $\frac{q-1}{q}$ fraction of elements in $\mathbb{F}_q^d$.

**Lemma 1.** *Each two distinct functions $f, h \in \mathbf{Lin}_q(d)$ agree on a fraction $1/q$ of the space and the $0 - 1$ risk of the function $h$ over samples from $\mathcal{D}_f = f \diamond \mathcal{U}(\mathbb{F}_q^d)$ is given by*

$$L_{\mathcal{D}_f}(h) = \begin{cases} 0 & h = f \\ 1 - 1/q & h \ne f \end{cases}.$$

For this class, we consider the following set of algorithms:

**Definition 5.** *Let $\mathcal{X} = \mathbb{F}_q^d$, $\mathcal{Y} = \mathbb{F}_q$, and $\mathcal{H} = \mathbf{Lin}_q(d)$. We say a learning algorithm $\mathcal{A} : (\mathcal{X} \times \mathcal{Y})^* \to \mathcal{H}$ (possibly randomized) is an ERM algorithm with a linear bias if for every sample size $n \le d$, we can associate $(\mathcal{A}, n)$ with a linear subspace $\mathcal{H}_n \subset \mathcal{H}$ of dimension $n$ such that if $|S| = n$ and $S$ is consistent with some function in $\mathcal{H}_n$, then $\mathcal{A}(S) \in \mathcal{H}_n$. We denote by $\mathbb{A}_{lin}$ the set of all ERM algorithms with a linear bias.*

Such choice of algorithms is natural with respect to Theorem 3. They perform well on distributions that are associated with their linear bias. Such algorithms are not estimable, as the following theorem shows. For brevity, we let $F(q, n)$ denote a certain function computed from the rank distribution of a matrix composed of i.i.d. discrete random variables (see the appendix for a precise statement).

**Theorem 4.** *For every $\mathcal{A} \in \mathbb{A}_{lin}$ and every $n \le d - 1$, $\mathcal{A}$ is not $(1/2 - 1/2q, \eta, n)$-estimable with respect to $\mathbf{Lin}_q(d)$ and the $0 - 1$ loss where $\eta = F(q, n)$. In particular, for $q > 10$, it holds that $\eta > 0.4$, and generally, $\eta = 1/2 - 1/q + o(1/q)$.*

Theorem 4 shows that already for multiclass classification with $q = 11$ labels (comparable to the ten labels of MNIST and CIFAR datasets), with probability at most $0.6$ we estimate the performance of an algorithm with an accuracy of $^{10}/_{22} \approx 0.45$. This is not much better than a random guess that estimates with an accuracy of $0.45$ with probability $0.5$. This holds since the loss of any algorithm in $\mathbb{A}_{lin}$ is either $0$ or $1 - 1/q \approx 0.9$, as Lemma 1 shows. So, we can always flip a coin, declare either $0$ or $0.9$, and be correct with probability $0.5$. Finally, as $q$ grows, our estimation can truly only be as good as a random guess.

When we consider the case of $q = 2$ (parity functions) for Theorem 4, we get that the algorithms we consider are only $(1/4, 0.025)$-not estimable. To stengthen our results, the following theorem provides a separate analysis that includes a different technique from Theorem 3 and that works specifically for $q = 2$.

**Theorem 5.** *For every $\mathcal{A} \in \mathbb{A}_{lin}$ (possibly randomized) and every $n \le d-1$, $\mathcal{A}$ is not $(0.25, 0.14, n)$-estimable with respect to $\mathbf{Lin}_2(d)$ and the $0 - 1$ loss. More so, for every deterministic $\mathcal{A} \in \mathbb{A}_{lin}$ and every $6 \le n \le d$, $\mathcal{A}$ is not $(0.25, 0.32, n)$-estimable with respect to $\mathbf{Lin}_2(d)$ and the $0 - 1$ loss.*

It might appear that Theorems 4 and 5 are unrelated to practical overparameterized models because their setting is too artificial. However, as mentioned above, neural network can represent parities (e.g., Lemma 2 in Nachum & Yehudayoff, 2020) and, in some specific cases, can learn parities using SGD (Abbe & Sandon, 2020). Hence, these theorems are relevant at least to some neural networks.

## 7 CONCLUSIONS

We have proved mathematically that specific types of generalization bounds are subject to specific limitation in a precise formal sense. The interpretation of these formal results, and the extent to which they apply to existing generalization bounds in the literature on large neural networks, is a matter of some debate. It is possible to argue that our results do not apply to various generalization bounds in the literature, e.g., because our definition of an overparameterized setting doesn't hold in cases of interest, because those bounds do not satisfy our definitions of distribution-independence or algorithm-independence, or for other reasons. Alternatively, one could argue that uniform-tightness is not an important property for a generalization bound, and that for specific practical cases of interest, it is easy to tell empirically whether a bound is tight or not when applying it to a trained model. Below we outline our view, but we also acknowledge that a variety of scholarly positions exist. We encourage the reader to develop an independent opinion on this matter.

In our view, Theorems 2 and 3 point to two possibilities for obtaining uniformly tight generalization bounds in overparameterized settings. The first option is that when stating a generalization bound, the statement explicitly specifies a set of 'nice' or 'natural' population distributions for which the bound is tight. Thus, the 'bad' population distributions on which the bound is not tight are clearly excluded from the set of distributions for which the bound is intended to work. The second option for obtaining a tight generalization bound is to make explicit assumptions about the learning algorithm, which in particular imply that for *any* choice of classes $\mathcal{H}, \mathcal{H}_0$ suitable for Theorem 3, the algorithm *cannot* learn $\mathcal{H}_0$. We suggest that every proposal of a generalization bound for the overparameterized setting explicitly include one of these two types of assumptions. Otherwise, if the setting is overparameterized, there provably exist pairs of learning algorithms and population distributions for which the bound applies and is valid, but is not tight. See Appendix C for further illustrations.

Explicitly stating the assumptions underlying generalization bounds is not only necessary for the bounds to be uniformly tight in the overparameterized setting, but can also promote more clarity within the scientific community, and guide future research.

ACKNOWLEDGEMENTS

MG, IN and TW were supported in part by the Swiss National Science Foundations under Grant Nos. 200364 and 200020_182517. Part of this work was done while JS was at UC Berkeley, supported by DARPA contract # HR001120C0015 and the Simons Collaboration on The Theory of Algorithmic Fairness. Part of this work was done while JS was visiting the Technion – Israel Institute of Technology, hosted by Shay Moran. Views and opinions expressed are those of the authors only and do not necessarily reflect those of the Swiss National Science Foundations, DARPA, or the Simons Foundation.

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

## A  NOTATION

Following is a summary of the standard learning theory notation used in this paper.

| | |
|---:|:---|
| $\mathcal{X}$ | the set of possible inputs (the domain) |
| $\mathcal{Y}$ | the set of possible labels |
| $\mathcal{D}$ | a distribution over $\mathcal{X} \times \mathcal{Y}$ |
| $\mathcal{D}_{\mathcal{X}}$ | the marginal distribution of $\mathcal{D}$ on $\mathcal{X}$ |
| $\mathbb{D}$ | a set of distributions over $\mathcal{X} \times \mathcal{Y}$ |
| $\mathcal{H} \subseteq \mathcal{Y}^{\mathcal{X}}$ | a hypotheses class |
| $S = ((x_1, y_1), ..., (x_n, y_n)) \sim \mathcal{D}^n$ | an i.i.d. sample or a training set |
| $(\mathcal{X} \times \mathcal{Y})^* = \cup_{k=1}^{\infty} (\mathcal{X} \times \mathcal{Y})^k$ | the set off all possible finite samples |
| $\mathcal{A} : (\mathcal{X} \times \mathcal{Y})^* \to \mathcal{Y}^{\mathcal{X}}$ | a learning algorithm (possibly randomized) |
| $\mathbb{A}$ | a set of learning algorithms |
| $\ell : \mathcal{H} \times \mathcal{X} \times \mathcal{Y} \to \mathbb{R}$ | a loss function |
| $\ell_{0-1}(h, x, y) = \mathbf{1}_{h(x) \neq y}$ | the 0-1 loss function |
| $L_S(h) \coloneqq \frac{1}{n} \sum_{i=1}^{n} \ell(h, x_i, y_i)$ | the empirical loss of $h$ on the sample $S$ |
| $L_{\mathcal{D}}(h) \coloneqq \mathbb{E}_{(x,y) \sim \mathcal{D}} \ell(h, x, y)$ | the population loss of $h$ with respect to distribution $\mathcal{D}$ |
| $f \diamond \mathcal{D}_{\mathcal{X}}$ | the distribution of the random variable $(X, f(X))$ where $X \sim \mathcal{D}_{\mathcal{X}}$ |
| $\mathcal{U}(\Omega)$ | the uniform distribution over a set $\Omega$ |
| $[m]$ | the set $\{1, 2, \ldots, m\}$ |
| $\mathbb{F}_q$ | the finite field with $q$ elements, where $q$ is prime |
| $\mathbf{Lin}_q(d)$ | the set of all linear functionals over $\mathbb{F}_q^d$ |
| $R_q(n_1, n_2, r)$ | the probability of an $n_1$ times $n_2$ random matrix with entries drawn i.i.d. uniformly at random from $\mathbb{F}_q$ to have rank $r$ (see Lemma 2) |
| $\begin{bmatrix} n \\ k \end{bmatrix}_q$ | the Gaussian coefficient $\prod_{i=0}^{k-1} \frac{q^{n-i}-1}{q^{k-i}-1}$ |

For simplicity, all spaces we consider are **finite**.

**Definition 6.** *A distribution $\mathcal{D}$ is* realizable *with respect to a hypothesis class $\mathcal{H}$ if there exists $h \in \mathcal{H}$ such that $L_{\mathcal{D}}(h) = 0$.*

**Note:** in the case of a random algorithm $\mathcal{A}$, we define $L_{\mathcal{D}}(\mathcal{A}(S)) \coloneqq \mathbb{E}_{h \sim Q(S)} L_{\mathcal{D}}(h)$ where $Q(S)$ is the posterior distribution over $\mathcal{Y}^{\mathcal{X}}$ that $\mathcal{A}$ outputs. This fits the PAC-Bayes framework that upper bounds this quantity.

## B  FURTHER RELATED WORKS

### B.1  EXAMPLES OF GENERALIZATION BOUNDS THAT ARE DISTRIBUTION-INDEPENDENT AND ALGORITHM-INDEPENDENT

Our definition of an estimator $\mathcal{E}$ captures the formalism of generalization bounds that are distribution-independent *and* algorithm-independent. Following are some bounds from the literature that, in our view, satisfy these independence requirements.

1. **VC bounds:** Vapnik & Chervonenkis (1971); Bartlett et al. (1998; 2019). For a class $\mathcal{H}$ of VC dimension $d$, the complexity measure is $\Theta\left(\frac{d+\log(1/\delta)}{n}\right)$. The complexity term depends solely on the hypothesis class (for fixed $\epsilon$, $\delta$, and $n$).

2. **Rademacher bounds:** Mohri et al. (2012). $\text{Rad}_{\mathcal{H}}(S) = \frac{1}{n}\mathbb{E}_{\sigma}\sup_{f\in\mathcal{H}}\sum_{i=1}^{n}\sigma_i f(x_i)$ where $\sigma \sim \mathcal{U}\{\pm 1\}^n$. This yields the complexity measure $2 \cdot \text{Rad}_{\mathcal{H}}(S) + 4\sqrt{\frac{2\ln(2/\delta)}{n}}$. The complexity term measures how rich the hypothesis class $\mathcal{H}$ is with respect to $S$.

3. **Norm-based and margin-based bounds:** Liang et al. (2017); Neyshabur et al. (2015a); Nagarajan & Kolter (2019b); Pitas et al. (2017); Bartlett & Mendelson (2002); Bartlett et al. (2017); Neyshabur et al. (2019); Golowich et al. (2018); Neyshabur et al. (2018; 2015c). Norm-based bounds typically consist of a complexity term that depends on the algorithm's output. For example, $\sum_{i=1}^{L}|W_i|_F^2$ is the sum of the Frobenius norms of all the neural network layers. Margin-based bounds quantify a measure of separation for the induced representation of the training set.

4. **PAC-Bayes with a fixed prior:** McAllester (1999). This yields the complexity measure $\sqrt{\frac{D(Q\|P)+\ln(n/\delta)}{2(n-1)}}$ where $P$ is a fixed distribution (prior) over the hypothesis class and $Q$ is the output distribution of a randomized algorithm.

5. **Sharpness-based measures:** Nagarajan & Kolter (2019a); Neyshabur et al. (2017); Keskar et al. (2017). These measures (inspired by the PAC-Bayes framework) quantify how flat the solution generated by the algorithm is with respect to the empirical loss. For example, $1/\sigma^2$ where $\sigma = \max\left\{\alpha : \mathbb{E}_{W\sim\mathcal{N}(w,\alpha I)}L_S(f_W(x)) < 0.05\right\}$, $w$ is a weight vector of the algorithm's output, $W$ is a random perturbation of $w$, and $f_W$ is the hypothesis realized by the network's architecture with respect to $W$. $C$ depends both on $\mathcal{A}(S)$ and $S$. Such measures received considerable attention as candidates to explain neural network generalization (see Dziugaite et al. 2020).

In all these bounds, the explicit expression in the bound depends solely on the hypothesis class, the selected hypothesis, and the training set. They do not explicitly depend on any property of the learning algorithm or the population distribution. Hence, in our view, they are subject to Theorems 1 and 2.

## B.2    EXAMPLES OF GENERALIZATION BOUNDS THAT ARE DISTRIBUTION-INDEPENDENT AND ALGORITHM-DEPENDENT

The following two works present algorithm-dependent bounds.

Zhang et al. (2023): The paper studies convex optimization, so the results can hold only for a single neuron. Nevertheless, although it gives matching lower and upper bounds, the bounds match only asymptotically when $n$ is very large so the scenario is far away from the overparameterized regime (which is the focus of interest for neural networks).

Nikolakakis et al. (2023): The paper suggests a new generalization bound that is an algorithm-dependent bound that does not include any distributional assumptions (it is distribution-independent).

In our view, Theorem 3 implies that these bounds cannot be tight for all population distributions.

## B.3    EXAMPLES OF GENERALIZATION BOUNDS THAT ARE DISTRIBUTION-DEPENDENT AND ALGORITHM-DEPENDENT

The following are information-theoretic generalization bounds that are both algorithm and distribution-dependent (which we advocate for in this paper). However, bounds are sometimes hard to approximate numerically in a tight manner (for example the bound in Theorem 1 in Xu & Raginsky (2017)). On a high level, such bounds are part of the PAC Bayes framework; see proof 4 for Theorem 8 in Bassily et al. (2018), which is equivalent to Theorem 1 in Xu & Raginsky (2017). Unfortunately, when the PAC-Bayes/information-theoretic bounds can be approximated in a tight manner such as in Issa et al. (2019); Esposito et al. (2021); Harutyunyan et al. (2021); Hellström

& Durisi (2022); Wang & Mao (2023); Dziugaite et al. (2021); Haghifam et al. (2022); Issa et al. (2023), they do not reveal what properties of the pair (distribution, algorithm) allowed for such success of learning and estimation (so the utility they offer is similar to that of a validation set).

## C  EXAMPLE APPLICATIONS

Understanding how our formal results relate to generalization bounds in the literature on large neural networks, and to their use in practical settings, is a nontrivial question. In this appendix we illustrate our position via two examples, but we also recognize that other positions are possible.

### C.1  MARGIN BOUNDS

There are many margin- and norm-based bounds in the literature (see Appendix B.1). We start with a toy example, illustrating why these bounds are not uniformly tight, and why we believe that Theorem 2 applies to these types of bounds.

For simplicity, let the domain $\mathcal{X} = \{x \in \mathbb{R}^d : \|x\|_2 = 1\}$ be the unit sphere in $\mathbb{R}^d$, and consider a half-space classifier with hypothesis class $\mathcal{H} = \{h_w : w \in \mathcal{X}\}$, where $h_w(x) = \text{sign}(\langle w, x \rangle)$.

For a fixed sample size $n$, consider a margin bound of the form $f = f(\gamma)$, where $\gamma$ is the minimum distance of a point in the training set from the learned decision boundary. $f(\gamma)$ is an upper bound on the difference between the population and empirical losses. $f(\gamma)$ is small only if $\gamma$ is large. We will show that $f$ is not uniformly tight.

Indeed, consider a population distribution $\mathcal{D}$ that has a uniform marginal on the domain $\mathcal{X}$, with labels that are generated by a specific half-space $h^* \in \mathcal{H}$. Consider an algorithm $A$ that has a bias towards $h^*$, in the sense that it will output $h^*$ whenever the training set is consistent with $h^*$.

In this case, given samples from $\mathcal{D}$, with probability $1$, $A$ will output $h^*$. Hence, the population loss and the empirical loss are the same (they are both exactly $0$). Because the marginal of the population distribution on the domain is uniform, the population margin is $0$, and therefore the empirical margin $\gamma$ will be close to $0$ (even for a sample size $n$ that is small compared to the dimension $d$). Thus, $f(\gamma)$ is large, but the generalization gap is $0$. Hence, the bound $f(\gamma)$ is not tight for the pair $(\mathcal{D}, A)$.

Note that $f(\gamma)$ is a quantity that depends only on the selected hypothesis and on the training set. Namely, this is a distribution- and algorithm-independent bound. Hence, Theorem 2 implies a stronger limitation, in the sense that it shows that there exist many (distribution, algorithm) pairs for which the bound is not tight, not just a single pair. The same holds for bounds of the form $f(\|W\|, \gamma)$, where $\|W\|$ is some norm-like function of the parameters of the learned classifier.

### C.2  PAC-BAYES BOUNDS

The contributions of this paper can be illustrated by applying them, for example, to the landmark work of Dziugaite & Roy (2017). They presented PAC-Bayes bounds for neural networks that were the first generalization bounds to be non-vacuous in some real-world overparameterized settings. However, that paper did not state formal assumptions on the set of population distributions or the set of algorithms to which it is intended to apply. Therefore, our results imply the following for any fixed choice of a countable collection of priors as in the bound of Dziugaite & Roy (2017). First, by Theorem 2, the bounds is not tight for roughly half of all (distribution, algorithm) pairs. Second, by Theorem 3, for any specific learning algorithm, if the algorithm can learn a suitable set of functions $\mathcal{H}_0$, then the bound is not tight on a sizable collection of population distributions.

One way to understand the forgoing example is that the non-vacuous numerical bound presented for the MNIST dataset by Dziugaite & Roy (2017) relies on implicit assumptions about the algorithm and the population distribution (for other tasks different priors would be required), such as: 'the population distribution satisfies that with high probability, SGD (on the specific network architecture of interest) finds a flat global minimum point (with respect to the empirical loss) in the parameter space not too far away from the initialization point.' Or alternatively, 'SGD (on the specific network architecture of interest) cannot learn any collection of functions $\mathcal{H}_0$ that is suitable for Theorem 3.' Laid out explicitly in this manner, it becomes clear that these assumptions are not obviously true. This invites further research to understand for which population distributions and algorithms the

assumptions hold, and whether there might exist more natural and compelling assumptions that suffice for obtaining bounds of comparable tightness.

## D    ON DEFINITIONS OF OVERPARAMETERIZATION

There are a number of definitions of overparameterization that are common in the machine learning literature, but there does not appear to be a single formal definition that is universally agreed upon. One contribution of this paper is that we offer an alternative formal definition of overparameterization that is close to well-known notions, and also enables proving meaningful mathematical theorems. In this appendix we discuss our definition and its connections to some common definitions.

Following are three common definitions. In these definitions, there is a learning task in which a machine learning system (e.g., a neural network) is trained using a training set of $n$ labeled samples. The hypothesis class $\mathcal{H}$ is the collection of classification functions that the machine learning system can represent. We state these definition informally, in the sense some value is required to be large without specifying exact quantities.

> **Definition A.** *The number of independently-tunable parameters in the machine learning system is significantly larger than the number of samples in the training set. Namely, $\mathcal{H} = \{h_w : w \in \mathbb{R}^k\}$ and $k \gg n$.*
>
> **Definition B.** *The learning system can interpolate arbitrary data. Namely, $\mathcal{H}$ can realize any labeling for any distinct $x_1, \ldots, x_m$, for some $m \gg n$.*
>
> **Definition C.** *The size of the training set is smaller than the VC dimension, namely, $\mathsf{VC}(\mathcal{H}) \gg n$.*

For convenience, our definition of overparameterization is restated here. As before, one should think of $\mathcal{H}$ as the collection of classification functions that are realizable by a learning system of interest.

**Definition 2** (Restated). *Let $\mathcal{H}$ be a hypothesis class, let $n, T \in \mathbb{N}$, let $\alpha, \beta \geq 0$, and let $\mathbb{D} = \{\mathcal{D}_i\}_{i=1}^T$ be a finite collection of $\mathcal{H}$-realizable distributions. We say that $(\mathcal{H}, \mathbb{D})$ is an $(\alpha, \beta, n)$-overparameterized setting if $(\mathcal{H}, \mathbb{D})$ is not $(\alpha, \beta, n)$-learnable (as in Definition 1).*

Our definition is more general than Definitions A, B and C in that our definition is (typically) implied by the other definitions. This means that when we prove that a bound cannot be tight in the overparameterized setting according to our definition, that in particular implies that it cannot be tight in any setting that is overparameterized according to the other definitions. This makes our impossibility results stronger.

The implications hold as follows:

> - Definition A $\implies$ Definition C. This implication holds for many neural networks. It is well-known that large neural networks have large VC dimension. For instance, Theorem 1 in Maass (1994) states that under mild assumptions, any neural network with at least 3 layers has VC dimension linear in the number of edges in the network. Maass (1994) showed this for networks with threshold activation functions and binary inputs, and Bartlett et al. (1998) note that "It is easy to show that these results imply similar lower bounds" for networks with ReLU activation. A similar result holds also for networks with 2 layers (Sakurai, 1993).
> - Definition B $\implies$ Definition C. If a class $\mathcal{H}$ can realize any labeling for some $x_1, \ldots, x_m$ then the VC of $\mathcal{H}$ is at least $m$.
> - Definition C $\implies$ Definition 2. This implication is Item 2 in the proof of Theorem 2 on page 21.

Our definition does not merely generalize the common definitions listed above — it generalizes them in a desirable way. Definitions B and C pertains to a case where the hypothesis class can express every possible labeling of the training set or of a shattered set. However, this seems a bit rigid. What if the network or hypothesis class can express every labeling except one? Or can express most but not all labelings? Intuitively, the network is still very much overparameterized, even if there are some specific choices of labelings that it cannot express. The essence of overparameterization

is that the network can express too many of the possible labelings — not necessarily every single one of them. What is 'too many'? Our answer, as captured in Definition 2, is that a network is overparameterized (can express 'too many' labelings) if it can typically express many different labelings that are consistent with the training set but disagree on the test set, leading to poor expected performance on the test set.

In conclusion, Definition 2 is natural, impossibility results for Definition 2 imply impossibility results for the other definitions, and Definition 2 generalizes the other definitions in a desirable way.

We end with two additional remarks concerning definitions of overparmetrization.

1. Note that Definition A also has the weakness that it is sometimes possible to parameterize the same hypothesis class with varying numbers of parameters. For instance, a 1-layer fully-connected network with linear activations and a multi-layer fully-connected network with linear activations represent the same hypothesis class (both networks simply multiply the input vector by a matrix), but the number of parameters can be very different between these two networks.

2. A curious reader might wonder why we do not use the definition of PAC learning in Definition 2. That is, why not say that $\mathcal{H}$ is an $(\alpha, \beta, n)$-overparameterized setting if $\mathcal{H}$ is not $(\alpha, \beta, n)$-PAC-learnable. Working with such a definition will yield quantitatively weaker results. For example, an analogue of Theorem 1 (that uses the aforementioned definition) will still prove that VC classes are not estimable, so any estimator fails for at least one combination of an algorithm and a distribution; our definition yields the stronger result of Theorem 1, that the failure of each estimator is across many combinations of algorithms and distributions.

## E  EQUIVALENCE BETWEEN ESTIMABILITY AND EXISTENCE OF TIGHT GENERALIZATION BOUNDS

**Definition 7** (Uniformly tight generalization bound). *Let $\mathcal{X}$ and $\mathcal{Y}$ be sets, let $\mathcal{H} \subseteq \mathcal{Y}^{\mathcal{X}}$, let $\epsilon, \delta > 0$ and let $n \in \mathbb{N}$. Let $C : \mathcal{H} \times (\mathcal{X} \times \mathcal{Y})^n \to \mathbb{R}$ be a measure of complexity as in Eq. (1). We say that $C$ yields an $(\epsilon, \delta, n)$-uniformly tight generalization bound for $\mathcal{H}$ if for every algorithm $\mathcal{A}$ that always outputs a hypothesis from $\mathcal{H}$ and for every $\mathcal{H}$-realizable distribution $\mathcal{D}$,*

$$\mathbb{P}_{S \sim \mathcal{D}^n}\Big[L_S(\mathcal{A}(S)) + C(\mathcal{A}(S), S) - \epsilon \le L_{\mathcal{D}}(\mathcal{A}(S)) \le L_S(\mathcal{A}(S)) + C(\mathcal{A}(S), S)\Big] \ge 1 - \delta. \quad (4)$$

Definition 7 captures generalization bounds that are tight in the realizable setting. More generally, we are interested in generalization bounds that are tight in the more realistic agnostic setting. However, because we are interested in showing that certain types of generalization bounds cannot be tight, it suffices to show that they cannot be tight in the realizable case, as in Definition 7.

We now show an equivalence (up to a factor of 2) between estimability (Definition 3) and the existence of uniformly tight generalization bounds (Definition 7). From this equivalence, to show that certain generalization bounds cannot be tight, it suffices to show that a class is not estimable.

**Claim 1.** *Let $\mathcal{X}$ and $\mathcal{Y}$ be sets, let $\mathcal{H} \subseteq \mathcal{Y}^{\mathcal{X}}$, let $\epsilon, \delta > 0$ and let $n \in \mathbb{N}$.*

1. *If there exists $C : \mathcal{H} \times (\mathcal{X} \times \mathcal{Y})^n \to \mathbb{R}$ that yields an $(\epsilon, \delta, n)$-uniformly tight generalization bound for $\mathcal{H}$ then $\mathcal{H}$ is $(\epsilon, \delta, n)$-estimable.*

2. *If $\mathcal{H}$ is $(\epsilon, \delta, n)$-estimable then there exists $C : \mathcal{H} \times (\mathcal{X} \times \mathcal{Y})^n \to \mathbb{R}$ that yields a $(2\epsilon, \delta, n)$-uniformly tight generalization bound for $\mathcal{H}$.*

*Proof.* For Item 1, let $C : \mathcal{H} \times (\mathcal{X} \times \mathcal{Y})^n \to \mathbb{R}$ be a measure of complexity that yields an $(\epsilon, \delta, n)$-uniformly tight generalization bound for $\mathcal{H}$. Let $\mathcal{A}$ be an algorithm that always outputs a hypothesis in $\mathcal{H}$ and let $\mathcal{D}$ be an $\mathcal{H}$-realizable distribution. Then Eq. (4) holds. Define $\mathcal{E} : \mathcal{H} \times (\mathcal{X} \times \mathcal{Y})^n \to \mathbb{R}$ by

$$\mathcal{E}(h, S) = L_S(h) + C(h, S).$$

Plugging the definition of $\mathcal{E}$ into Eq. (4) yields

$$\mathbb{P}_{S\sim\mathcal{D}^n}\left[\left|L_{\mathcal{D}}(\mathcal{A}(S)) - \mathcal{E}(\mathcal{A}(S),S)\right| \leq \epsilon\right] \geq 1-\delta. \tag{5}$$

This shows that $\mathcal{H}$ is $(\epsilon,\delta,n)$-estimable.

For Item 2, assume that $\mathcal{H}$ is $(\epsilon,\delta,n)$-estimable and let $\mathcal{E}$ be an estimator that witnesses this. Let $\mathcal{A}$ be an algorithm that always outputs a hypothesis in $\mathcal{H}$ and let $\mathcal{D}$ be an $\mathcal{H}$-realizable distribution. By choice of $\mathcal{E}$, Eq. (5) holds. Define $C: \mathcal{H} \times (\mathcal{X} \times \mathcal{Y})^n \to \mathbb{R}$ by

$$C(h,S) = \mathcal{E}(h,S) - L_S(h) + \epsilon.$$

Plugging $C$ into Eq. (5) yields that with probability at leat $1-\delta$ over the sample $S \sim \mathcal{D}^n$,

$$L_{\mathcal{D}}(\mathcal{A}(S)) \leq \mathcal{E}(\mathcal{A}(S),S) + \epsilon = L_S(h) + C(h,S)$$

and also

$$L_{\mathcal{D}}(\mathcal{A}(S)) \geq \mathcal{E}(\mathcal{A}(S),S) - \epsilon = L_S(h) + C(h,S) - 2\epsilon.$$

Hence, $C$ yields a $(2\epsilon,\delta,n)$-uniformly tight generalization bound for $\mathcal{H}$. $\qquad\square$

## F    PROOFS OF THEOREM 1 AND THEOREM 2

Theorem 2 is a corollary of the more general Theorem 1 which shows that in the overparameterized setting, any estimator of the true loss will fail over many combinations of ERM algorithms and distributions.

Before the proof we present the type of distributions over $ERM$ algorithms Theorem 1 applies for.

We remind the reader that an $ERM$ algorithm is a function $\mathcal{A}: (\mathcal{X} \times \mathcal{Y})^* \to \mathcal{Y}^{\mathcal{X}}$ whose output is any consistent function with the sample $S$ that belongs to $\mathcal{H}$.

**Definition 8** (Bayes-like Random ERM). *Let $\mathcal{H}$ be a hypothesis class and $\mathbb{D} = \{\mathcal{D}_i\}_{i=1}^T$ a set of distributions. We say that the distribution $\mathcal{D}_{ERM}$ over $ERM$ algorithms is Bayes-like if for any fixed $S$ it holds that*

$$P_{\mathcal{D}_{ERM}}\left(\mathcal{A}_{ERM}(S) = h\right) = \frac{\sum_{i:\, h_i = h} P_{\mathcal{D}_i}(S)}{\sum_{j=1}^T P_{\mathcal{D}_j}(S)}$$

*where $h_i$ is the labeling function associated with the distribution $\mathcal{D}_i$. If $\mathcal{A}_{ERM} \sim \mathcal{D}_{ERM}$, we say $\mathcal{A}_{ERM}$ is Bayes-like Random ERM.*

Reminder: All spaces we consider in the paper are finite so the definition of Bayes-like Random ERM is a well-defined random variable over a finite sample space.

Clearly, any algorithm $\mathcal{A}$ chosen with non-zero probability over $\mathcal{D}_{ERM}$ is an $ERM$ algorithm.

**Theorem 1.** *Let $\mathcal{H}$ be a hypothesis class, $\ell$ be the $0-1$ loss, $\mathbb{D} = \{\mathcal{D}_i\}_{i=1}^T$ be a finite collection of $\mathcal{H}$-realizable distributions each associated with a hypothesis $h_i$, and $(\mathcal{H},\mathbb{D})$ an $(\alpha,\beta,n)$ overparameterized setting. For any $h \in \mathcal{H}_X$, let $\mathcal{A}_h$ be an ERM algorithm that outputs $h$ for any input sample $S$ consistent with $h$. Then, for any Bayes-like distribution $\mathcal{D}_{ERM}$ over ERM algorithms and any estimator $\mathcal{E}$ of $\mathcal{H}$ at least one of the following conditions does not hold:*

1. *With probability at least $1-\gamma$ over $I \sim \mathcal{U}([T])$ and $S \sim \mathcal{D}_I^n$,*

$$\left|\mathcal{E}\left(\mathcal{A}_{h_I}(S),S\right) - L_{\mathcal{D}_I}(\mathcal{A}_{h_I}(S))\right| < \epsilon.$$

2. *With probability at least $1-\beta+\gamma$ over $I \sim \mathcal{U}([T])$, $S \sim \mathcal{D}_I^n$, and $\mathcal{A}_{ERM} \sim \mathcal{D}_{ERM}$,*

$$\left|\mathcal{E}\left(\mathcal{A}_{ERM}(S),S\right) - L_{\mathcal{D}_I}(\mathcal{A}_{ERM}(S))\right| < \alpha - \epsilon,$$

.

*In particular, $\mathcal{H}$ is not $(\alpha/2,\beta/2,n)$-estimable.*

*Proof.* To see why $\mathcal{H}$ is not $(\alpha/2, \beta/2, n)$-estimable if item 1 or item 2 do not hold, choose $\gamma := \beta/2$ and $\epsilon := \alpha/2$.

If item 1 does not hold, it means there exists an algorithm $\mathcal{A}_{h_i}$ and a distribution $\mathcal{D}_i$ such that

$$|\mathcal{E}\left(\mathcal{A}_{h_i}(S), S\right) - L_{\mathcal{D}_i}(\mathcal{A}_h(S))| \geq \epsilon$$

with probability greater than $\beta/2$ over $\mathcal{D}_i$ so $\mathcal{H}$ is not $(\alpha/2, \beta/2, n)$-estimable.

If item 2 does not hold, it means there exists an ERM algorithm $\mathcal{A}_{ERM}$ and a distribution $\mathcal{D}_i$ such that

$$|\mathcal{E}\left(\mathcal{A}_{ERM}(S), S\right) - L_{\mathcal{D}_i}(\mathcal{A}_h(S))| \geq \alpha - \epsilon$$

with probability greater than $\beta/2$ over $\mathcal{D}_i$ so $\mathcal{H}$ is not $(\alpha/2, \beta/2, n)$-estimable.

Let $\mathcal{A}_{ERM} \sim \mathcal{D}_{ERM}$ be a Bayes-like Random $ERM$. The key ingredient in our proof is to show that the following two probability measures $P_1$ and $P_2$ over $(\mathcal{X} \times \mathcal{Y})^n \times \mathcal{H}$ are equal so $P_1(S, h) = P_2(S, h)$ for every pair $(S, h) \in (\mathcal{X} \times \mathcal{Y})^n \times \mathcal{H}$.

1. $P_1$: the pair $(S_I, h_I)$ is generated by choosing an index $I \sim \mathcal{U}[T]$, then $S_I \sim \mathcal{D}_I$, and $h_I$ is the labeling function associated with $\mathcal{D}_I$.

2. $P_2$: the pair $(S_I, \mathcal{A}_{ERM}(S_I))$ is generated by choosing an index $I \sim \mathcal{U}[T]$, then $S_I \sim \mathcal{D}_I$, and $\mathcal{A}_{ERM} \sim \mathcal{D}_{ERM}$ independently.

Equalities (4) - (8) and equalities (9) - (14) show that $P_1$ and $P_2$ are indeed equal.

$$P_1(S, h) = \sum_{i=1}^{T} P(I = i) \cdot P(S|I = i) \cdot P(h_I = h|I = i, S) \tag{6}$$

$$= \sum_{i=1}^{T} P(I = i) \cdot P(S|I = i) \cdot P(h_I = h|I = i) \tag{7}$$

$$= \frac{1}{T} \sum_{i=1}^{T} P(S|I = i) \cdot P(h_I = h|I = i) \tag{8}$$

$$= \frac{1}{T} \sum_{i:\, h_i = h} P(S|I = i) \tag{9}$$

$$= \frac{1}{T} \sum_{i:\, h_i = h} P_{\mathcal{D}_i}(S) \tag{10}$$

The first equality holds by law of total probability, the second equality holds since $h_I$ is independent of $I$ given $S$, the third equality holds since $I$ is uniformly distributed, the fourth equality holds since $P(h_I = h|I = i)$ is either 1 or 0, and the fifth equality holds by the definition of $\mathcal{D}_i$.

$$P_2(S, h) = \sum_{j=1}^{T} P(I = j) \cdot P(S|I = j) \cdot P(\mathcal{A}_{ERM}(S) = h|I = j, S) \tag{11}$$

$$= \sum_{i=1}^{T} P(I = j) \cdot P(S|I = j) \cdot P(\mathcal{A}_{ERM}(S) = h|S) \tag{12}$$

$$= \frac{1}{T} \sum_{j=1}^{T} P(S|I = j) \cdot P(\mathcal{A}_{ERM}(S) = h|S) \tag{13}$$

$$= \frac{1}{T} \sum_{j=1}^{T} P(S|I = j) \cdot \frac{\sum_{i: h_i = h} P_{\mathcal{D}_i}(S)}{\sum_{i=1}^{T} P_{\mathcal{D}_j}(S)} \tag{14}$$

$$= \frac{1}{T} \frac{\sum_{i: h_i = h} P_{\mathcal{D}_i}(S)}{\sum_{i=1}^{T} P_{\mathcal{D}_j}(S)} \sum_{j=1}^{T} P_{\mathcal{D}_i}(S) \tag{15}$$

$$= \frac{1}{T} \sum_{i: h_i = h} P_{\mathcal{D}_i}(S) \tag{16}$$

The first equality holds by law of total probability, the second equality holds since $\mathcal{A}_{ERM}(S)$ is independent of $I$ given $S$, the third equality holds since $I$ is uniformly distributed, the fourth equality holds by the definition of the distribution $\mathcal{D}_{ERM}$, and the fifth and sixth equalities are trivial.

Assume item 1 in the theorem holds: $|\mathcal{E}(\mathcal{A}_{h_I}(S), S) - L_{\mathcal{D}_I}(\mathcal{A}_{h_I}(S))| < \epsilon$ with probability $1 - \gamma$ over $I \sim \mathcal{U}[T]$ and $S \sim \mathcal{D}_I$. Since $L_{\mathcal{D}_I}(\mathcal{A}_{h_I}(S)) = 0$, we have that $\mathcal{E}(\mathcal{A}_{h_I}(S), S) < \epsilon$ with probability $1 - \gamma$ over $I \sim \mathcal{U}[T]$ and $S \sim \mathcal{D}_I$.

So, on the one hand, we have

$$\mathcal{E}(h, S) < \epsilon$$

with probability $1 - \gamma$ over the probability measure $P_1$.

Since we are in an $(\alpha, \beta, n)$-overparameterized setting, then for any algorithm $\mathcal{A}$ we have that $L_{\mathcal{D}_I}(\mathcal{A}(S)) > \alpha$ with probability at least $\beta$ over $I \sim \mathcal{U}[T]$ and $S \sim \mathcal{D}_I$.

So, on the other hand, we have

$$L_{\mathcal{D}_I}(\mathcal{A}_{ERM}(S)) > \alpha$$

with probability of at least $\beta$ over $I \sim \mathcal{U}[T]$, $S \sim \mathcal{D}_I$, and $\mathcal{A}_{ERM} \sim \mathcal{D}_{ERM}$.

Since $P_1 = P_2$ we can combine the two statements above about $P_1$ and $P_2$ and have by the union bound that

$$|\mathcal{E}(\mathcal{A}_{ERM}(S), S) - L_{\mathcal{D}_I}(\mathcal{A}_{ERM}(S))| > \alpha - \epsilon,$$

holds with probability of at least $\beta - \gamma$ over $I \sim \mathcal{U}[T]$, $S \sim \mathcal{D}_I$, and $\mathcal{A}_{ERM} \sim \mathcal{D}_{ERM}$. So item 2 in the theorem cannot hold which concludes the proof.

$\square$

We now use Theorem 1 and the definition of a Bayes-like random $ERM$ to prove Theorem 2.

**Theorem 2.** *Let $\mathcal{H}$ be a hypothesis class of VC dimension $d \gg 1$, and $\ell$ be the $0 - 1$ loss. Let $X \subset \mathcal{X}$ be a set of size $d$ shattered by $\mathcal{H}_X = \{h_i\}_{i=1}^{2^d} \subset \mathcal{H}$ and let $\{\mathcal{D}_i\}_{i=1}^{2^d}$ be the set of realizable distributions that correspond to $\mathcal{H}_X$, where for all $i$ the marginal of $\mathcal{D}_i$ on $\mathcal{X}$ is uniform over $X$. Let $\mathrm{ERM}_{\mathcal{H}_X}$ be the set of all deterministic ERM algorithms for $\mathcal{H}_X$. For any $h \in \mathcal{H}_X$, let $\mathcal{A}_h$ be an ERM algorithm that outputs $h$ for any input sample $S$ consistent with $h$. Then, for any estimator $\mathcal{E}$ of $\mathcal{H}$, at least one of the following conditions does not hold:*

1. *With probability at least $1/2$ over $I \sim \mathcal{U}([2^d])$ and $S \sim \mathcal{D}_I^n$,*

$$|\mathcal{E}(\mathcal{A}_{h_I}(S), S) - L_{\mathcal{D}_I}(\mathcal{A}_{h_I}(S))| < \frac{d - n}{4d}.$$

2. *With probability at least $1/2 - o(1)$ over $I \sim \mathcal{U}([2^d])$, $S \sim \mathcal{D}_I^n$, and $\mathcal{A}_{ERM} \sim \mathcal{U}(\mathrm{ERM}_{\mathcal{H}_X})$,*

$$\left| \mathcal{E}\left(\mathcal{A}_{ERM}(S), S\right) - L_{\mathcal{D}_I}(\mathcal{A}_{ERM}(S)) \right| < \frac{d-n}{4d} - o(1),$$

*where $\mathcal{U}(\mathrm{ERM}_{\mathcal{H}_X})$ denotes the uniform distribution over $\mathrm{ERM}_{\mathcal{H}_X}$.*

*In particular, $\mathcal{H}$ is not $\left(\frac{d-n}{4d} - o(1), 1/2 - o(1), n\right)$-estimable for any $n \leq d/2$. The notation $o(1)$ denotes quantities that vanish as $d$ goes to infinity.*

*Proof.* The proof is an application of Theorem 1 with the following two items:

1. $\mathcal{D}_{ERM} = \mathcal{U}[\mathrm{ERM}_{\mathcal{H}_X}]$ is a Bayes-like distribution over $ERM$ algorithms for $\{\mathcal{D}_i\}_{i=1}^{2^d}$,.

2. For any algorithm $\mathcal{A}$, with probability $1 - o(1)$ over $I \sim \mathcal{U}([2^d])$ and $S \sim \mathcal{D}_I^n$, it holds that

$$L_{\mathcal{D}_I}(\mathcal{A}(S)) > \frac{d-n}{2d} - o(1).$$

To apply Theorem 1 we have from Item 2 that $\alpha = \frac{d-n}{2d} - o(1)$ and $\beta = 1 - o(1)$ for any $n \leq d/2$.

Proof for Item 1:

On the one hand, $P_{\mathcal{D}_{ERM}}(\mathcal{A}_{ERM}(S) = h) = \frac{1}{|\{h: h \text{ is consistent with } S\}|}$ for $h$ that is consistent with $S$ and $P_{\mathcal{D}_{ERM}}(\mathcal{A}_{ERM}(S) = h) = 0$ otherwise. This follows from the definition of $\mathcal{U}[\mathrm{ERM}_{\mathcal{H}_X}]$.

On the other hand, for any fixed $h$ we have the quantity $\frac{\sum_{i: h_i = h} P_{\mathcal{D}_i}(S)}{\sum_{j=1}^T P_{\mathcal{D}_j}(S)}$. The summands in the denominator are non-zero only for indices that correspond to functions $h_i$ that are consistent with $S$. All such summands are equal since we have the same underlying distribution over the space $\mathcal{X}$. Since all the functions $\{h_i\}_{i=1}^{2^d}$ are different, the sum in the numerator contains one summand which equals any other summand in the denominator if $h$ is consistent with $S$ and equals $0$ otherwise.

Combining the two statements above completes the proof:

$$P_{\mathcal{D}_{ERM}}(\mathcal{A}_{ERM}(S) = h) = \frac{\sum_{i: h_i = h} P_{\mathcal{D}_i}(S)}{\sum_{j=1}^T P_{\mathcal{D}_j}(S)}.$$

Proof for Item 2:

Let $\mathcal{A}$ be an ERM algorithm. We note that for every realizable $S$ over $\mathcal{H}_X$ that consists of $m$ distinct data points we have

$$\mathbb{E}_I L_{\mathcal{D}_I}(\mathcal{A}(S)) = \frac{d-m}{2d}. \tag{17}$$

where $I \sim U[2^d]$.

More so, for a fixed $S$ that consist of $m$ distinct data points, $L_{\mathcal{D}_I}(\mathcal{A}(S))$ is a random variable which is a sum of $d - m$ i.i.d. Bernoulli random variables with parameter $1/2$ that is divided by $d$. By Hoeffding's inequality,

$$\mathbb{P}\left(L_{\mathcal{D}_I}(\mathcal{A}(S)) < \frac{d-m}{2d} - \frac{d^{0.6}}{d}\right) < \exp\left(-\frac{2d^{1.2}}{d-m}\right) < \exp(-2d^{0.2}),$$

which implies that for any $S$ with $|S| = n$ it holds that

$$\mathbb{P}\left(L_{\mathcal{D}_I}(\mathcal{A}(S)) < \frac{d-n}{2d} - o(1)\right) < o(1)$$

Then, with probability $1 - o(1)$ over $I \sim \mathcal{U}([2^d])$ and $S \sim \mathcal{D}_I^n$, it holds that

$$L_{\mathcal{D}_I}(\mathcal{A}(S)) > \frac{d - n}{2d} - o(1).$$

$\square$

## G  PROOF OF THEOREM 3

We start with a remark.

**Remark.** *The converse of Theorem 3 is false. Over the same overparameterized setting as in Theorem 3, an algorithm can simultaneously not learn $\mathcal{H}_0$ well and be not estimable. To see why, consider binary classification with the $0 - 1$ loss and take any algorithm $\mathcal{A}$ that $(\epsilon, \delta)$-learns $\mathcal{H}_0$ and define $\mathcal{A}_{neg}(S) = \overline{\mathcal{A}(S)}$ where $\bar{h}$ is the hypothesis with opposite labels to $h$. $\mathcal{A}_{neg}$ does not $(1 - \epsilon, 1 - \delta, n)$-learn $\mathcal{H}_0$ and by the same arguments as in Theorem 3, we get that it is also not $\left( \frac{\alpha - \epsilon}{2}, \frac{\gamma}{2} - \frac{1 - \beta \frac{T}{T_1} + \delta \left( 1 + \frac{T_0}{T_1} \right)}{2}, n \right)$-estimable.*

*Proof of Theorem 3.* If $(\mathcal{H}, \mathbb{D})$ is an $(\alpha, \beta, n)$-over parametrized setting and $\mathcal{A}$ $(\epsilon, \delta, n)$-learns $(\mathcal{H}, \mathbb{D}_0)$, then $\mathcal{A}$ does not $\left( \alpha, 1 - \frac{\beta T}{T_1} + \frac{\delta T_0}{T_1}, n \right)$-learn $(\mathcal{H}, \mathbb{D}_1)$. This stems from the following steps.

$$
\begin{aligned}
\beta &\leq \mathbb{E}_{S,I} \mathbf{1}_{L_{\mathcal{D}_I}(\mathcal{A}(S)) \geq \alpha} \\
&= \frac{T_0}{T} \mathbb{E}_{S,I_0} \mathbf{1}_{L_{\mathcal{D}_{I_0}}(\mathcal{A}(S)) \geq \alpha} + \frac{T_1}{T} \mathbb{E}_{S,I_1} \mathbf{1}_{L_{\mathcal{D}_{I_1}}(\mathcal{A}(S)) \geq \alpha} \\
&\leq \frac{T_0}{T} \mathbb{E}_{S,I_0} \mathbf{1}_{L_{\mathcal{D}_{I_0}}(\mathcal{A}(S)) \geq \epsilon} + \frac{T_1}{T} \mathbb{E}_{S,I_1} \mathbf{1}_{L_{\mathcal{D}_{I_1}}(\mathcal{A}(S)) \geq \alpha} \\
&\leq \frac{T_0}{T} \delta + \frac{T_1}{T} \mathbb{E}_{S,I_1} \mathbf{1}_{L_{\mathcal{D}_{I_1}}(\mathcal{A}(S)) \geq \alpha} \\
&= \frac{T_0}{T} \delta + \frac{T_1}{T} \mathbb{P} \left( L_{\mathcal{D}_{I_1}}(\mathcal{A}(S)) \geq \alpha \right)
\end{aligned}
$$

The first inequality holds by the definition of the overparameterized setting. The first equality from the law of total expectation. The second inequality holds because decreasing $\alpha$ to $\epsilon$ only increase the probability that the indicator function outputs 1. The third inequality holds since $\mathcal{A}$ $(\epsilon, \delta, n)$-learns $(\mathcal{H}, \mathbb{D}_0)$. The second equality holds since the expectation of the indicator function equals the probability for the event the indicator function underscores to happen.

This yields that with probability at least $\frac{\beta T}{T_1} - \frac{\delta T_0}{T_1}$ over $I_1 \sim T_0 + U[T_1]$ and $S \sim \mathcal{D}_{I_1}^n$

$$L_{\mathcal{D}_{I_1}}(\mathcal{A}(S)) \geq \alpha.$$

Now we show that $\mathcal{A}$ is not $\left( \frac{\alpha - \epsilon}{2}, \eta, n \right)$-estimable with respect to $\mathcal{H}$ and loss $\ell$. For that end, we use Theorem 1 in Angel & Spinka (2021). We denote as $\omega$ the coupling between $S_0 \sim \mathcal{D}_{I_0}^n$ and $S_1 \sim \mathcal{D}_{I_1}^n$. We have that with probability $\gamma$ over $\omega$, $S_0 = S_1$. Now, for any estimator $\mathcal{E}$, we get our claim through the following steps where $B = \left\{ S_0 = S_1, L_{D_{I_0}}(\mathcal{A}(S_0)) < \epsilon, L_{D_{I_1}}(\mathcal{A}(S_1)) > \alpha \right\}$.

$$\mathbb{E}_{\omega}\left[\mathbf{1}_{\left|\mathcal{E}(\mathcal{A},S_0)-L_{D_{I_0}}(\mathcal{A}(S_0))\right|\geq\frac{\alpha-\epsilon}{2}}+\mathbf{1}_{\left|\mathcal{E}(\mathcal{A},S_1)-L_{D_{I_1}}(\mathcal{A}(S_1))\right|\geq\frac{\alpha-\epsilon}{2}}\right]=$$

$$\mathbb{P}(B)\,\mathbb{E}_{\omega|B}\left[\mathbf{1}_{\left|\mathcal{E}(\mathcal{A},S_0)-L_{D_{I_0}}(\mathcal{A}(S_0))\right|\geq\frac{\alpha-\epsilon}{2}}+\mathbf{1}_{\left|\mathcal{E}(\mathcal{A},S_1)-L_{D_{I_1}}(\mathcal{A}(S_1))\right|\geq\frac{\alpha-\epsilon}{2}}\right]+$$

$$\mathbb{P}(B^c)\,\mathbb{E}_{\omega|B^c}\left[\mathbf{1}_{\left|\mathcal{E}(\mathcal{A},S_0)-L_{D_{I_0}}(\mathcal{A}(S_0))\right|\geq\frac{\alpha-\epsilon}{2}}+\mathbf{1}_{\left|\mathcal{E}(\mathcal{A},S_1)-L_{D_{I_1}}(\mathcal{A}(S_1))\right|\geq\frac{\alpha-\epsilon}{2}}\right]\geq$$

$$\mathbb{P}(B)\,\mathbb{E}_{\omega|B}\left[\mathbf{1}_{\left|\mathcal{E}(\mathcal{A},S_0)-L_{D_{I_0}}(\mathcal{A}(S_0))\right|\geq\frac{\alpha-\epsilon}{2}}+\mathbf{1}_{\left|\mathcal{E}(\mathcal{A},S_1)-L_{D_{I_1}}(\mathcal{A}(S_1))\right|\geq\frac{\alpha-\epsilon}{2}}\right]\geq$$

$$\mathbb{P}(S_0=S_1,L_{D_{I_0}}(\mathcal{A}(S_0))<\epsilon,L_{D_{I_1}}(\mathcal{A}(S_1))\geq\alpha)\geq$$

$$\gamma-\delta-\left(1-\frac{\beta T}{T_1}+\frac{\delta T_0}{T_1}\right)=$$

$$2\eta$$

The first equality holds by the law of total expectation for any event $B$. The first inequality holds since the expectations are over non-negative random variables. The second inequality holds since we assigned $B=\left\{S_0=S_1,L_{D_{I_0}}(\mathcal{A}(S_0))<\epsilon,L_{D_{I_1}}(\mathcal{A}(S_1))>\alpha\right\}$ and when $B$ occurs for we have $\mathcal{E}(\mathcal{A},S_0)=\mathcal{E}(\mathcal{A},S_1)$ and any such assignment of $\mathcal{E}(\mathcal{A},S_0)$ forces at least one of the indicator functions to take value 1. The third inequality holds by the union bound.

The proof now concludes since

$$\mathbb{E}_{I_0,S_0}\mathbf{1}_{\left|\mathcal{E}(\mathcal{A},S_0)-L_{D_{I_0}}(\mathcal{A}(S_0))\right|\geq\frac{\alpha-\epsilon}{2}}=\mathbb{E}_{\omega}\mathbf{1}_{\left|\mathcal{E}(\mathcal{A},S_0)-L_{D_{I_0}}(\mathcal{A}(S_0))\right|\geq\frac{\alpha-\epsilon}{2}}$$

and

$$\mathbb{E}_{I_1,S_1}\mathbf{1}_{\left|\mathcal{E}(\mathcal{A},S_1)-L_{D_{I_1}}(\mathcal{A}(S_1))\right|\geq\frac{\alpha-\epsilon}{2}}=\mathbb{E}_{\omega}\mathbf{1}_{\left|\mathcal{E}(\mathcal{A},S_1)-L_{D_{I_1}}(\mathcal{A}(S_1))\right|\geq\frac{\alpha-\epsilon}{2}}$$

so at least one of the following items holds:

- With probability at least $\eta$ over $I_0$ and $S_0$ it holds that

$$\left|\mathcal{E}(\mathcal{A},S_0)-L_{D_{I_0}}(\mathcal{A}(S_0))\right|\geq\frac{\alpha-\epsilon}{2}.$$

- With probability at least $\eta$ over $I_1$ and $S_1$ it holds that

$$\left|\mathcal{E}(\mathcal{A},S_1)-L_{D_{I_1}}(\mathcal{A}(S_1))\right|\geq\frac{\alpha-\epsilon}{2}.$$

$\square$

## H  FINITE FIELDS AND LIMITATIONS ON ESTIMABILITY IN MULTICLASS CLASSIFICATION

Let us first revise Def. 5 from the main text, since it is a preliminary for the proofs to follow:

**Definition 5.** *Let $\mathcal{X}=\mathbb{F}_q^d$, $\mathcal{Y}=\mathbb{F}_q$, and $\mathcal{H}=\mathbf{Lin}_q(d)$. We say a learning algorithm $\mathcal{A}:(\mathcal{X}\times\mathcal{Y})^*\to\mathcal{H}$ (possibly randomized) is an ERM algorithm with a linear bias if for every sample size $n\leq d$, we can associate $(\mathcal{A},n)$ with a linear subspace $\mathcal{H}_n\subset\mathcal{H}$ of dimension $n$ such that if $|S|=n$ and $S$ is consistent with some function in $\mathcal{H}_n$, then $\mathcal{A}(S)\in\mathcal{H}_n$. We denote by $\mathbb{A}_{lin}$ the set of all ERM algorithms with a linear bias.*

We restate the definition of linear functions and add further definitions:

**Definition 9.**

- *Let $q$ be prime and denote by $\mathbb{F}_q$ the finite field with $q$ elements. The hypothesis class of linear functionals $\mathbf{Lin}_q(d)$ over the vector space $\mathbb{F}_q^d$ is defined as*

$$\mathbf{Lin}_q(d) \equiv \left(\mathbb{F}_q^d\right)^* := \left\{ f_a : \mathbb{F}_q^d \to \mathbb{F}_q : a \in \mathbb{F}_q^d, \ f_a(x) = \sum_{i=1}^d a_i \cdot x_i \mod q \right\}.$$

- *We partition $\mathbf{Lin}_q(d)$ into the linear subspaces $\mathbf{Lin}_{q,i}(d) := \{f_a \in \mathbf{Lin}_q(d) : a_1 = i\}$ with $|\mathbf{Lin}_{q,i}(d)| = q^{d-1}$ and $\mathbf{Lin}_{q,i}(d,n) := \{f_a \in \mathbf{Lin}_q(d) : (a_1 = i) \wedge (\forall j \in [n+2, \dots, d] : a_j = 0)\}$, where $i \in \{0, \dots q-1\}$ and $|\mathbf{Lin}_{q,i}(d,n)| = q^n$.*

- *We denote by $X \in \mathbb{F}_q^{n \times d}$ the matrix obtained by stacking the inputs $\{x_i\}_{i=1}^n$ as rows.*

- *We denote by $e_i$ the canonical basis vector with entry $1$ in position $i$ and $0$ everywhere else.*

- *Further, we denote by $\mathbb{A}_i \subset \mathbb{A}_{lin}$ the class of ERMs that are biased towards $\mathbf{Lin}_{q,i}(d,n)$. In more detail: whenever $\mathcal{A} \in \mathbb{A}_i$ receives a sample $S$ of size $n$ and there exist hypotheses in $\mathbf{Lin}_{q,i}(d,n)$ consistent with the labels of $S$, $\mathcal{A}$ outputs one of them. Otherwise $\mathcal{A}$ outputs an arbitrary hypothesis from $\mathbf{Lin}_q(d,n)$.*

## H.1 PROOF OF LEMMA 1

**Lemma 1.** *Each two distinct functions $f, h \in \mathbf{Lin}_q(d)$ agree on a fraction $1/q$ of the space and the $0-1$ risk of the function $h$ over samples from $\mathcal{D}_f = f \diamond \mathcal{U}(\mathbb{F}_q^d)$ is given by*

$$L_{\mathcal{D}_f}(h) = \begin{cases} 0 & h = f \\ 1 - 1/q & h \neq f \end{cases}.$$

*Proof.* Denote the coefficient vectors of $f, h$ by $a_f, a_h$, respectively. Moreover, denote by $\mathcal{X}_a \subseteq \mathcal{X}$ the subset of the domain on which $f$ and $h$ agree, i.e., the fraction over which they agree is $|\mathcal{X}_a|/|\mathcal{X}|$. Then, for any given input $x \in \mathbb{F}_q^d$, $f(x) = h(x)$ if and only if $c \cdot x = 0 \mod q$ where $c := a_f - a_h \mod q$.

Now assume w.l.o.g. that the respective last entries of $a_h$ and $a_f$ are distinct (if not, perform an appropriate permutation). Then, the first $d-1$ entries of $x$ can be chosen arbitrarily and there are $q^{d-1}$ distinct such choices. Thereafter, the last entry $x_d$ is fixed to be $c_d \cdot x_d = q - \sum_{i=1}^{d-1} c_i \cdot x_i \mod q$. Note that for $q$ prime such an $x_d \in \mathbb{F}_q$ always exists and is unique. In summary, we have that $|\mathcal{X}_a| = q^{d-1}$ which together with $|\mathcal{X}| = q^d$ gives the first desired claim.

The second claim then simply follows from observing that $x \sim \mathcal{U}(\mathcal{X})$ together with the fact that we are using the $0-1$ loss. $\square$

## H.2 LEMMA 2

**Lemma 2.** *The probability that an $n_1 \times n_2$ matrix with coefficients drawn i.i.d. from $\mathcal{U}(\{0, \dots, q-1\})$ has rank $r$ is given by*

$$R_q(n_1, n_2, r) = \begin{bmatrix} n_2 \\ r \end{bmatrix}_q \sum_{l=0}^r (-1)^{r-l} \begin{bmatrix} r \\ l \end{bmatrix}_q q^{n_1(l-n_2) + \binom{r-l}{2}} \tag{18}$$

*where $\begin{bmatrix} n \\ k \end{bmatrix}_q := \prod_{i=0}^{k-1} \frac{q^{n-i}-1}{q^{k-i}-1}$ denote the so-called Gaussian coefficients.*

*In particular, the probability that an $n_1 \times n_2$ matrix with coefficients drawn i.i.d. from $\mathcal{U}(\{0, \dots, q-1\})$ has full rank is given by*

$$R_q(n_1, n_2, c_1) = \prod_{k=0}^{c_1-1} (1 - q^{k-c_2}). \tag{19}$$

*where $c_1 := \min\{n_1, n_2\}$ and $c_2 := \max\{n_1, n_2\}$*

*Proof.* The first statement is a Corollary of (Blake & Studholme, 2006, Corollary 2.2).

The second statement can be inferred from the first, but also follows directly from the following simple iterative construction: assume w.l.o.g. that $n_1 \leq n_2$. Then, at each time $t = 0, \ldots n_1 - 1$ we add one row to the matrix. Assuming that each entry of the matrix is sampled i.i.d. uniformly at random, the probability that at time $t$ the new row is linearly independent of all previous rows is then given by $1 - q^{t-n_2}$ since there are $q^t$ possible linear combinations of $t$ rows, and there are $q^{n_2}$ vectors of length $n_2$ in total. □

### H.3 Lemma 3

**Lemma 3.** *Denote by $X^- \in \mathbb{F}_q^{n \times n+1}$ the matrix obtained from $X \in \mathbb{F}_q^{n \times d}$ by dropping all but the first $n + 1$ columns. Denote $k := rank(X^-)$ and assume that the labels $y$ of the samples $S = (X, y)$ are generated by some $f \in \textbf{Lin}_{q,i}(d, n)$, i.e., $y = f(X)$ where $f : \mathbb{F}_q^{n \times d} \to \mathbb{F}_q^n$ is understood to be applied row-wise. Then,*

- *If the vector $e_1$ is spanned by the rows of $X^-$, there exist $q^{n+1-k}$ functions in $\textbf{Lin}_{q,i}(d, n)$ consistent with $S$ and no consistent function in all other classes $\textbf{Lin}_{q,j}(d, n)$, $j \neq i$.*

- *If the vector $e_1$ is not spanned by the rows of $X^-$, there exist $q^{n-k}$ functions consistent with $S$ in each $\textbf{Lin}_{q,j}(d, n)$, $j \in \{0, \ldots, q - 1\}$.*

*Proof.* Let $f_a$ be a linear functions parametrized by coefficient vectors $a = [a_1, \ldots, a_d]$. Further, let $a' \in \mathbb{F}_q^{n+1}$ denote the truncated coefficient vectors of $a$ over the first $n + 1$ coordinates. Assume first that $e_1$ is spanned by the rows of $X^-$. Then, only functions in $\textbf{Lin}_{q,i}(d, n)$ can be consistent with $S$. This is because $a'_1 = c^\top \cdot X^- \cdot a' = c^\top \cdot y$ is uniquely determined for $c \in \mathbb{F}_q^n$ if $c$ is chosen such that it encodes the linear combination of rows of $X^-$ that yields $e_1$. The preceding equation hence holds iff $a_1 = i$ due to the assumption that $f \in \textbf{Lin}_{q,i}(d, n)$. Since the rank of $X^-$ is $k$, its null space has dimension $n + 1 - k$ and there exist $q^{n+1-k}$ consistent functions in total because $X^- \cdot (a + b) = y$ is also consistent for all $b \in \text{null}(X^-)$ assuming that the data was generated by $f_a$.

Assume now that the rows of $X^-$ do not span $e_1$. Then, we can construct the reduced matrix $X_{red} \in \mathbb{F}_q^{k \times (n+1)}$ and a vector $y_{red} \in \mathbb{F}_q^k$ by retaining only $k = \text{rank}(X^-)$ linearly independent rows of $X^-$ and the corresponding entries of $y$. Note that since the sample $S$ is generated by some $f \in \textbf{Lin}_{q,i}(d, n)$, we have that $X^- \cdot a' = y$ if and only if $X_{red} \cdot a' = y_{red}$. Hence it is sufficient to work with the reduced system of equations in order to check for the consistency of a function $f_a$.

Next, we append above the top of the matrix $X_{red}$ the row $e_1$ and $n - k$ further canonical basis vectors $e_{i_1}, \ldots, e_{i_{n-k}}$ in order to obtain a full rank matrix $X_{ext}$. This is always possible because $e_1$ is not spanned by assumption and furthermore there always exist at least $n + 1 - k$ canonical basis vectors which are not spanned by the rows of $X_{red}$.

With this setup, one can then obtain for given $X$ and $y$ a function $f_h \in \textbf{Lin}_{q,m}(d, n)$ with $m \in \{0, \ldots, q - 1\}$ arbitrary and reduced coefficient vector $h' := [m \ \hat{h}^\top]^\top \in \mathbb{F}_q^{n+1}$ by computing the unique solution of the fully determined system of equations

$$\begin{bmatrix} e_1 \\ e_{i_1} \\ \vdots \\ e_{i_{n-k}} \\ X_{red} \end{bmatrix} \begin{bmatrix} m \\ \hat{h} \end{bmatrix} = \begin{bmatrix} m \\ z \\ y_{red} \end{bmatrix} \tag{20}$$

for $z \in \mathbb{F}_q^{n-k}$ arbitrarily chosen. By construction, the function $f_h$ with $h = [h', 0, \ldots, 0]^\top$ will then be in $\textbf{Lin}_{q,m}(d, n)$ and be consistent with the sample $(X, y)$. Moreover, since $z \in \mathbb{F}_q^{n-k}$ can be chosen freely, there are $q^{n-k}$ such functions in each class $\textbf{Lin}_{q,j}(d, n)$, $j \in \{0, \ldots, q - 1\}$. □

### H.4 Proof of Theorem 4

We consider the learning settings with realizable distributions over $\mathcal{H}' := \mathcal{H}_0 \cup \mathcal{H}_1$ where $\mathcal{H}_0 := \textbf{Lin}_{q,0}(d, n)$ and $\mathcal{H}_1 := \textbf{Lin}_{q,1}(d, n)$. The algorithms we consider come from $\mathbb{A}_0$, the class of

ERMs that are biased to $\mathcal{H}_0 = \mathbf{Lin}_{q,0}(d,n)$, that is, if there exists a consistent hypothesis in $\mathcal{H}_0$, the algorithm must choose one such hypothesis. As mentioned previously, it is sufficient to consider such ERMs in order to show learnability and estimability results over the larger class of linearly constrained ERMs $\mathbb{A}_{lin}$ due to a symmetry argument.

We first show in Lemma 4 that the learnability condition (condition 1 in Theorem 3) holds for $\mathcal{H}_0$. In Lemma 5 we show that attempting to learn above $\mathcal{H}'$ corresponds to an overparameterized setting. It then follows Theorem 4 then follows from an upper bound on the TV-distance which implies that condition 2 in Theorem 3 cannot hold, i.e., $\mathcal{H}'$ is not estimable. Since $\mathbf{Lin}_q(d) \supseteq \mathcal{H}'$, this directly implies that $\mathcal{H} = \mathbf{Lin}_q(d)$ is also not estimable with the same parameters.

**Lemma 4.** *$\mathbf{Lin}_{q,0}(d,n)$ is $(0,\delta,n)$-learnable over $\mathbb{D}_0 = \{f \diamond \mathcal{U}(\mathcal{X}) | f \in \mathbf{Lin}_{q,0}(d,n)\}$ with any $\mathcal{A} \in \mathbb{A}_0$ for $n \in [d-1]$ and $\delta > \sum_{k=0}^{n-1}(1 - q^{k-n}) \cdot R_q(n,n,k)$ where $R_q$ is defined as in Lemma 2.*

*Proof.* For the analysis one can drop the columns $1, n+2, n+3, \ldots d$ of $X$ as hypotheses in $\mathbf{Lin}_{q,0}(d,n)$ are invariant w.r.t. to these coordinates and it is sufficient for $\mathcal{A}$ to only check for consistency. Thereby we obtain the reduced input matrix $X'$ of dimensions $n \times n$.

Clearly, any $\mathcal{A} \in \mathbb{A}_0$ outputs the ground truth hypothesis w.p. 1 as soon as $X'$ has full rank. On the other hand, once $X'$ does not have full rank, $\mathcal{A}$ might output the wrong hypothesis depending on its preference, i.e., depending on which of the multiple consistent functions it outputs. It can be easily verified that any preference (be it deterministic or random) of $\mathcal{A}$ amongst consistent hypotheses leads to the same error probability since the random variable $I$ that selects the labeling function is uniformly distributed.

Setting $\epsilon = 0$, we get by the law of total probability that $\mathbf{Lin}_{q,0}(d,n)$ is $(0,\delta,n)$-learnable by any $\mathcal{A} \in \mathbb{A}_0$ if

$$\delta > \mathbb{P}(\{\mathcal{A}_{\mathrm{ERM}}(S) \neq f\}) \tag{21}$$

$$= \sum_{k=0}^{n-1} \mathbb{P}(\{\mathcal{A}_{\mathrm{ERM}}(S) \neq f\} | \{X' \text{ has rank } k\}) \cdot \mathbb{P}(\{X' \text{ has rank } k\}) \tag{22}$$

$$= \sum_{k=0}^{n-1}(1 - q^{k-n}) \cdot R_q(n,n,k) \tag{23}$$

where for the last equality we used the following two facts:

Conditioned on the event that $\mathrm{rank}(X') = k$, $\mathcal{A}$ returns the ground truth with probability $q^{k-n}$. This is because $\mathbb{P}_f(\{\mathcal{A}(S) = f\} | S)$ is uniform over all $f \in \mathbf{Lin}_{q,0}(d,n)$ consistent with $S$ and since there are $q^{n-k}$ functions consistent with $k$ linearly independent samples (see the argument in Lemma 3).

For controlling the probability of rank deficiency we used the result stated in Lemma 2. $\qquad\square$

**Lemma 5.** *$\mathbf{Lin}_{q,0}(d,n) \cup \mathbf{Lin}_{q,1}(d,n)$ is not $(\alpha,\beta,n)$-learnable with any $\mathcal{A} \in \mathbb{A}_0$ over $\mathbb{D} = \{f \diamond \mathcal{U}(\mathcal{X}) | f \in \mathbf{Lin}_{q,0}(d,n) \cup \mathbf{Lin}_{q,1}(d,n)\}$ for $n \in [d-1]$ and any $(\alpha,\beta)$ such that simultaneously $\alpha < (q-1)/q$ and $\beta < \frac{1}{2} \sum_{k=0}^{n}\left((q^{k-n} - 2q^{k-n-1})\frac{q^k-1}{q^{n+1}-1} + 2 - q^{k-n}\right) \times R_q(n,n+1,k)$.*

*Proof.* Note that learnability is trivial for $\alpha \geq (1 - 1/q) = \frac{q-1}{q}$ since this is an upper bound for the risk of linear hypotheses according to Lemma 1. Hence we consider the case $\alpha < (q-1)/q$. Then, by definition, $(\alpha,\beta)$-learnability is precluded for any algorithm that outputs a linear hypothesis once $\beta < \mathbb{P}(\{\mathcal{A}(S) \neq f\})$ since $\{\mathcal{A}(S) \neq f\}$ implies $L_{\mathcal{D}_f}(\mathcal{A}(S)) = \frac{q-1}{q}$. In the sequel we aim to derive $\mathbb{P}(\{\mathcal{A}(S) \neq f\})$ for the case where $\mathcal{A} \in \mathbb{A}_0$.

Denote by $X^- \in \mathbb{F}_q^{n \times (n+1)}$ the matrix obtained by dropping all but the first $n+1$ columns of $X$. Denoting the events $E_1 := \{\mathcal{A}(S) \neq f\}$, $E_{2,i} := \{f \in \mathbf{Lin}_{q,i}(d,n)\}$, $E_3 := \{e_1 \text{ spanned by the rows of } X^-\}$ and $E_{4,k} := \{X^- \text{ has rank } k\}$, we have by the law of total prob-

ability that for $i \in \{0, 1\}$,

$$\mathbb{P}(E_1|E_{2,i} \cap E_{4,k}) = \mathbb{P}(E_1|E_{2,i} \cap E_3 \cap E_{4,k})\mathbb{P}(E_3|E_{2,i} \cap E_{4,k}) \tag{24}$$

$$+ \mathbb{P}(E_1|E_{2,i} \cap E_3^c \cap E_{4,k})\mathbb{P}(E_3^c|E_{2,i} \cap E_{4,k}) \tag{25}$$

$$= \mathbb{P}(E_1|E_{2,i} \cap E_3 \cap E_{4,k})\mathbb{P}(E_3|E_{4,k}) + \mathbb{P}(E_1|E_{2,i} \cap E_3^c \cap E_{4,k})\mathbb{P}(E_3^c|E_{4,k}). \tag{26}$$

We can quantify the terms appearing above as follows:

$\mathbb{P}(E_1|E_{2,i} \cap E_3 \cap E_{4,k}) = 1 - q^{k-n-1}$ for $i \in \{0, 1\}$ since we know from Lemma 3 that $E_3 \cap E_{4,k}$ implies that there are $q^{n+1-k}$ consistent functions in $\mathbf{Lin}_{q,i}(d, n)$. As $f$ is selected uniformly at random from $\mathbf{Lin}_{q,i}(d, n)$, the statement follows.

$\mathbb{P}(E_1|E_{2,0} \cap E_3^c \cap E_{4,k}) = 1 - q^{k-n}$ because we know from Lemma 3 that $E_3^c \cap E_{4,k}$ implies that there are $q^{n-k}$ consistent functions in each $\mathbf{Lin}_{q,i}(d, n)$.

$\mathbb{P}(E_1|E_{2,1} \cap E_3^c \cap E_{4,k}) = 1$ since $\mathcal{A}$ will almost surely select a hypothesis from $\mathbf{Lin}_{q,0}(d, n)$ even though $f \in \mathbf{Lin}_{q,i}(d, n)$ if $e_1$ is not spanned.

$\mathbb{P}(E_3|E_{4,k}) = 1 - \mathbb{P}(E_3^c|E_{4,k}) = \frac{q^k-1}{q^{n+1}-1}$ follows from the fact that $X^-$ spans some $k$-dimensional row space uniformly at random and there are $q^k - 1$ possible non-zero linear combinations of $k$ linearly independent rows and $q^{n+1} - 1$ non-zero vectors of length $n + 1$.

Combining above facts we have that $\mathbf{Lin}_{q,0}(d, n) \cup \mathbf{Lin}_{q,1}(d, n)$ is not $(\alpha, \beta, n)$-learnable by $\mathcal{A}$ for $\alpha < \frac{q-1}{q}$ and

$$\beta < \mathbb{P}(E_1) = \sum_{i=0}^{1} \sum_{k=0}^{n} \mathbb{P}(E_1|E_{2,i} \cap E_{4,k})\mathbb{P}(E_{2,i} \cap E_{4,k}) \tag{27}$$

$$= \sum_{i=0}^{1} \mathbb{P}(E_{2,i}) \sum_{k=0}^{n} \left( \mathbb{P}(E_1|E_{2,i} \cap E_3 \cap E_{4,k}) \cdot \mathbb{P}(E_3|E_{4,k}) \right. \tag{28}$$

$$\left. + \mathbb{P}(E_1|E_{2,i} \cap E_3^c \cap E_{4,k}) \cdot \mathbb{P}(E_3^c|E_{4,k}) \right) \cdot \mathbb{P}(E_{4,k}) \tag{29}$$

$$= \frac{1}{2} \sum_{k=0}^{n} \left( (1 - q^{k-n-1}) \cdot \frac{q^k-1}{q^{n+1}-1} + (1 - q^{k-n}) \cdot (1 - \frac{q^k-1}{q^{n+1}-1}) \right. \tag{30}$$

$$\left. + (1 - q^{k-n-1}) \cdot \frac{q^k-1}{q^{n+1}-1} + (1 - \frac{q^k-1}{q^{n+1}-1}) \right) \times R_q(n, n+1, k) \tag{31}$$

$$= \frac{1}{2} \sum_{k=0}^{n} \left( (q^{k-n} - 2q^{k-n-1}) \frac{q^k-1}{q^{n+1}-1} + 2 - q^{k-n} \right) \times R_q(n, n+1, k). \tag{32}$$

$\square$

Using Lemmas 4 and 5, we are now ready to prove Theorem 4:

**Theorem 4.** *For every $\mathcal{A} \in \mathbb{A}_{lin}$ and every $n \leq d - 1$, $\mathcal{A}$ is not $((q-1)/2q, \eta, n)$-estimable with respect to $\mathbf{Lin}_q(d)$ and the $0 - 1$ loss where $\eta = \frac{1}{2} \sum_{k=0}^{n} \left[ (q^{k-n} - 2q^{k-n-1} - 1) \frac{q^k-1}{q^{n+1}-1} + 2 - q^{k-n} \right] \times R_q(n, n+1, k) - \sum_{k=0}^{n-1}(1 - q^{k-n}) \cdot R_q(n, n, k)$. In particular, for $q > 10$, it holds that $\eta > 0.4$, and generally, $\eta = \frac{1}{2} - \frac{1}{q} + o(1/q)$.*

*Proof.* For all product distributions appearing in this section assume that $\mathcal{D}_{\mathcal{X}} = \mathcal{U}(\mathbb{F}_q^d)$. Let $\mathbb{D} = \{\mathcal{D}_i\}_{i=1}^{q^{n+1}}$ be the set of $\mathbf{Lin}_q(d, n)$-realizable distributions such that $\mathbb{D}_0 = \{\mathcal{D}_i\}_{i=1}^{q^n}$ is realizable over $\mathbf{Lin}_{q,0}(d, n)$, and $\mathbb{D}_1 = \{\mathcal{D}_i\}_{i=q^n+1}^{q^{n+1}}$ is realizable over $\mathbf{Lin}_{q,1}(d, n)$. As we are working with countable measures,

$$d_{TV}(S_0, S_1) = \frac{1}{2}\| P - Q \|_1 = \frac{1}{2} \sum_{S} |P(S) - Q(S)| \tag{33}$$

where $P$ and $Q$ denote the probability measures associated with the distributions $\mathcal{D}_{I_0}^n$ and $\mathcal{D}_{I_1}^n$, respectively, where $I_0 \sim \mathcal{U}([q^n])$ and $I_1 \sim q^n + \mathcal{U}([q^n])$. First, we note that we need only sum over samples $S$ such that their corresponding input matrices span $e_1$ since Lemma 3 asserts that once $e_1$ is not spanned, the number of consistent functions in both classes is the same. Since the measures $P$ and $Q$ are uniform both w.r.t. to the inputs and the labeling functions, they both are proportional to the numbers of consistent functions (each multiplied by the same constant factor). Further, we know from Lemma 3, that once $e_1$ is spanned, the number of consistent functions in $\mathbf{Lin}_{q,i}(d,n)$ is non-zero iff $S \sim f \diamond \mathcal{D}_{\mathcal{X}}$ with $f \in \mathbf{Lin}_{q,i}(d,n)$. Hence only one of $P(S)$ and $Q(S)$ can be non-zero at a time. Therefore,

$$d_{TV}(S_0, S_1) = \frac{1}{2} \sum_{S:\{e_1 \text{ spanned}\}} |P(S) - Q(S)| \tag{34}$$

$$= \frac{1}{2} \left( \sum_{S:\{e_1 \text{ spanned}\}} P(S) + \sum_{S:\{e_1 \text{ spanned}\}} Q(S) \right) \tag{35}$$

$$= \mathbb{P}(\{e_1 \text{ spanned}\}). \tag{36}$$

According to the definitions in Theorem 3 we can hence pick

$$\gamma = 1 - \mathbb{P}(\{e_1 \text{ spanned}\}) \tag{37}$$

$$= 1 - \sum_{k=0}^{n} \frac{q^k - 1}{q^{n+1} - 1} \times R_q(n, n+1, k) \tag{38}$$

where we used of the fact that $\mathbb{P}(\{e_1 \text{ spanned}\}|\{\text{rank}(X^-) = k\}) = \frac{q^k - 1}{q^{n+1} - 1}$.

Plugging in above $\gamma$ and values of $\alpha, \beta, \delta$ in accordance to Lemmas 4 and 5 into Theorem 3, we finally obtain that $\mathbf{Lin}_{q,0}(d,n) \cup \mathbf{Lin}_{q,1}(d,n)$ is not $(\nu, \eta, n)$-estimable if simultaneously $\nu < (\alpha - \epsilon)/2 = (q-1)/2q$ and

$$\eta < \frac{\gamma}{2} - \frac{1 + \delta - \beta \frac{T}{T_1} + \delta \frac{T_0}{T_1}}{2} \tag{39}$$

$$= \frac{\gamma}{2} + \beta - \delta - \frac{1}{2} \tag{40}$$

$$= \frac{1}{2} \cdot \left[ 1 - \sum_{k=0}^{n} \frac{q^k - 1}{q^{n+1} - 1} \times R_q(n+1, n, k) \right] \tag{41}$$

$$+ \frac{1}{2} \sum_{k=0}^{n} \left[ (q^{k-n} - 2q^{k-n-1}) \frac{q^k - 1}{q^{n+1} - 1} + 2 - q^{k-n} \right] \times R_q(n, n+1, k) \tag{42}$$

$$- \sum_{k=0}^{n-1} (1 - q^{k-n}) \cdot R_q(n, n, k) - \frac{1}{2} \tag{43}$$

$$= \frac{1}{2} \sum_{k=0}^{n} \left[ (q^{k-n} - 2q^{k-n-1} - 1) \frac{q^k - 1}{q^{n+1} - 1} + 2 - q^{k-n} \right] \times R_q(n, n+1, k) \tag{44}$$

$$- \sum_{k=0}^{n-1} (1 - q^{k-n}) \cdot R_q(n, n, k) \tag{45}$$

where we used the fact that $T_0 = T_1 = T/2$.

To get the asymptotic result, we make use of the following two bounds:

$$1/q - o(1/q) \leq \frac{q^n - 1}{q^{n+1} - 1} \leq 1/q \tag{46}$$

$$R_q(n, n+1, n) \geq 1 - o(1/q). \tag{47}$$

To see that equation 47 holds, recall from Lemma 2 that $R_q(n, n+1, n) = (1 - q^{-n-1}) \cdot \ldots \cdot (1 - q^{-2})$. Now assume towards induction that the partial product $\pi_m := \prod_{j=2}^{m}(1 - q^{-j})$, $m \geq 2$ is in $1 - o(1/q)$. Then, $\pi_{m+1} \in (1 - o(1/q)) \cdot (1 - q^{-m+1})$ is also in $1 - o(1/q)$ since $q^{-m+1} \in o(1/q)$. We have for the base case that $\pi_2 = 1 - q^{-2}$ is in $1 - o(1/q)$, hence the claim follows.

Using above facts we can then lower bound equation 44 − equation 45 by lower bounding the first sum by only considering the term for which $k = n$, and upper bounding the sum in equation 45 by $1 - R_q(n, n, n)$ where $R_q(n, n, n) \geq 1 - 1/q - o(1/q)$ due to an induction argument similar as above.

Thereby we obtain that equation 44 − equation 45 is lower bounded by

$$\frac{1}{2}\left[(1 - 2/q - 1)(1/q - o(1/q)) + 2 - 1\right](1 - o(1/q)) - \left[1 - \left(1 - 1/q - o(1/q)\right)\right] \tag{48}$$

$$= \frac{1}{2} - \frac{1}{q} - o(1/q). \tag{49}$$

We thereby proved non-estimability of algorithms $\mathcal{A} \in \mathbb{A}_0$ with respect to $\mathcal{H} = \mathbf{Lin}(d)$ and distributions families $\mathbb{D}_0, \mathbb{D}_1$ (defined at the beginning of the proof).

Finally, we extend this result to the class of all linearly biased ERMs in $\mathbb{A}_{lin}$ via the following reduction:

**Remark** (Reduction). *Without loss of generality, one can focus on the class $\mathbb{A}_0$ whenever proving learnability and estimability results about $\mathbb{A}_{lin}$ (see Definition 5). This follows from a reduction of non-estimability with algorithms $\mathcal{A}' \in \mathbb{A}_{lin}$ to non-estimability with algorithms $\mathcal{A} \in \mathbb{A}_0$ as informally discussed below.*

*First note that any linearly constrained ERM $\mathcal{A} \in \mathbb{A}_{lin}$ is implicitly parametrized by some $\sigma \in \mathbb{F}_q^d \setminus \{0\}$ and $\kappa \in \mathbb{F}_q$ such that $\mathcal{A}$ is biased towards selecting hypotheses $f_b \in \mathbf{Lin}_q(d)$ such that $\sum_{i=1}^d \sigma_i \cdot b_i = \kappa$. For example, in the case of $\mathcal{A} \in \mathbb{A}_0$, we have $\sigma = e_1$ and $\kappa = 0$.*

*Let us briefly recap the setup in Theorem 4.*

- *First, we introduced the $d$ dimensional function space $\mathbf{Lin}_q(d)$.*

- *Based on $n$, we linearly mapped the space onto the $n + 1$ dimensional subspace $\mathbf{Lin}_q(d, n)$.*

- *We assumed that $\mathcal{A}$ is biased to the $n$ dimensional subspace of functions $f_a$ with $a_1 = 0$.*

- *Finally, we showed learnability (with $\mathcal{A}$) over the subspace $\mathcal{H}_0 = \mathbf{Lin}_{q,0}(d, n)$ and non-learnability over the subspace $\mathcal{H}_0 \cup \mathcal{H}_1$, where $\mathcal{H}_1 = \mathbf{Lin}_{q,1}(d, n)$.*

*We now sketch how for arbitrary $\mathcal{A}' \in \mathbb{A}_{lin}$ with bias coefficients $(\sigma, \kappa)$ one can find appropriate subspaces of $\mathbf{Lin}_q(d)$ that admit essentially the same properties as the ones studied in the proof of Theorem 4 for $\mathcal{A} \in \mathbb{A}_0$.*

*For $W \subset \mathbb{F}_q^d$ some linear subspace define by $\mathbf{Lin}_q(W)$ the space of linear functions over $\mathbb{F}_q^d$ with coefficients from $W$. Now consider the following setup: given $\sigma, \kappa$ as defined above, project $\mathbb{F}_q^d$ onto an $n + 1$ dimensional subspace $V$ such that $\mathbf{Lin}_q(V) \subset \mathbf{Lin}_q(d)$ via an appropriate linear map $\Pi$ such that $\sigma$ is not in its kernel. Now define $\sigma' = \Pi\sigma \neq 0$ and let $\mathcal{H}'_0$ be the $n$ dimensional subspace of $V$ consisting of functions $f_b$ such that $\sum_{i=1}^{n+1} \sigma'_i \cdot b_i = \kappa$. Similarly, define $\mathcal{H}'_1$ to be the set consisting of functions $f_b$ such that $\sum_{i=1}^{n+1} \sigma'_i \cdot b_i = \kappa + 1$.*

*The reduction now boils down to the fact that showing learnability of $\mathcal{H}'_0$ and non-learnability of $\mathcal{H}'_0 \cup \mathcal{H}'_1$ (together with distribution families $\mathbb{D}'_0, \mathbb{D}'_1$ consisting of distributions with uniform marginals over $\mathbb{F}_q^d$ and labelings from $\mathcal{H}'_0, \mathcal{H}'_1$, respectively) with $\mathcal{A}'$ can be shown analogously to the setup $(\mathcal{A}, \mathcal{H}_0, \mathcal{H}_1, \mathbb{D}_0, \mathbb{D}_1)$ from Theorem 4 via simple some adaptions of the steps involved in proving it. In some more (but not full) detail:*

- *Since the linear spaces $\mathbf{Lin}_q(d, n)$ and $\mathbf{Lin}_q(V)$ (and their above mentioned subspaces) have the same cardinalities, it follows that all proofs based solely on counting functions subject to a fixed number of linear constraints carry over one-to-one.*

- *Combining linear coefficients of two linear functions $f_a, f_b \in \mathbf{Lin}_q(d)$ trivially yields another linear function $f_{a+b} \in \mathbf{Lin}_q(d)$. Hence Lemma 1 applies.*

- *Any linear combination (modulo q) of i.i.d. random variables uniform over $\mathbb{F}_q$ is again uniform. This fact together with our assumption of i.i.d. uniform inputs means that Lemma 2 carries over.*

- *For deriving the numbers of consistent functions in Lemma 3, we first 'preprocess' the inputs via the mapping $\Pi X$ with $\Pi$ as defined above. Recall that in the original setup this mapping simply amounted to truncating the input vectors.*

- *One can also easily adapt Lemma 5 since spanning $e_1$ has the same probability as spanning any arbitrary fixed $\sigma'$. The TV distance appearing in the final steps of showing Theorem 4 is the same in both setups for the same reason.*

$\square$

## I   LIMITATIONS ON ESTIMABILITY IN BINARY CLASSIFICATION

Let us restate Theorem 5 for convenience:

**Theorem 5.** *For every $\mathcal{A} \in \mathbb{A}_{lin}$ (possibly randomized) and every $n \le d-1$, $\mathcal{A}$ is not $(0.25, 0.14, n)$-estimable with respect to $\mathbf{Lin}_2(d)$ and the $0-1$ loss. More so, for every deterministic $\mathcal{A} \in \mathbb{A}_{lin}$ and every $6 \le n \le d$, $\mathcal{A}$ is not $(0.25, 0.32, n)$-estimable with respect to $\mathbf{Lin}_2(d)$ and the $0-1$ loss.*

### I.1   PROOF OF THEOREM 5 FOR $\mathcal{A}$ DETERMINISTIC

*Proof.* We consider the case $q = 2$ and examine when estimability is not possible for any algorithm $\mathcal{A} \in \mathbb{A}_0$ by bounding the accuracy of the optimal estimator $\mathcal{E}^*$. The result once again carries over to the more general case $\mathcal{A} \in \mathbb{A}_{lin}$ due the reduction argument described in the remark appearing at the end of the previous section.

Due to an argument analogous to the one in Lemma 3, we have that for $n \le d$ and $\text{rank}(X) = k$, there are $2^{d-k}$ consistent functions in $\mathbf{Lin}_2(d)$. By definition, in order to have $(\nu, \eta, n)$-estimability, $\mathcal{E}^*$ can have failure probability at most $\eta(\nu, n)$ for given $\nu, n$.

Since for any given $S$, $f$ is uniform over all consistent functions in $\mathbf{Lin}(d)$, the optimal estimator $\mathcal{E}^*$ assigns 0 whenever $k = n$, since in this case any $\mathcal{A} \in \mathbb{A}_0$ outputs the ground truth almost surely.

Moreover, any estimator $\mathcal{E}^*$ that is optimal for a fixed error level $\nu$ must necessarily assign a value $c_\nu \in (0.5 - \nu, 0.5]$ whenever $k < n$, since then there are multiple consistent functions in $\mathbf{Lin}_2(d)$, and simultaneously $\mathbb{P}(\{\mathcal{A}(\mathcal{S}) = h\}) \le \frac{1}{2}$ for ground truth hypothesis $h$. To expand on this, note that if $\mathcal{E}^*$ were to assign any value in $[0, 0.5 - \nu]$ while there exist $m \ge 2$ functions consistent with $S$, this would cause the fidelity terms $F_i := |\mathcal{E}^*(\mathcal{A}, S) - L_{\mathcal{D}_i}(\mathcal{A}(S))|$ to exceed the threshold $\nu$ in $m-1$ of the instances and only in a single instance fall below $\nu$. This would obviously yield an increased overall error probability and hence be suboptimal. We can therefore assume w.l.o.g. that $\mathcal{E}^*$ has $c_\nu = 0.5$.

Given that $\nu = 0.25$ and $n < d$, this estimator then fails (i.e. exceeds the error level $\nu$) over $\{f \diamond \mathcal{U}(\mathcal{X}) | f \in \mathbf{Lin}_{2,0}(d, n) \cup \mathbf{Lin}_{2,1}(d, n)\}$ with probability

$$\mathbb{P}(\{\mathcal{E}^*(\mathcal{A}, S) \ne L_{\mathcal{D}}(\mathcal{A}(S))\}) = \frac{1}{2} \sum_{k=0}^{n} \left( 2^{k-n-1} \cdot \frac{2^k - 1}{2^{n+1} - 1} + 2^{k-n} \cdot \left(1 - \frac{2^k - 1}{2^{n+1} - 1}\right) \right) \tag{50}$$

$$+ 2^{k-n-1} \cdot \frac{2^k - 1}{2^{n+1} - 1} \right) \times R_2(n, n+1, k) \tag{51}$$

$$= \sum_{k=0}^{n} 2^{k-n-1} \cdot R_2(n, n+1, k) \tag{52}$$

where the derivation is analogous to the one of $\beta$ in Lemma 5.

On the other hand, for $n = d$, from a similar argument it follows that over $\{f \diamond \mathcal{U}(\mathcal{X}) | f \in \mathbf{Lin}_2(d)\}$, given that $\text{rank}(X) = k$, $\mathcal{A}$ picks the ground truth with probability $2^{k-d}$ and hence $\mathcal{E}^*$ fails with

probability

$$\mathbb{P}(\{\mathcal{E}^*(\mathcal{A}, S) \neq L_{\mathcal{D}}(\mathcal{A}(S))\}) = \sum_{k=0}^{d-1} 2^{k-d} \cdot R_2(n, n, k). \tag{53}$$

Combining equation 52 with equation 53 we obtain that for all $n \leq d$, over $\mathbf{Lin}_2(d)$, the optimal estimator $\mathcal{E}^*$ fails with probability

$$\mathbb{P}(\{\mathcal{E}^*(\mathcal{A}, S) \neq L_{\mathcal{D}}(\mathcal{A}(S))\}) = \sum_{k=0}^{m} 2^{k-m-1} \cdot R_2(n, m+1, k). \tag{54}$$

where $m \coloneqq \min\{n, d-1\}$. It can be easily verified that for fixed $d$, equation 54 is monotonically decreasing in $n$. Moreover, when fixing $n = d$, equation 54 is increasing in $d$ and exceeds $0.32$ for all $d \geq 6$. $\qquad\square$

### I.2 PROOF OF THEOREM 5 FOR $\mathcal{A}$ RANDOM

*Proof.* Assume w.l.o.g. that $\mathcal{A}$ randomly outputs a consistent function in $\mathbf{Lin}_{2,0}$ whenever possible. Define $E_- = \{\mathrm{rank}(X^-) = n\} \cap \{e_1 \text{ not spanned}\}$. Over the class $\mathbf{Lin}_2(d, n)$, it follows from equation 46 together with the fact that $R_2(n, n+1, n) > 0.57$ for all $n \geq 1$ that

$$\mathbb{P}(E_-) > 0.57 \cdot 0.5 \tag{55}$$

where $X^-$ denotes the reduced $n \times (n+1)$ input matrix (recall that over $\mathbf{Lin}_2(d, n)$ it is sufficient to process the first $n + 1$ coordinates of the inputs). But in this case, there exists exactly one consistent function in each $\mathbf{Lin}_{2,0}(d, n)$ and $\mathbf{Lin}_{2,1}(d, n)$. Due to a argument similar to the one presented in the preceding subsection, we know that upon observing $E_-$, an optimal estimator $\mathcal{E}^*$ of the population risk assigns $0$ since this yields the optimal accuracy. But then, $\mathcal{E}^*$ exceeds the error level $\nu = 0.25$ under the event $\{h \in \mathbf{Lin}_{2,1}\}$ for $h$ denoting the ground truth labeling function. Since this event has probability $0.5$, we can conclude that $\mathcal{E}^*$ fails with probability at least $0.57 \cdot 0.5 \cdot 0.5 > 0.14$. Since this implies that $\mathbf{Lin}_{2,0}(d, n) \cup \mathbf{Lin}_{2,1}(d, n)$ is not $(0.25, 0.14, n)$-estimable, if follows that the superset $\mathbf{Lin}_2(d)$ is also not estimable with the same parameters. $\qquad\square$

