# OpenReview forum: "Fantastic Generalization Measures are Nowhere to be Found"
_ICLR.cc/2024/Conference — ICLR 2024 poster_

### Official Review · Reviewer_Hb6n · 2023-10-24

**Soundness:** 3 good
**Presentation:** 2 fair
**Contribution:** 3 good
**Rating:** 8
**Confidence:** 3

**Summary:**

This paper presents several novel results regarding the non-existence of tight generalization bound in the over-parameterized setting. In particular, this paper proves that a tight generalization bound cannot exist without making assumption about the distribution of training data. In the case where no assumption is made about the learning algorithm, the authors show a much stronger version "no free lunch" that any generalization bound is not tight on a large number of (Algorithm, Distribution) pairs. And in the case where we are concerned with a specific algorithm family, then the authors show that there is a inherent trade-off between the tightness of generalization bound and the quality of the learned model.

**Strengths:**

I find the results of this paper be very impactful. Understanding generalization properties in the over-parameterized regime is of great interest in recent literature and this paper basically shows that such analysis is impossible (both theoretically and practically) without assuming specific properties about the data distribution. This in turn shows that great number of existing bounds in the literature are not useful and any future work should direct to data-dependent bounds.

The authors did a good job by putting their results into different cases that are intuitive on a high level. And I find Table 1 to be very clean.

Section 2 is well-written and gets the main points across without being entangled with the details.

The proofs are generally clean and easy to understand

**Weaknesses:**

I find the formal results stated in Sections 5 and 6 to be extremely difficult to follow. While the informal statements in Section 2 are understandably vague, Sections 5 and 6 failed to clarify my confusions from Section 2. I think this is due to two issues:

1. Section 4 did a poor job at explaining the formal notation. I feel that Definitions 1 and 2 are not particularly well-motivated (more on this later). And shoving everything else into the Appendix does not help either.

2. Certain points in the intro were not explained properly in later sections. a) The term "vacuous" was never formally defined, b) the connection between tightness of generalization bound (eq 1) and the notion of estimability is also not discussed in depth (why are they equivalent? I know the argument is not hard, but this is provides important contexts for the main theorems).

3. The theorem statements are pretty mouthful themselves and except for Theorem 2, the authors did not offer discussions that helps with parsing the theorem statements.

Next, some *major gaps* I found in the paper:

4. The definition of over-parameterizaton (Definition 2) in this paper is not standard and my impression is that they are phrased to make the proofs simpler. While Definition 2 does intuitively fit the idea of over-parameterization, I feel strongly that the author should add: a) detailed discussions on why this definition is consistent with the standard setting in the literature, b) examples.

5. Similar to the previous point, in Theorem 3, the condition on TV distance is unmotivated and seems to only exist to make the problem easier.

6. In the proof of Theorem 1, the authors did not show the existence of Bayes-like Random ERM.

A few minor comments regarding the proofs (I did not check Appendix G or H):

Page 18: the theorem statement is about Theorem 2, *not* 1.

Page 19: the final sentence should start with "the second equality holds"

Page 20: the result "Theorem 1 in Angel & Spinka (2021)" is just a standard fact on the existence of coupling, so it is better to just say that directly.

Page 20: the big equation block looks atrocious, please left align the lines and use indentation to make the + sign on the second line more visible.

Page 20: Please define what event $B$ is before that big equation block. Also in the definition of $B$, the final inequality $L_{D_{I_1}}(A(S_1)) \ge \alpha$ is missing its RHS.

**Questions:**

In addition to all of my concerns above, how is over-parameterization used in the argument between Definition 3 and Theorem 1?

Overall, I feel that some restructuring is necessary for Sections 4-6. The claims in this paper are very interesting and impactful, but I find them to be unnecessarily difficult to understand.

---

> ### Author Response · Authors · 2023-11-13
>
> After carefully reading your review, we would like to confirm (or clarify) our understanding before providing our response in full detail.
>
> Is there an agreed-upon definition of overparametrization in the literature? So we can refer to it in our answers. If not, how would you mathematically define overparametrization? How would you define such a concept over non-parametrized spaces? Please note that Section 4 gives the motivation for our definition.
>
>
>
>
> > In addition to all of my concerns above, how is over-parameterization used in the argument between Definition 3 and Theorem 1?
>
> We are not sure we understood your question. Are you asking for a high-level idea about how overparametrization is being used in the proof of Theorem 1? If not, can you please clarify/rephrase your question?
>
>
> Regarding (6) and Definition 7 (Bayes-like Random ERM), we missed mentioning in our paper that all spaces we consider are finite. With this in mind, the definition of Bayes-like Random ERM  is a well-defined random variable over a finite sample space. Does this answer your query?

---

> > ### Comment · Reviewer_Hb6n · 2023-11-13
> >
> > Thank you for your prompt response. I am not aware of any formal definition of over-parameterization that consistently shows up in the literature. I will discuss about this in more details in the top-level thread. But, Definition 2 in your paper seems very different from my basic intuition of this concept. While I feel that there is nothing inherently wrong about your interpretation, I would love to a much more substantial discussion than Section 4.
> >
> > > We are not sure we understood your question. Are you asking for a high-level idea about how overparametrization is being used in the proof of Theorem 1? If not, can you please clarify/rephrase your question?
> >
> > I am talking about the paragraph between the statements of Definition 3 and Theorem 1. This is in the middle of page 6.
> >
> > > We are not sure we understood your question. Are you asking for a high-level idea about how overparametrization is being used in the proof of Theorem 1? If not, can you please clarify/rephrase your question?
> >
> > Sounds good. Can you add this comment immediately following Definition 7?

---

> ### Author Response · Authors · 2023-11-15
>
> > I am talking about the paragraph between the statements of Definition 3 and Theorem 1. This is in the middle of page 6.
>
> Thank you for the pointer. After careful re-reading, we polished the paragraph further and added more details, which now hopefully makes it easier to parse:
>
> Note that given an algorithm-independent and distribution-independent bound $C$, it is not hard to construct a \emph{single} (algorithm, distribution) pair that makes it vacuous. To illustrate this, consider the ERM defined as $\mathcal A_C(S):=\arg\min_{h\in \mathcal H: \: L_S(h)=0} C(h,S)$. Assume that $C$ is a valid generalization measure, i.e., (1) holds almost surely for every $\mathcal H$ realizable distribution $\mathcal D$ with labeling function $h_\mathcal D$. Then, by construction, $L_{\mathcal D(\mathcal A_C(S))} < L_S(\mathcal A_C(S)) + C(\mathcal A_C(S),S)\leq C(h_\mathcal D,S)$ almost surely over $S\sim \mathcal D^n$.
>
> Now assume that the setting is overparametrized (say with large $\alpha$ and $\beta$ for a fixed $n$), i.e., no algorithm can learn (with error $\alpha$) the ground truth labeling w.p. at least $1- \beta$ jointly over the uniform choice of the distributions, and $n$ samples from the chosen distribution. This implies that for every algorithm $\mathcal A$, there exists at least one distribution $\mathcal D’$ such that $\mathcal A$ cannot learn w.p. at least $\beta$ over $S \sim \mathcal D’^n$. Hence there exists a realizable distribution $\mathcal D'$ with deterministic labeling function $h_{\mathcal D'}$ such that w.h.p. $L_{\mathcal D'}(\mathcal A_C(S))> \alpha$ which implies by the above inequality $C(h_{\mathcal D'},S)>\alpha$ w.h.p. But then it holds w.h.p. that $C(h_{\mathcal D'},S)\gg L_{\mathcal D'}(h_{D'}) - L_S(h_{D'})= 0$ and hence the bound is w.h.p. not $\alpha$-tight (in the sense of Eq. 2) for the pair $(\mathcal A_{h_{\mathcal D'}},\mathcal D')$, where $\mathcal A_{h_{\mathcal D'}}$ is the constant algorithm that always outputs $h_{\mathcal D'}$.
>
> In particular, we removed the use of the no-free-lunch theorem and made the connection to the overparametrized setting more direct. Does this clarify things? If not, which step causes confusion?
>
> > Can you add this comment immediately following Definition 7?
>
> Yes, will add this comment. Thank you for helping us improve the paper!

---

> > ### Author Response · Authors · 2023-11-16
> >
> > > Section 4 did a poor job at explaining the formal notation. I feel that Definitions 1 and 2 are not particularly well-motivated (more on this later).
> >
> > We hope that we addressed sufficiently with our global response concerning overparametrization. Please let us know if this is not the case.
> >
> > > Certain points in the intro were not explained properly in later sections. a) The term "vacuous" was never formally defined, b) the connection between tightness of generalization bound (eq 1) and the notion of estimability is also not discussed in depth (why are they equivalent? I know the argument is not hard, but this is provides important contexts for the main theorems).
> >
> > a) A generalization bound for binary classification with 0-1 loss is called vacuous if it exceeds the numerical value 1, which is the trivial upper bound of the population risk. We will add this in a footnote on the first page.
> >
> > b) Let’s focus on algorithm-independent estimability. The existence of a tight generalization bound implies estimability, directly by Def. 3. In the other direction, estimability (with parameters $\epsilon,\delta$) implies the existence of a $2 \epsilon$- tight generalization bound, with the same probability parameter $\delta$. This is because estimability implies that there exists some estimator $E$, which might for some samples underestimate the risk by $\epsilon$, and for other samples overestimate the risk by $\epsilon$. But then the generalization bound $E + \epsilon$ upper bounds the risk with error at most $2 \epsilon$ with probability at least $1-\delta$. The exact same argument applies also for algorithm-dependent estimability.
> >
> > > The theorem statements are pretty mouthful themselves and except for Theorem 2, the authors did not offer discussions that helps with parsing the theorem statements.
> >
> > The statement Theorem 1 makes essentially the same qualitative statement w.r.t. non-estimability as the less general(\*) Theorem 2, but it applies to all overparametrized settings rather than classes of finite VC dimension.
> > Since these Theorems are so similar (Thm. 2 is essentially a Corollary of Thm. 1), we only discuss Thm. 2 in detail. We decided to focus on Theorem 2 since it is stated in terms of VC-dimension, which is a well-known quality for readers familiar with learning theory, whereas our notion of overparametrization is non-standard.
> >
> > (\*) As we argued in the global discussion concerning overparametrization, finite VC dimension implies overparametrized.
> >
> > Let us also add some more details to the discussion of Theorem 3:
> >
> > $H_0$ is to be understood as the class that capture's the learner's "bias". For example, we have an algorithm that performs well on $H_0$, say with accuracy $\epsilon=0.01$ and confidence $\delta=0.01$.
> > The intuition behind $H_1$ is the following: first note that $H_1$ cannot be arbitrary: If $H_1$ is very small, for example, a set of three functions, $H=H_0 \cup H_1$ would be learnable with comparable parameters $\alpha \approx 0.01$ and $\beta \approx 0.01$. That would render item 2 in the theorem to be useless since $\alpha-\epsilon \approx 0$ would be very small. If $H_0$ is very small, for example, a set of three functions, and $H_1$ would consist of all other functions not in $H_0$, we could easily distinguish between samples from $H_0$ and samples from $H_1$ so the total variation is close to 1 and $\gamma \approx 0$. That would render item 2 in the theorem to be ineffective since $\eta \approx 0$ would be very small.
> > Finally, an intuitive way to understand Theorem 3 is a scenario where it is possible for an algorithm to learn $H_0$ or $H_1$ separately, but not both (which is consistent with the above statements). Theorem 3 shows that any algorithm that performs well in such a case (learns well one of $H_0$ or $H_1$) would not be estimable over the union $H=H_0 \cup H_1$.
> >
> > Does this help with parsing Theorem 3?
> >
> > Regarding Thms. 4 and 5:
> >
> > Since Theorems 4 and 5 are formulated directly in terms of estimability, we believe that they can be parsed on a technical level without problems once Def. 4 is understood, and we address their implications in subsection 6.1. We are of course happy to answer further questions regarding them (or any of the other Theorems) – perhaps we missed a detail that makes them hard to parse for first-time readers. If so, please let us know what this detail might be.
> >
> > Nevertheless, a few more remarks reg. Thm. 5:
> >
> > Since Theorem 4 provides only weak numerical values when applied to parities (a particularly interesting/easy to grasp subset of linear functions), we show non-estimability of (parities functions, linearly biased ERMs) directly, which leads to significant numerical values. This implies that no tight bound (exceeding these numbers $(\epsilon, \delta, n)$) can exist in this setting.
> > We will incorporate a version of the above explanations in the forthcoming updated manuscript, which should improve the overall presentation.

---

> ### Author Response · Authors · 2023-11-16
>
> > The definition of over-parameterizaton (Definition 2) in this paper is not standard and my impression is that they are phrased to make the proofs simpler. While Definition 2 does intuitively fit the idea of over-parameterization, I feel strongly that the author should add: a) detailed discussions on why this definition is consistent with the standard setting in the literature, b) examples.
>
> Please see the global discussion regarding overparametrization.
>
> > Similar to the previous point, in Theorem 3, the condition on TV distance is unmotivated and seems to only exist to make the problem easier.
>
> Theorem 3 gives a sufficient condition for non-estimability. Our proof essentially boils down to the fact that samples drawn from two different distributions – with labelings that disagree over a significant fraction of the space – “look the same” or “are easily confusable” for samples of size $n$, with reasonably large probability. If then the algorithm can learn over one of the distributions, but fails to learn over the other one (both of these items are part of the assumptions of the Theorem), any estimator must be loose.
> In the above context, the TV distance between these two distributions is a natural measure for the “confusability” of (the samples from) these two distributions.
>
> As mentioned, this technique only yields a sufficient condition, so in theory it might be that it cannot yield meaningful numerical values of non-estimability. This is why we show in Theorem 4 that this is not the case (showcased for linear functions, together with ERMs with linear bias): we get essentially the strongest possible result already for $q=11$: that is, no estimator can have confidence much higher than a random guess, if it is supposed to do better than the trivial error level (see the discussion under Thm. 4).

---

> ### Comment · Reviewer_Hb6n · 2023-11-17
>
> > Thank you for the pointer. After careful re-reading, we polished the paragraph further and added more details, which now hopefully makes it easier to parse:
>
> Thank you for the revised argument. While I feel the writing has improved, I am still somewhat confused. Namely, what is this "labeling function" and what is its significance? Also, shouldn't $\alpha$ and $\beta$ be small?
>
> And regarding rest of your response. They are very thorough, but I did not ask for those clarifications. I believe that I understood your arguments reasonably well, and your response does not seem to change that impression (up to whatever I have forgotten between now and when I wrote the review). However, I expended considerable efforts into understanding your paper. Therefore, I am asking you to revise the paper in way that is easier to read for future readers. You have not done that thus far.
>
> I would like you remind you that you can submit a revised paper. I am afraid that I will have to *lower* my score if I am still not seeing a revision incorporating suggestions from me and other reviewers. I will write what I personally what to see as a reply to one of your top-level response.

---

> ### Author Response · Authors · 2023-11-18
>
> > what is this "labeling function" and what is its significance?
>
> The data distributions $\mathcal D$ are characterized by their joint distribution over inputs $x \in \mathcal X$ and labels $y \in \mathcal Y$. We assume that the marginal distribution of the labels $y$ is $h \circ \mathcal D_{\mathcal X}$, i.e., the marginal input distribution composed with a deterministic labeling function $h: \mathcal X \rightarrow \mathcal Y$.
>
> The labeling fcts. play a crucial role for how strongly overparametrized a setting is, since the ‘richness’ of the labeling functions $\\{h_i\\}_i\subset \mathcal H$ corresponding to the distribution family $\mathbb D = \\{\mathcal D_i \\}_i$ plays an important role in how hard it is to learn over $(\mathcal H,\mathbb D)$.
>
> > shouldn't $\alpha$ and $\beta$ be small?
>
> In our example we assume for simplicity that they are both large. This corresponds to a ‘strongly overparametrized’ setting, i.e., the impossibility to learn even with loose accuracy and low confidence.
> With this assumption, we get in the above toy example a strong non-estimability result. But the above example works for any $\alpha,\beta$. Had we assumed $\alpha,\beta$ to be small, we would only get a weak non-estimability result (as one should expect).

---

> > ### Author Response · Authors · 2023-11-21
> >
> > Dear Reviewer Hb6n,
> >
> > Kindly take a look at _**General Answer 3: New Revision of The Paper**_ above, and at the PDF version of the paper.
> >
> > Best,
> > The authors

---

> ### Comment · Reviewer_Hb6n · 2023-11-23
>
> I would like to thank the authors for their detailed new revision. I believe that my concerns regarding the definition of over-parameterization is largely resolved. So I will bump the score up to **8**.
>
> However, I did not closely track your discussion with Reviewer bL1w. While it seems that all reviewers including me think that this paper is interesting, Reviewer bL1w did raise some good points on how the results could be applied to the existing generalization bounds in the literature. Because I did not fully digest those discussions to tell whether the new Appendix C offers a comprehensive answer, I will also lower my confidence to to **3**.

---

### Official Review · Reviewer_bL1w · 2023-10-28

**Soundness:** 4 excellent
**Presentation:** 3 good
**Contribution:** 4 excellent
**Rating:** 6
**Confidence:** 4

**Summary:**

This paper provides various quantitative results that demonstrate that in an "overparameterized setting", generalization bounds cannot be tight unless they depend on the structure of the sampling distribution: if a generalization bound applies indiscriminately to all possible distributions, it cannot be tight for the result of a successful algorithm in the overparametrized setting (which is defined as a setting where there is not enough data for there to exist an algorithm which will perform ($\alpha$-) well with high probability (1-$\beta$) over a random draw of a realizable distribution. In a sense, the results can be interpreted as a quantitative version of the contrapositive of the fundamental lemma of statistical learning theory (which says that if a function class is estimable, it is learnable).
Several results (including theorems 2 and 4) study specific toy settings where the overparmetrization can be more concretely described (e.g. the number of samples is less than VC dimension or the actual dimension of the space) and the statement holds thanks to an assumption that the marginal distribution over the input is uniform. These results are used to argue that the only type of generalization bounds that can correctly explain the success of Machine Learning models such as neural networks in the overparametrized setting must exhibit dependence on the sampling distribution.



More precisely

Theorem 1 states that if we consider a function class associated to an classification problem with the 0-1 loss and a finite set of realizable distributions, as long as the problem is overparametrized (i.e., not reliably learnable over most distributions), then there exists an ERM algorithm whose error cannot be reliably estimated either.

Theorem 2 shows that if the number of samples is less than the O(VC dimension) and the sampling distribution over the inputs is a uniform distribution over a finite set (which is shattered by the hypothesis set), then the class is not estimable.

Theorem 3 shows that if a function class and a set of candidate ground truth distributions can be split into two parts, each realizable with respect to each other, and if the total variation distance between the resulting mixture distributions over the samples is not close to 1, then no estimator can work uniformly well for all algorithms with high probability. The proof heavily relies on the assumption that the total variation is not too high and a coupling argument from [2].

Theorem 4 concerns the learning problem of learning linear maps over $F^d$ where $F$ is the finite field of size $q$. Here, similarly to Theorem 2, it is assumed that the inputs are drawn from a uniform distribution, which allows one to define the overparametrization more concretely ($n<<d$). One considers the class of algorithms which "are biased towards a given set of subspaces" (i.e. for all $n$, there exists a subspace of dimension $n$ such that the algorithm always outputs a consistent hypothesis from that space if there exists one).  This class is shown not to be estimable. The proof relies on several elegant combinatorial arguments together with existing results form [1] concerning the proportion of matrices over a finite field which have a given rank.

Theorem 5 specifically improves the rate (in terms of the failure probabilities) for the case where $q=2$.


References:

[1] IAN F. BLAKE AND CHRIS STUDHOLME, PROPERTIES OF RANDOM MATRICES AND APPLICATIONS, 2006

[2] Omer Angel and Yinon Spinka. Pairwise optimal coupling of multiple random variables, 2021. (Arxiv)

**Strengths:**

1. Understanding generalization bounds at a fundamental level is an extremely important and popular topic, and this paper makes a nontrivial contribution in this direction.

2. All the formal results and proofs are sound and reasonably well written.

3. Some of the **proofs are nontrivial**, especially the **proof of Theorem 4**, which is quite impressive and relies on an ingenious splitting into many different situations depending on the rank of the design matrix and whether or not the first unit vector is in the span of its rows. **This is a solid maths paper.**

4. The main claim of the paper, if appropriately toned down, is an interesting variation on known observations that may hitherto never been individually explained in such detail. This makes the paper highly worthy of publication.

5. It is very nice that the relationship with existing works such as [5] is explained.

**Weaknesses:**

The paper makes sensational claims that lack nuance: in the main paper and in extensive discussions in the appendix, the paper uses the argument derived from Theorems 1 to 3 to explain that most existing generalization analyses are invalid or at least "lacking" since they "do not take the distribution into account".

As far as I am concerned, there are several serious problems with this claim which can be summarized as follows:

1. The general argument is partially valid, but not as strongly as the authors claim:  it applies to any "overparametrized bound that doesn't depend on the structure of the sampling distribution". At an abstract level, this description very loosely "can be argued or conjectured to apply to a large part of the existing literature. However, it doesn't strictly or objectively apply to almost any of the existing literature.

2. The part of the paper's general argument which is valid is not completely novel. Although all the theoretical results appear novel and are indeed interesting, once they are compressed into the authors' high-level statement that "a bound cannot be tight if it works uniformly over all distributions and the setting is overparametrized", they no longer consist in a novel idea. Theorem 1 can be summarised as saying "not learnable implies not estimable", which is the contrapositive of the fundamental lemma of SLT which says that "estimable function classes are learnable via empirical risk minimization". The novel component of the results consists in turning this result into a quantitative version where a distribution over the set of realizable distributions is considered.

3. There are plenty of papers that actively take the sampling distribution into account in their analysis and are not even cited here. The most obvious examples would be [3] and the NTK literature [4].




More detailed explanation of point 1 above:

All theorems presented rely on the assumption that the problem is "in the overparametrized setting", which is defined in a very strict way that goes very far beyond saying that the number of parameters is greater than the number of samples and makes the abstract message of the theorems (theorems 1 and 3) approach a tautology. By definition, the authors assume that there is not enough data to find an algorithm that will reliably estimate most sampling distributions considered. This is only a short step from directly assuming that "any generalization bound which only relies on the sampling distribution is vacuous". All the concrete examples given in Theorems 2 and 4 make use of unrealistic assumptions about the uniformity of the sampling distribution.

I understand that this does not fully invalidate the authors' point because it does show that no bound that relies on a "fixed" hypothesis class definition can be uniformly tight for all distributions: the authors' argument that "for a bound to be nonvacuous in the overparametrized regime, it must be dependent on the sampling distribution" is correct, but I feel like the authors catastrophically overstate the extent to which this directly invalidates any of the existing literature.

Which bounds in the existing literature are really "in the overparametrized setting"? Which of them really "do not depend on the sampling distribution"?


The only example of existing bounds that I can think of where it is genuinely known for certain that one is in the overparametrized setting would be parameter counting bounds (e.g. [6]). As far as I understand, even early norm-based uniform convergence bounds such as those of [7,12] cannot technically be proved to be "in the overparametrized setting" except by directly showing that they are vacuous, which makes the argument circular.  Likewise, although the idea that existing bounds such as margin bounds (e.g. [7]) "do not take the distribution into account" makes some sense, it is not technically true since the margin depends on the data: the idea that generalization bounds for neural networks in the setting where there are more parameters than samples must rely on the sampling distribution can arguably be placed as early as 2017 with the advent of norm-based bounds (this is reflected in the text of the introduction of [7]).  Data dependency is also given an increasingly important role in many more modern results such as those of [8,9], which replace the product of spectral norms in the bound by an empirical analog, and later in [10,11], which rely on empirical estimates of the norms of the activations in their analysis. Understandably, none of these bounds are traditionally considered to be in the category of "data dependent" bounds (because the data-dependency isn't "strong" enough), but none of the theorems of the present paper actually apply to any of them. This means the argument in the paper, although valid at an abstract/intuitive level, should not be presented as mathematically invalidating almost any existing works. Beyond the above works, **the results in [3,4] (including the NTK literature)** certainly take data dependency to the next level and arguably **provide a genuinely satisfying answer to the issue touched on in the present work**. They should be discussed. Note also that basically any bound that relies on the rank of the weight matrices or a similar quantity such as the sparsity of the trained connections would be non trivially using the sampling distribution (Cf also [15])


In the current version of the paper, on page 13, **margin bounds are presented as** part of examples of bounds that are **"distribution independent", which is quite disputable**.  The paper also similarly presents a **blanket dismissal of so-called "Rademacher bounds" on page 13**. Almost every bound, including actively distribution-dependent bounds such as [3,4] relies on Rademacher arguments in some part of the proof. Given the current nature of the reviewing process and the hype-driven personality of the community, publishing/highlighting the current version of the paper would put a lot of legitimate future research at high risk of being unfairly dismissed. Overall, the main paper and appendices B,C and D should be somewhat toned down.

It would also be nice to describe the relationship between the current work and the benign overfitting literature [13,14]




=====================Minor errors/issues of a mathematical/presentation nature=======


Statement of Theorem 3 on Page 8: I think the authors meant to write $S_0~D^n_{I_{0}}$... before "where $I_0=...$" rather than the other way around.

In Lemma 1, the symbol represented by a small rhombus presumably refers to function composition but is not defined.


In the statement of Theorem 5, the fact that $q=2$ is not stated, though I believe it is an assumption in the Theorem.


The top of page 21 is not very well organized: the last component of the "definition 8", the definition of $A_{i}$ is problematic since it uses the words "is biased towards selecting consistent hypotheses in $lin_{q,i}(d,n)$" without explaining what it means (I know it can be inferred by analogy with definition 5 in the main paper, but it technically should still be defined more properly here).

Importantly, the remark on page 21 only makes sense if one already has made an attempt at reading the proof of Theorem 4. It should be incorporated into the proof of Theorem 4 and explained more rigorously.


There are several minor issues with the proof of Lemma 3: firstly, in the first line, $f_a,f_b$ are defined as "parity functions", which seems to hint that we are working with the case $q=2$ (that is not the case). Furthermore, only $f_a$, not $f_b$, is used in that particular paragraph, though an arbitrary $f_b$ in the null space of $X^{-}$ is used in the next paragraph. The last sentence of the third paragraph takes some time to be digested: perhaps a quick mention of realizability can help the reader.

In the fourth paragraph of the proof of Lemma 3, I think writing "there always exist **at least** $n+1-k$ canonical basis vectors which are not spanned..." would be much better than the current "there always exist in total $n+1-k$ canonical basis vectors which are not spanned...", since this statement takes a bit of effort to mentally process and I think there can be more than $n+1-k$ vectors which are not spanned!


In the second line of Section G.4 on page 23, I think the authors meant to write $\mathcal{H}_1$, not $\mathcal{H}_0$. There should also be a better definition of $\mathbb{A}_0$ in this part of the text (just a reminder would help, "an algorithm is in $\mathcal{A}_0$ if the following statement holds: "if there exists a consistent hypothesis in $\mathcal{H}_0$, the algorithm must chose one such hypothesis".


The end of the proof of lemma 4 is badly organized: the last two paragraphs **should be swapped**, and the sentence "the claim follows" should follow as a separate last paragraph.

On page 24, in the paragraph that begins with the estimation of $\mathbb{P}(E_3|E_{4,k})$, I think the authors meant $\mathbb{P}(E_3|E_{4,k})=1-\mathbb{P}(E_3^{c}|E_{4,k})$, not $\mathbb{P}(E_3|E_{4,k})=1-\mathbb{P}(E_3|E_{4,k}^{c})$.


The argument at the top of page 26 needs to be reformulated: "which implies that $1-q^{-2} \in 1-o(1/q)$ is not the right statement: the fact that $1-q^{-2} \in 1-o(1/q)$ is not deduced from the previous statement, instead, it is the first statement in the induction case to show that the expression on the line above is in $1-o(1/q)$.

On page 26 (just like in the main paper), the statement of Theorem 5 should include the assumption that $q=2$.


The proof of Theorem 3 could be slightly improved: the sentence "this yields what we wanted" at the top of page 20 feels a bit abrupt since it has not yet been made clear that the statement below will be used to show that $\mathcal{A}$ is not estimable.

Right in the middle of page 20 in the proof of Theorem 3, the definition of the even $B$ is missing a $>\alpha$ before the last curly bracket.

===========Typos/very minor presentation comments=============


The table of notations should be substantially improved to include all the notations in the paper including $R_q(.,.,.)$, $Lin_q(d)$ etc.


In the appendix, the names of Theorems 1 and 2 have been interchanged (Theorem 2 in the appendix is actually Theorem 1), and Theorem 1 in the appendix is actually Theorem 2 (as can be seen from its introduction on page 18 which says, "we now use Theorem 1 to show the following")


In the Statement of Theorem 1 (referred to as "theorem 2") on page 16, there is an issue with the punctuation (,.)


There is a period missing at equation (8).


In page 22, proof of Lemma 2, the second part of the proof actually doesn't rely on the theorem from [1], so it is a shortcut. This is not clear from the formulation. At a minimum, the first sentence of the second paragraph could read "the second statement can be inferred from the first, but also follows directly from a simple..."


In the proof of Lemma 4 on page 23, first paragraph, there is an extra space after "w.r.t.".
In the proof of Lemma 4 on page 23,  "be it deterministic of random" should be "be it deterministic or random"


The paragraph starting with $\mathbb{P}(E_1|E_{2,i}\cup E^{c}_3\cup E_{4,k})=1-q^{k-n-1}$ should be slightly reorganised to avoid the repeated use of "since".

There seems to be an extra space before equation (25) and there is a period missing at equation (30).


At the top of Page 25, there is an extra space between  $\|$ and $P$ in equation (31).


In pages 26 and 27, there are plenty of equation numbers which are not in brackets. The corresponding "\ ref {}" should be changed to "~ \ eqref { } "


In the proof of Theorem 5 on page 27, there are plenty of "$lin_{2,0}$ which should probably be $lin_{2,0}(d,n)$.

There is a period missing at the last equation of page 19.

Just before the bullet points on page 20: "at least one of the following items hold" should be "at least one of the following items holds"


=====================================

References:

[1] IAN F. BLAKE AND CHRIS STUDHOLME, PROPERTIES OF RANDOM MATRICES AND APPLICATIONS, 2006

[2] Omer Angel and Yinon Spinka. Pairwise optimal coupling of multiple random variables, 2021. (Arxiv)

[3] C Wei, T Ma, Improved Sample Complexities for Deep Networks and Robust Classification via an All-Layer Margin, ICML 2019.

[4] Sanjeev Arora, Simon S. Du, Wei Hu, Zhiyuan Li, Ruosong Wang, Fine-Grained Analysis of Optimization and Generalization for Overparameterized Two-Layer Neural Networks, ICML 2019.

[5] Vaishnavh Nagarajan, J. Zico Kolter, Uniform convergence may be unable to explain generalization in deep learning, NeurIPS 2019

[6] Philip M. Long, Hanie Sedghi, Generalization bounds for deep convolutional neural networks, ICLR 2020

[7] Peter Bartlett, Dylan J. Foster, Matus Telgarsky. Spectrally-normalized margin bounds for neural networks, NeurIPS 2017.

[8]  Wei and Ma, Data-dependent Sample Complexity of Deep Neural Networks via Lipschitz Augmentation, NeurIPS 2019

[9] Vaishnavh Nagarajan and Zico Kolter. Deterministic PAC-bayesian generalization bounds for deep networks via generalizing noise-resilience, ICML 2019.

[10] Antoine Ledent, Waleed Mustafa, Yunwen Lei, Marius Kloft, Norm-based generalisation bounds for multi-class convolutional neural networks, AAAI 2021.

[11] Florian Graf, Sebastian Zeng, Bastian Rieck, Marc Niethammer, Roland Kwitt, On Measuring Excess Capacity in Neural Networks, NeurIPS 2023.

[12] Size-Independent Sample Complexity of Neural Networks.  Noah Golowich, Alexander Rakhlin, Ohad Shamir, COLT 2018.

[13] Peter L. Bartlett, Philip M. Long, Gábor Lugosi, Alexander Tsigler,  Benign Overfitting in Linear Regression

[14] Shamir, The Implicit Bias of Benign Overfitting, COLT 2022.

[15 ] Tomer Galanti, Mengjia Xu, Liane Galanti, Tomaso Poggio ,  Norm-based Generalization Bounds for Compositionally Sparse Neural Networks

**Questions:**

1. Does Theorem 1 only apply to a discrete state space? In the proof towards the end of Page 16, you write down $P_1(S,h)=P_2(S,h)$, which only makes sense if we are in the discrete setting. This should be clearly stated as an assumption in the theorem.


2. Could you explain the symmetry argument from the remark on Page 21 a bit more carefully? I am having trouble fully understanding it.


3. Why does the argument you rely on in the proof of Theorem 5 not work in general for Theorem 4?

---

> ### Author Response · Authors · 2023-11-13
>
> Thank you for your detailed feedback, which includes much appreciated comments regarding the fine details in the appendix.
>
> Before addressing your questions in detail, we would like to first clarify a few high-level points.
>
>
> It seems to us that some of your questions stem from a disconnect of our definition of ‘distribution-dependent’ and the one you have in mind. For example in your third point, you mention:
> > There are plenty of papers that actively take the sampling distribution into account in their analysis and are not even cited here. The most obvious examples would be [3] and the NTK literature [4].
>
> However, the bounds in these works only take into account the training set and the output hypothesis, but not the properties of the distributions themselves.
> By ‘distribution dependent’ bound we mean (throughout the paper) that the bound takes into account some property of the underlying data distribution (e.g., boundedness, information about the moments, smoothness…). However, note that receiving a training sample does not qualify a bound for being distribution dependent.
>
> Relatedly, you are asking:
> > In the current version of the paper, on page 13, margin bounds are presented as part of examples of bounds that are "distribution independent", which is quite disputable.
>
> All bounds we mention in Appendix B do not assume anything about the algorithm being considered or the underlying distribution. Hence they fall within the setting of Theorem 1. If you disagree, can you lay out your argument based on one paper from that list? For example, why is this disputable over the margin-based bounds? They are just a function of the training set and the output hypothesis.
>
> Does this clarify some of your questions regarding distribution dependence?
>
> Further, you are asking:
> > Which bounds in the existing literature are really "in the overparametrized setting"?
>
> We are not sure what you mean when you call a bound being in the overparametrized setting – overparametrization is a quantifier of a learning task, not of a generalization bound.
>
> Since ‘overparametrized’ means (roughly speaking) not having enough training samples to learn over all distributions, and none of these bounds specifically exclude this regime, all our results apply to all of the distribution-independent bounds that we cite, once one is situated in this learning setting.
>
> Does this answer your question? If not, could you please expand on this point?

---

> > ### Comment · Reviewer_bL1w · 2023-11-14
> > **Distribution dependency**
> >
> > Thanks for your quick reply. Let's try to figure this out.
> >
> > I disagree with your points regarding distribution dependence and overparametrization. Although as I said, the case of margin bounds is a borderline one (although your results don't apply at face value, they may apply in principle), it seems absurd to me that you are claiming that **NTK bounds** such as [4] are not distribution dependent simply because the **distribution dependency is estimated from the data**. Admittedly, it is necessary to assume boundedness of inputs (as nearly all results do), beyond that, more subtle properties of the distribution can be estimated. For instance, if the data lies in a low dimensional hyperplane (thereby de facto reducing the effective number of parameters), this can be observed from the data and leveraged in the results. This is partially what makes the bounds in [4] so good: only the NTK direction which aligns with the labels matters.
> >
> >
> >
> > Let us start with simple examples and work our way to the complicated ones later (discussing overparametrization, algorithm dependence *and* distribution dependence):
> >
> > **Example 1** (For this example I am not questioning that you are right, but I think it is still worth discussing): test set bounds. Suppose I split my dataset into two and use half of it for training and half of it for validation. Suppose the loss is bounded by 1 and the validation loss is $v$, my bound is now $v+\sqrt{\frac{2\log(1/\delta)}{N}}$. As far as I understand, this gives a very tight bound for any function in the class with probability $\geq 1-\delta$. It is also true that the estimate **only takes into account the training set**. This bound is "just a function of the training set" (your words).
> >
> > The only caveat being that we need to train only on the first part of the set which means that this is **not an ERM algorithm**. As far as I understand, **this is why** Theorem 1 doesn't apply in this case.  Why does Theorem 3 not apply now? It seems to me it is because Theorem 3 is extremely specific (to a situation with two distinct regimes with large enough total variation) and doesn't actually invalidate all "algorithm dependent bounds".
> > Now suppose you want to claim that margin bounds as in [13] are invalid by Theorem 1, you also  need to ensure that the algorithm is a strict ERM. That is a harder point to make than it seems: maybe because of the gradient descent procedure, the final solution is more likely to have a large margin.  If you want to claim that the bounds in [13] fall under theorem 3 instead (because of the implicit algorithmic dependency in the margin), it is not clear how to proceed either.
> >
> >
> > **1.2** That is in addition to the "overparametrization" issue above: there is no guarantee that the learning problem corresponding to a function class as in [13] is overparametrized.
> > Consider an extremely simplified version of margin bounds: as is common in the literature, we assume the inputs $x\in\mathbb{R}^d$  are of norm less than $1$ and we have a two class classification problem with the hypothesis of linear classifiers $w$ of norm less than $R$.  Consider the following margin-bound (grossly simplified version of the main result in [13]):
> >
> > $$\ell_{0-1}(test)\leq  \ell_{\gamma} (train) +\widetilde{O}(\frac{R}{\gamma\sqrt{N}}),$$
> > where $\ell_\gamma$ is the ramp loss. The bound doesn't depend on the dimensions and is certainly non vacuous for small values of the number of samples $N$. You will argue that it is not overparametrized, but what could possibly tell us that the same cannot happen when we add more layers? In this example, if we keep adding parameters by increasing dimension, we don't change the overparametrization because the norms are fixed. **There is no guarantee that this couldn't happen in the case of deeper networks** (especially if scaling issues are handled with loss function augmentation [5,8,10,11]).
> >
> >
> > **Going further**, suppose we have two layers and both are restricted to have spectral norm of 1. If, with inputs of norm 1, we still get a margin greater than $\gamma$ ($\gamma$ close to 1), then the only way this can happen is if the top eigenvectors of both layers approximately align. **This restricts function class capacity indirectly through the (optimized) margin parameter**. Suppose that the bound takes this into account (something similar to this happens in [3,4]), which of your theorems do you claim applies? It seems Theorem 1 doesn't apply because we are not doing ERM over a fixed function class, and Theorem 3 doesn't apply either. Would you count this as "taking the distribution into account" (even though "the bound is just a function of the training set"), or would you say that this is not an overparametrized setting?

---

> > > ### Author Response · Authors · 2023-11-14
> > >
> > > Thanks again for your comprehensive response!
> > >
> > >
> > >
> > > Let us continue by first addressing one source of confusion that appears in your review that is related to your answer above.
> > >
> > > > Theorem 1 states that if we consider a function class associated to a classification problem with the 0-1 loss and a finite set of realizable distributions, as long as the problem is overparametrized (i.e., not reliably learnable over most distributions), then there exists an ERM algorithm whose error cannot be reliably estimated either.
> > >
> > > This is not an exact description of Theorem 1. Theorem 1 shows that any estimator does not only fail for one (algorithm, distribution) pair, as you state above, but for a large fraction of possible (algorithm, distribution) pairs. In fact, when applied to VC classes, this fraction is 1/2 (Theorem 2). This is what makes Theorem 1 technically and quantitatively different from the no-free-lunch theorem (see our proof sketch after Definition 3).
> > >
> > >
> > >
> > >
> > > >  As far as I understand, this is why Theorem 1 doesn't apply in this case.
> > >
> > > This is not the only reason. We could derive a similar Theorem 1 for general algorithms. Since Theorem 1 is a negative result, restricting the set of possible algorithms the estimator considers to ERMs only makes the theorem stronger.
> > >
> > > > As far as I understand, this gives a very tight bound for any function in the class
> > >
> > > But the thing is, you want to estimate the performance of an algorithm you have in hand and not the loss of some specific function. Moreso, you want your estimator to work for any algorithm without knowing beforehand what algorithm that would be (Theorem 1 is about algorithm-independent bounds). Theorem 1 shows that no estimator can estimate a large fraction of (algorithm, distribution) pairs. The one you presented included. It succeeds for the specific algorithm you presented but will fail for many other (algorithm, distribution) pairs.  Theorem 1 deals with algorithm-independent bounds, and yours is an algorithm-dependent one that works only for the specific algorithm you suggested (or close variants of).
> > >
> > >
> > > Lastly, we now noticed that papers [4] and [13] are about regression, and since our work here is about classification, we think it is better not to address them in our current comment (so we don't add unnecessary confusion).
> > >
> > > Can you maybe address another paper about the generalization of neural networks for classification? Alternatively, we mention that we specifically address [7] (margin and norm-based bound) in Appendix C.
> > >
> > > Does this clarify matters more?

---

> > > > ### Comment · Reviewer_bL1w · 2023-11-15
> > > >
> > > > Thanks for the further explanations about theorem 1. It has been a while since the original review.
> > > >
> > > >
> > > > I feel you still haven't addressed my point: in the main paper, you make strong claims that almost all existing research is invalid or trivial because your theorems prove that those bounds are necessarily vacuous. However, existing results are complicated to interpret and your theorems rely on strong assumptions which are impossible to verify in practice.
> > > >
> > > > You can't go back and forth between vagueness and precision to your own advantage. For instance, regarding overparametrization, it not fair to, as you do, **first** write in the paper that it is commonly accepted that "most real world problems are in the overparametrized setting" and use this justify saying that all of your theorems apply to those papers, and **then**, when challenged by another reviewer about your own definition of overparametrization, argue that you can do anything you want any way because the concept has not been rigorously defined in the community in a strict mathematical sense yet (which is true). In most real life scenarios, the only thing that is really clear about overparametrization is that the number of parameters is greater than the number of samples, a much weaker condition.
> > > >
> > > >
> > > >
> > > > Back to margin bounds, the results typically state that with high probability over the training set, the generalization gap is bounded by $f(\|W\|,\gamma)$ where $\|W\|$  is some norm-like quantity calculated from the weights of the network and gamma is a margin parameter or set of margin parameters (see [3]).  Here, the quantities $\|W\|$ and $\gamma$ are calculated *after* training and the theorem nonetheless holds with high probability over **all** such choices thanks to a *union bound*. It is very hard to strictly interpret this as falling under the conditions of your Theorem 1 or 3, because the choice of the function class is influenced by the (success of) the *training algorithm*, which itself is influenced by the *distribution*: it is possible that we reach very small weights and a very large margin, in a way that can only be achieved when the weights are constrained to some region of space with certain properties, or with empirical distributions that satisfy certain properties. For instance, if we work with points on the unit sphere in a binary classification problem with weights of unit norm, a margin of 0.99 is only achievable if it happens that the samples in both classes lie in nearly exactly diametrically opposite positions. This restricts the choices of possible distributions and the bound improves as a result. Effects analogous to this are taken into account to some extent even in rudimentary margin bounds such as [13], and are taken into account in much more comprehensive ways in works such as [3] and [4].
> > > >
> > > >
> > > > Whilst [4] and [13] indeed work with regression rather than classification, this is only a technicality: it doesn't influence the philosophy of the arguments. You certainly don't make it abundantly clear in your paper that your results should only be interpreted as applying to classification rather than regression and that the situation could be widely different in the regression case. Thus, I am still not convinced that any of the following is wrong: (1) it is questionable to what extent margin bounds (or any bounds other than VC or parameter-counting bounds) are covered and (2) NTK results such as [4] should be interpreted as distribution dependent and do not fall under your framework. Note that [4] still uses Rademacher complexity in parts of their argument (of course) despite indirectly relying on the distribution quite a bit, and could therefore be misunderstood as being covered by your theorems by a casual reader of your blanket dismissal of all ``Rademacher bounds''.

---

> ### Comment · Reviewer_bL1w · 2023-11-14
>
> **Example 2**
>
> About the main result in [4]:  Generalization bound $\leq \widetilde{O}\sqrt{\frac{y^\top H^{-1}y}{n}}$. Where would you classify this example? Does Theorem 3 apply to it? It seems highly likely that such a bound is not vacuous in practice, and almost impossible to determine whether your definition of "overparametrization" holds in this case.

---

> ### Author Response · Authors · 2023-11-17
>
> > in the main paper, you make strong claims that almost all existing research is invalid or trivial because your theorems prove that those bounds are necessarily vacuous.
>
> To be clear, we do not claim that "almost all existing research is invalid or trivial". We claim that many published bounds are not uniformly tight, meaning that there exist (distribution, algorithm) pairs for which the bound will not be tight (in fact, we show there are many such pairs). This does not mean that exisiting research is "invalid", "trivial", or "vacuous". We are showing a specific type of limitation for bounds that are distribution and algorithm independent.
>
> > However, existing results are complicated to interpret and your theorems rely on strong assumptions which are impossible to verify in practice.
> >
> > You can't go back and forth between vagueness and precision to your own advantage. For instance, regarding overparametrization, it not fair to, as you do, **first** write in the paper that it is commonly accepted that "most real world problems are in the overparametrized setting" and use this justify saying that all of your theorems apply to those papers, and **then**, when challenged by another reviewer about your own definition of overparametrization, argue that you can do anything you want any way because the concept has not been rigorously defined in the community in a strict mathematical sense yet (which is true). In most real life scenarios, the only thing that is really clear about overparametrization is that the number of parameters is greater than the number of samples, a much weaker condition.
>
> Following your comments, we have written a detailed and careful answer explaining why our results in Theorem 1 and 2 apply to deep learning settings. Kindly read this explanation in **_General Answer 1: Our Definition of Overparametrization_** above (including the part **_Why Theorems 1 and 2 Apply to Generalization Bounds For Deep Learning_**). As we explain there, our results in Theorem 1 and 2 basically require only two assumptions:
> 1. Distribution and algorithm independent generalization bound. Namely, the generalization bound of interest does not explicitly state which population distributions it applies to, and does not rely on the specifics of the training algorithm.
> 2. This is a deep learning setting, so in particular the number of edges in the neural network is much larger than the sample size.
>
> Both of these are straightforward assumptions that can readily be verified. While we agree that "existing results are complicated to interpret" in various ways, checking these two assumptions does not require any complicated interpretation. One can simply read the statement of the generalization bound of interest to see if it is stated for a general population distribution and a general algorithm, and check that the neural network has a number of edges much larger than the sample size. That's pretty much all that is needed for Theorems 1 and 2 to apply.

---

> ### Author Response · Authors · 2023-11-17
>
> ## **Margin Bounds**
>
> You raise the question of whether our negative results in Theorems 1 and 2 apply to margin bounds. We assert that they do indeed apply. We will explain this both via a concrete example, and then in a more general discussion.
>
> ### Example: A Margin Bound That Is Not Uniformly Tight
>
> To follow up on the specific example you mentioned in [your comment](https://openreview.net/forum?id=NkmJotfL42&noteId=92pbVP25Si) above, let the domain $\mathcal{X} = \\{x \in \mathbb{R}^d: ~ \\|x\\|_2 = 1\\}$ be the unit sphere in $\mathbb{R}^d$. Consider a half-space classifier, and a margin bound of the form $f(|W|, \gamma)$ as you described, such that $f$ (an upper bound on the difference between the population and empirical loss) is small only if the margin $\gamma$ is large. We will show that $f$ is not uniformly tight.
>
> Indeed, consider a population distribution $\mathcal{D}$ that has (say) a uniform marginal on the domain $\mathcal{X}$, with labels that are generated by a specific half space $h^*$ (e.g., the half space that passes through the origin and is perpendicular to the $x_0$ axis). Consider an algorithm $A$ the has a bias towards $h^*$, in the sense that it will output $h^*$ whenever the training set is consistent with $h^*$.
>
> In this case, given samples from $\mathcal{D}$, with probability $1$, $A$ will output $h^*$. Hence, the population loss and the empirical loss are the same (they are both exactly 0). Because the marginal of the population distribution on the domain is uniform, the population margin is 0, and therefore the empirical margin $\gamma$ will be close to $0$ (even for a training set that is small compared to the dimension $d$). Thus, $f(|W|, \gamma)$ is large, but the generalization gap is 0. Hence, the bound $f(|W|, \gamma)$ is not tight for the pair $(\mathcal{D}, A)$.
>
> Note that our results in the paper are stronger, in the sense that they show that there exist many (distribution, algorithm) pairs for which the bound is not tight, not just one.
>
>
> ### General Case: Why Margin Bounds Are Not Uniformly Tight
>
> More generally, in margin bounds the difference between the empirical and population loss is bounded by a quantity $f = f(|W|, \gamma)$. This quantity $f$ is strictly a function of the selected hypothesis and of the training sample. As such, a bound of this kind satisfies the algorithm- and distribution-independence requirements of Theorem 1. If the bound is applied to a neural network that has a lot more edges than the size of the training set, then this is an overparamterized setting (as we explain in detail in _**General Answer 1: Our Definition of Overparametrization**_), and therefore Theorem 1 implies that the bound $f$ will not be uniformly tight. This is a dry logical consequence of Theorem 1.
>
> We agree with the observation that the quantities $|W|$ and $\gamma$ depend in nontrivial ways on the population distribution and on the training algorithm, and that therefore the quantity $f(|W|, \gamma)$ depends indirectly on the on the population distribution and on the training algorithm. This is not a problem, because this is always true: a bound that depends on the selected hypothesis and on the training sample must always depend indirectly on the training algorithm and on the population distribution. That does not detract from the applicability of Theorem 1.

---

> ### Comment · Reviewer_bL1w · 2023-11-19
> **Please upload substantial a revision**
>
> Thanks for your comment.
>
>
> From this comment, I think we are getting closer to understanding each other. It seems that we agree on the strict mathematical facts, but not on the implications in terms of how you informally explain your results: I think the problem lies in our different opinions regarding how bad it is to not be "uniformly tight".
>
> I agree from your example that theorem applies there (assuming "overparametrization", which again, in this example, is NOT implied by the statement that there are more parameters than training samples), but the distributions for which the bound is vacuous will not be the "interesting" distributions towards which margin parameter is biased. For instance, if the margin is close to 1, it means the data has to lie approximately in two diametrically opposite points, and the bound can be very tight. It doesn't mean that there don't exist many distributions for which the bound is not tight, but they can be identified as unfavorable distributions by seeing that the margin is not close to 1. The art of learning theory in its current ("distribution-independent") form is to formulate the problem in a feature space where this analogy holds for real life observed empirical distributions.
>
> Again, I think your results are worthy of publication if the informal statements which accompany them are very substantially toned down (in the above example, it is interesting to know that there are many distributions for which the margin will not be that high **and** the bound will not be tight). Your paper reads as if the existing approaches are invalidated by your results. For the above reasons (**please read my more detailed comments in the main thread**), this is not the case at all.  Most of the statements you make in sections C and D can be interpreted as misleading when taking all those facts into account. For instance, saying that "alex has no specific description of $\mathcal{H}_0$ and "Alex's analysis is lacking" are both disputable statements: it is disputable whether "Alex" in fact has a way of telling if we are likely to be in $\mathcal{H}_0$ based on an empirical observation (of a high margin or low intrinsic dimension), and it is disputable whether Alex knows he is in an overparametrized setting.
>
> Also, you say "This story highlights our point in Theorems 1 and 2, which is that bounds that do not explicitly depend on the learning algorithm must be vacuous for many algorithms, and in particular are likely to be vacuous for SGD or any specific algorithm of interest."  This is only true for certain distributions  (it is not true of "SGD" in general): further, this is **known** (in less precise terms than the ones in your theorems): the bounds are only intended to work on data which is "natural" enough to make the margin/other parameters behave sufficiently favorably. Saying that your theorem implies that these existing results are vacuous for many distributions is like saying that a truly (explicitly) distribution dependent algorithm is vacuous/false when the distribution does not satisfy the prescribed condition. In both cases, this is hardly as much of a cause for concern as you hint.
>
>
> As I said in my review, the current version is dismissive of much of the literature in a way which is not fully justified and is very likely to mislead the reader and the community. You ought to upload a significant revision that substantially tones down your statements (I know it will take a lot of work, but I think it's necessary). I don't think **you results are** quite as far reaching as you claim in the first version of the paper, but they are still very **interesting**. If (and **only if**) you can describe them in a way that will not mislead the readers, they will become highly worthy of publication and greatly benefit the community.  If you stick to the current version, I will have to lower my score.

---

> > ### Author Response · Authors · 2023-11-21
> >
> > Dear Reviewer bL1w,
> >
> > Kindly take a look at _**General Answer 3: New Revision of The Paper**_ above, and at the PDF version of the paper.
> >
> > Best,
> > The authors

---

### Official Review · Reviewer_iLzH · 2023-10-30

**Soundness:** 3 good
**Presentation:** 4 excellent
**Contribution:** 3 good
**Rating:** 8
**Confidence:** 3

**Summary:**

This paper tackles the question of the tightness of existing generalisation bounds to any learning problem of interest. Authors provably show that bounds being algorithm and data-distribution independent cannot reach uniform tightness in the overparametrised setting. They also show that, as modern generalisation bounds are often algorithm-dependent (while still independent of the data distribution), it is impossible for this type of bound to reach tightness uniformly for all data-distribution. Those limits suggest for future work to focus on generalisation bounds exhibiting how data-distribution and algorithmic performances intricate.

**Strengths:**

- This work, especially the introduction, is utterly well-written and organised, it has been a pleasure to read it.
- The proposed results have the potential to be impacting for the whole generalisation field, as they suggest to give up the current shape of state-of-the-art generalisation bounds to direct future works on bounds focusing on the role of the data-distribution.

**Weaknesses:**

- I have no problems with results (although I did not read carefully the proofs). However, several paragraphs in this paper, not only provide unfair analysis of existing literature but also contains wrong claims concerning existing works. I strongly believe those paragraphs have to be re-written before acceptance, see the Questions section below for details.

**Questions:**

- I believe the notion of overparametrised setting should be explained more carefully and clarified. Indeed, in the introduction, authors precise that such a setting 'roughly means that the number of parameters in the networks is much larger than the number of examples in the training set' while Definition 2 does not even make intervene the notion of parametrised space. The link with neural nets should be explicited.
- page 6 'analouge of Theorem 1' -> analogue
- About the implications of Theorems 4 & 5, can you be more specific about the links with practical neural nets? I read Appendix D and I am wondering about the recommendation given at the end of the section: how is it possible for theoreticians to either 'exclude many distributions over linear functions' if one doesn't know which one to focus on? Second, how realistic is it, in terms of computational time, to verify that 'the neural network architecture with SGD cannot learn any large linear subspace of functions'?
- From a broader perspective, I believe that some claims in the paper are more a matter of personal interpretation than scientific facts and should be reformulated. For instance, author claims in section 5.1 that 'Most published generalization bounds do not restrict the set of distributions or algorithms the bound should apply for. Hence, they should work in all scenarios.' This is not true, having a bound holding for any distribution does not imply that the bound will be tight on all situations (and this is precisely what your work is proving). On the contrary, there may be several interests to derive a generalisation bound, a practical one is deriving generalisation-driven learning algorithms who may have a practical performance tighter than the associated bound as in Dziugaite & Roy 2017, or again simply propose a certain measure of complexity as in Neyshabur et al. 2017 to check if, in a few concrete learning problems, this complexity measure is tailored to explain generalisation. To me, the sentence at the end of Appendix D: 'At the very least, for a generalization bound for a neural network architecture trained with SGD to be meaningful, it must satisfy either of the following items...' is misleading as you assimilate the notion of 'meaningful' to 'precisely explaining the tightness on a given situation'. From my understanding of your work, I would say that existing generalisation bounds are too generic to provide such a precise understanding and are only an intermediary step which has to be completed by a study of the intrications between the data distribution and a learning problem.
   Similarly, the analysis of the bound of Dziugaite & Roy (2017) in section 2.3 would gain to be reworked. Indeed, the sentence 'the bound in Dziugaite & Roy (2017) relies on implicit assumptions about the algorithm and the population distribution' is simply not true as it relies on the McAllester's bound which does assume anything on the data distribution. However, the efficiency of their algorithm on MNIST is effectively not due to the PAC-Bayesian bound but to more subtle assumptions satisfied by the specific learning problem of interest and is not easily extendable to other learning problems.
   Finally, claiming that 'explicitly stating the assumptions underlying generalization bounds is not only necessary for the bounds to be mathematically correct...' is utterly misleading as it suggests that existing bounds are mathematically false while they are only not able to ensure tightness of learning algorithms in many situations: a vacuous bound is not mathematically flawed!
   To me it is necessary to rework those paragraphs in order for the paper to be published as they do not give a fair perspective on the existing literature.

**Conclusion** My current score is linked to the writing of the few paragraphs I mentioned above, I believe they have to be re-written to give a fairer comparison with literature and to remove false claims. That being said, I would be happy to increase my score conditionally to such rewriting, as I think this work is of the highest interest for the generalisation literature.

---

> ### Author Response · Authors · 2023-11-13
>
> After carefully reading your review, we would like to confirm (or clarify) our understanding before providing our response in full detail.
>
> We will be happy to reach a common understanding regarding the interpretation of our theorems.
>
> For example:
>
> > For instance, author claims in section 5.1 that 'Most published generalization bounds do not restrict the set of distributions or algorithms the bound should apply for. Hence, they should work in all scenarios.' This is not true, having a bound holding for any distribution does not imply that the bound will be tight on all situations (and this is precisely what your work is proving).
>
> We think the confusion stems from ‘should work’ which you interpret as ’should be tight’. Is that correct? We meant ‘should work’ = ‘should hold’.
>
> So we suggest changing our text as follows:
>
> Most published generalization bounds do not restrict the set of distributions or algorithms the bound should apply for. Hence, they _should upper bound the risk_ in all scenarios. Because these bounds do not distinguish between distributions that an algorithm performs well on and distributions that it does not, such bounds have no other option but to declare the largest true error possible (and become vacuous).
>
> Does this clear things up in this example?
>
> Is there an agreed-upon definition of overparametrization in the literature? So we can refer to it in our answers.
> If not, how would you mathematically define overparametrization? How would you define such a concept over non-parametrized spaces? Please note that Section 4 gives the motivation for our definition.

---

> > ### Author Response · Authors · 2023-11-16
> > **Response to all points of Reviwer iLzH**
> >
> > The following is a complete response to all points raised by reviewer iLzH. Please let us know if now your questions are properly addressed.
> >
> > > I believe the notion of overparametrised setting should be explained more carefully and clarified.
> >
> > Please see our general discussion **General Answer 1: Our Definition of Overparametrization** on the top of the comment section.
> >
> > > page 6 'analouge of Theorem 1' -> analogue
> >
> > Thanks! corrected.
> >
> > > About the implications of Theorems 4 & 5, can you be more specific about the links with practical neural nets?
> >
> > What we wrote in Appendix D is just the immediate **logical** implication of our work from Theorem 4:
> >
> > * Exclude in advance many distributions over linear functions.
> >
> > * Show that the neural network architecture with SGD cannot learn any large linear subspace
> > of functions.
> >
> > If either one is true, then the bound that was derived is necessarily vacuous in some scenarios for neural networks, even if the bound was specifically tailored for SGD with a specific architecture.
> >
> > > How is it possible for theoreticians to either 'exclude many distributions over linear functions' if one doesn't know which one to focus on?
> >
> > This is exactly our point. If one is willing to take a more **intuitive** leap or implication  (not a **logical** one), we argue and advocate that a more bottom-up approach is preferred. We think it is better to study very specific combinations of distributions with specific classes of algorithms while keeping in mind that no bound can work for all distributions. Essentially, let go of generality (distribution-independent bound) and aim towards tighter analysis of specific cases.
> >
> > Also, perhaps it's better first to uncover and study properties of distributions (such as natural images, natural language, etc.) and only then try and derive bounds tailored for distributions with said properties.
> >
> > > Second, how realistic is it, in terms of computational time, to verify that 'the neural network architecture with SGD cannot learn any large linear subspace of functions'?
> >
> > Probably, one cannot verify this empirically, maybe only provide a mathematical proof. Since this is so hard, it again softly implies the abovementioned direction.
> >
> >
> > > For instance, author claims in section 5.1 that 'Most published generalization bounds do not restrict the set of distributions or algorithms the bound should apply for. Hence, they should work in all scenarios.' This is not true, having a bound holding for any distribution does not imply that the bound will be tight on all situations (and this is precisely what your work is proving).
> >
> > We think the confusion stems from ‘should work’ which you interpret as ’should be tight’. Is that correct?
> > We meant ‘should work’ = ‘should hold’. So we suggest changing our text as follows:
> >
> > Most published generalization bounds do not restrict the set of distributions or algorithms the bound should apply for. Hence, they should upper bound the risk in all scenarios. Because these bounds do not distinguish between distributions that an algorithm performs well on and distributions that it does not, such bounds have no other option but to declare the largest true error possible (and become vacuous).
> >
> > Does this clear things up in this example?
> >
> >
> > > On the contrary, there may be several interests to derive a generalisation bound, a practical one is deriving generalisation-driven learning algorithms who may have a practical performance tighter than the associated bound as in Dziugaite & Roy 2017, or again simply propose a certain measure of complexity as in Neyshabur et al. 2017 to check if, in a few concrete learning problems, this complexity measure is tailored to explain generalisation.
> >
> > We agree with your perspective! This fits with the implications of Theorem 4 we discussed above!
> >
> >
> >
> > > To me, the sentence at the end of Appendix D: 'At the very least, for a generalization bound for a neural network architecture trained with SGD to be meaningful, it must satisfy either of the following items...' is misleading as you assimilate the notion of 'meaningful' to 'precisely explaining the tightness on a given situation'.
> >
> > We agree with your perspective. Suggested replacement:
> >
> > "A generalization bound for a neural network architecture trained with SGD can be at most an intermediary step to our understanding of neural network generalization if it doesn't satisfy one of the following items: ...".

---

> > > ### Author Response · Authors · 2023-11-16
> > > **Response to all points of Reviwer iLzH - continued**
> > >
> > > > Similarly, the analysis of the bound of Dziugaite & Roy (2017) in section 2.3 would gain to be reworked. Indeed, the sentence 'the bound in Dziugaite & Roy (2017) relies on implicit assumptions about the algorithm and the population distribution' is simply not true as it relies on the McAllester's bound which does assume anything on the data distribution.
> > >
> > > Here we disagree :) The fact that the McAllester bound does not assume anything on the data distribution in its derivation, does not mean one cannot make it distribution-dependent and algorithm-dependent. This is true since if you have some prior knowledge of your algorithm and the underlying distribution, you can use this prior knowledge in the prior $P$ in the term $KL(Q||P)$ in the bound. We believe that Dziugaite & Roy (2017)  achieved a non-vacuous bound for MNIST because for the prior they chose, they implicitly assumed that for the encountered distribution, the algorithm finds a flat minima not far from the initialization.
> > >
> > > > However, the efficiency of their algorithm on MNIST is effectively not due to the PAC-Bayesian bound but to more subtle assumptions satisfied by the specific learning problem of interest and is not easily extendable to other learning problems.
> > >
> > > We think that what you write here is consistent with our paragraph above. It is indeed not easily extendable to other learning problems. For other networks and datasets, different priors might prove more beneficial (an example of distribution-dependent).
> > >
> > >
> > > > Finally, claiming that 'explicitly stating the assumptions underlying generalization bounds is not only necessary for the bounds to be mathematically correct...'
> > >
> > > Oops, you are completely right! This wrong phrasing somehow escaped us… What we meant in that sentence was just uniformly tight. Corrected: we replaced 'mathematically correct' with 'uniformly tight'.
> > >
> > >
> > > > I strongly believe those paragraphs have to be re-written before acceptance.
> > >
> > > Did our response address your concerns?

---

> > > > ### Comment · Reviewer_iLzH · 2023-11-17
> > > > **Thank you for your detailed response.**
> > > >
> > > > I thank the authors for their detailed response.
> > > >
> > > > Most of my initial concerns are mostly addressed.
> > > >
> > > > - "Here we disagree :) ..."
> > > >
> > > >   I believe there might be an ambiguity here about what a 'bound' is. I refer to a 'bound' as a mathematical theorem. So in my understanding, there is NO bound from Dziugaite et al. *stricto sensu* (as they use the McAllester one) but only a *numerical evaluation of a bound'. As long as we talk about this numerical value, I am supporting your analysis. I suggest you to make this clearer in the new version.
> > > >
> > > > Also, thank you for your general response about the definition of overparametrisation, with this precise discussion, I understand now clearly why all your results hold for a huge part of deep nets. I believe this discussion is crucially needed in the main paper.
> > > >
> > > > That being said, my initial concerns are addressed, but the whole discussion raised many routes of improvements in terms of writing. That being said, I think it is fair to let you re-write the document by incorporating the suggestions made by all reviewers before enhancing my score. To my understanding, it is possible to update the manuscript during the reviewing process (please correct me if I'm wrong)

---

> > > > > ### Author Response · Authors · 2023-11-21
> > > > >
> > > > > Dear Reviewer iLzH,
> > > > >
> > > > > Kindly take a look at _**General Answer 3: New Revision of The Paper**_ above, and at the PDF version of the paper.
> > > > >
> > > > > Best,
> > > > > The authors

---

> > > > > > ### Comment · Reviewer_iLzH · 2023-11-23
> > > > > >
> > > > > > Dear authors, thank you for your revision. I believe the paper is in a far more acceptable shape right now and move my score to 8 (as 7 is not an option).
> > > > > >
> > > > > > Best,
> > > > > >
> > > > > > Rev. iLzH

---

### Official Review · Reviewer_jDS8 · 2023-11-09

**Soundness:** 4 excellent
**Presentation:** 3 good
**Contribution:** 3 good
**Rating:** 6
**Confidence:** 4

**Summary:**

This paper considers the problem of understanding the tightness of two general families of the generalization bounds in the literature. The first family is the class of complexity measures that depend on the output of the algorithm and the training set, the second class is the class of algorithms that depends on the “description of learning algorithm” and the training set. The main question of the paper is there a family of generalization bounds that are uniformly tight in the overparameterized setting?

The paper proposes some natural definitions such as “overparameterized setting”: They define the overparameterized setting as the setting that with a given number of samples, accuracy, and confidence, a realizable distribution with respect to a family of hypotheses is not learnable.

The main result of the paper is that for these two families of the generalization bounds we can't prove a generalization bound which provide "uniformly" tight bound.

**Strengths:**

I think the main message of the paper probably is the most interesting part:

We need to develop an understanding of the formal assumptions on the algorithms and distributions under which a generalization bound can be tight. Otherwise, we can develop lower bounds as shown in this paper.

**Weaknesses:**

-- The main issue I found in the paper is that in many places the discussions are not precise. As an example the claim about cross-validation can be misleading: The authors claim that cross-validation approaches do not lead to algorithm design principles. It is not completely correct. For instance, the well-known algorithm of the one-inclusion graph [Haussler et al 1988] is based on the cross-validation analysis.

[Haussler et al 1988] Haussler, David, Nick Littlestone, and Manfred K. Warmuth. "Predicting [0, 1]-functions on randomly drawn points." Annual Workshop on Computational Learning Theory: Proceedings of the first annual workshop on Computational learning theory. Vol. 3. No. 05. 1988.


-- Proof of Theorem 2 and failure of the generalization bound: I checked the proof and it seems that the proof implies that the generalization gap is large for algorithms with large population error. I think it is unsatisfactory as we are interested to show a generalization gap is small for "good" learning algorithms


-- There are generalization bounds based on conditional mutual information which proven to be "tight" in the realizable scenario (For instance Haghifam et al 2022, Thm. 3.3.) It seems this class of generalization bound has not been discussed in this paper.

Haghifam, Mahdi, et al. "Understanding generalization via leave-one-out conditional mutual information." 2022 IEEE International Symposium on Information Theory (ISIT). IEEE, 2022.

**Questions:**

-- Definition of tightness can be improved: In many cases, we are interested in correlation of complexity measure and the actual generalization gap. It is not clear to me that the definition of tightness is the best possible.


-- In Page 14, there is this paragraph which clearly is not about Negrea et al. (2020).  This paragraph is unrelated to this paper.

“Negrea et al. (2020): The paper studies convex optimization, so the results can hold only for a single neuron. Nevertheless, although it gives matching lower and upper bounds, the bounds match only asymptotically when n is very large so the scenario is far away from the overparametrized regime (which is the focus of interest for neural networks).”

-- what are the challenges to extend the results to agnostic settings?

---

> ### Author Response · Authors · 2023-11-13
>
> After carefully reading your review, we would like to confirm (or clarify) our understanding before providing our response in full detail.
>
> Thank you for the reference [Haussler et al 1988]! We were not familiar with it. Can you refer us to the location of your claim about cross-validation? After searching the paper, the word ‘validation’ does not appear there. Regardless, your point makes sense. To avoid being imprecise, we will change the term cross-validation (which is indeed an umbrella term for several methods) to using a validation set (the holdout method - the simplest form of cross-validation), which is the staple for tuning the hyperparameters of neural networks.
>
> Do you find the above now a precise statement?

---

> > ### Comment · Reviewer_jDS8 · 2023-11-13
> >
> > The starting point of designing one-inclusion graph algorithm is the observation that the expected leave-one-out error is equal to the population risk. In their paper they call it "permutation mistake bound" (Lemma 2.2
> > https://machinelearning.pbworks.com/f/J27.pdf)
> >
> > My main concern as also raised by other reviewers is that in many parts of the paper the text is the authors "personal take of generalization theory". I found many parts interesting, however, it needs to be more precise since for non-expert readers it may be misleading.  Our discussion above regarding leave-one-out error is an example of imprecise statement which leads to confusion.

---

> > > ### Author Response · Authors · 2023-11-15
> > > **Response to all points of Reviwer jDS8**
> > >
> > > The following is a complete response to all points raised by reviewer jDS8. Please let us know if now your questions are properly addressed.
> > >
> > > > The main issue I found in the paper is that in many places the discussions are not precise. As an example the claim about cross-validation can be misleading: The authors claim that cross-validation approaches do not lead to algorithm design principles. It is not completely correct. For instance, the well-known algorithm of the one-inclusion graph [Haussler et al 1988] is based on the cross-validation analysis.
> > >
> > > We will correct this as discussed in our previous comment.
> > >
> > >
> > > > Proof of Theorem 2 and failure of the generalization bound: I checked the proof and it seems that the proof implies that the generalization gap is large for algorithms with large population error. I think it is unsatisfactory as we are interested to show a generalization gap is small for "good" learning algorithms
> > >
> > > If item 1 holds, for algorithms $\mathcal A_h$ for $h \in \mathcal H$ ($\mathcal A_h$ is an ERM that outputs $h$ for any sample $S$ consistent with $h$) the generalization gap is small when they have small true error.
> > >
> > > The proof shows that if item 1 holds, then the generalization gap is large for many ERMs associated with Definition 7 (Bayes-like Random ERM) when those algorithms have a large true error. So item 2 does not hold. This shows that both items 1 and 2 cannot hold simultaneously.
> > >
> > >
> > >
> > > The proof can be written almost the same way if you assume that item 2 holds. In that case, you'll get that the generalization gap is large for algorithms $\mathcal A_h$ when their error is small. You can essentially replace, "Assume item 1 in the theorem holds:" at the bottom of page 17 with "Assume item 2 in the theorem holds:" while changing the necessary notations and have that item 2 invalidates item 1.
> > >
> > >
> > > > There are generalization bounds based on conditional mutual information which proven to be "tight" in the realizable scenario (For instance Haghifam et al 2022, Thm. 3.3.) It seems this class of generalization bound has not been discussed in this paper.
> > >
> > > Thanks for the reference! We will add it to the information-theoretic bounds we addressed in our paper. The bound is indeed distribution-dependent (as we argue for in the paper, see column 3 in Table 1). Yet, Theorem 3.3 is asymptotic, the bound is tight for $n \rightarrow \infty$ (as far as we understand), so it is not immediately related to our setting (overparametrization). Please correct us if we are wrong.
> > >
> > > We mention that we addressed several distribution-dependent and information-theoretic bounds  (similar to Haghifam et al 2022)  in our paper. See Appendix B.3.
> > >
> > >
> > >
> > >
> > > > Definition of tightness can be improved: In many cases, we are interested in correlation of complexity measure and the actual generalization gap. It is not clear to me that the definition of tightness is the best possible.
> > >
> > > Interesting comment and question!
> > >
> > > But what is the right correlation measure to use? There are many possible choices, and it is likely there is no consensus on which are the most beneficial.  Since the tightness of generalization bounds is a well-defined mathematical question it makes sense to us to focus on that at this point. Do you agree?
> > >
> > > We lay it as a conjecture that if such a good correlation measure exists, then there exists a good generalization measure. Intuitively this might hold because if one can prove that a correlation measure is really good then we could artificially adjust it to be a generalization bound that works well (albeit without proof because maybe the analysis is too hard). Then, since our work shows that a good generalization measure does not exist, a good correlation measure should not exist as well. So we also conjecture that good correlation measures should not exist as well.
> > >
> > >
> > > > In Page 14, there is this paragraph which clearly is not about Negrea et al. (2020). This paragraph is unrelated to this paper.
> > >
> > > True, thanks for noticing this! We intended to cite "Lower Generalization Bounds for GD and SGD in Smooth Stochastic Convex Optimization".
> > >
> > > >  what are the challenges to extend the results to agnostic settings?
> > >
> > > There is a small comment regarding that on page 2: "Furthermore, if H is not estimable then there exists no uniformly tight bound as in Eq. (1) also for learning in the agnostic (non-realizable) setting."
> > >
> > > Our results are negative and show that with no assumption over the distribution, bounds will be vacuous in some scenarios in the realizable setting. This makes the theorems stronger compared to the agnostic setting. If an estimator does not exist for the realizable setting it doesn't exist for the agnostic setting, since the agnostic setting requires more from the estimator. We will add this explanation to the paper.

---

> > > > ### Author Response · Authors · 2023-11-21
> > > >
> > > > Dear Reviewer jDS8,
> > > >
> > > > Kindly take a look at _**General Answer 3: New Revision of The Paper**_ above, and at the PDF version of the paper.
> > > >
> > > > Best,
> > > > The authors

---

### Author Response · Authors · 2023-11-13

We thank the reviewers for their very detailed reviews and keen interest in our paper!

After carefully reading all the reviews and since there are many points to address, we would like to confirm (or clarify) our understanding of a few points raised by the reviewers before providing our response in full detail.

A question for all reviewers: Is there an agreed-upon definition of overparametrization in the literature? So we can refer to it in our answers.
If not, how would you mathematically define overparametrization? How would you define such a concept over non-parametric spaces?
Please note that Section 4 gives the motivation for our definition.

Please see other questions below each review that are associated with each reviewer.

---

> ### Comment · Reviewer_Hb6n · 2023-11-13
>
> I am not aware of a formal definition of over-parameterization that is universal among the literature. But I do believe that there is an agreed-upon intuition that an over-parameterized function class would have enough capacity to perfectly fit to "arbitrary data." Some papers also require a "large number" of solutions that perfectly fit any particular dataset. Often, people use the rule-of-thumb that $d \gg n$, which $d$ is the number of parameters, but this only makes sense when the function class is linear. One paper that discusses over-parameterization in great details is [1], and of course there are other approaches.
>
> From my reading of this paper, it is not immediately clear whether Definition 2 in this paper fits the intuitive notion that an over-parameterized should "interpolate" any dataset of certain size. However, the author already addressed this concern in Theorem 2, where their definition implies that the size of training data is almost half the VC-dim of the function class. One easy fix to this issue is to simply bring a sketch of this argument to Section 4.
>
> Alternatively, I am thinking one could also appeal to the Rademacher complexity. For binary classification problem, a function class $\\{ f(\theta; \cdot) : \theta \in \Theta\\}$ would be intuitively over-paremeterized over a set $\\{x_i\\}_{i=1}^n$ if for any iid Bernoulli r.v. $\epsilon_i$, there exists $\theta$ so that $\epsilon_i f(\theta; x_i) \ge 1$ for all $i$. I think it is equally valid, and perhaps to easier for the readers to understand, if the authors show that Definition 2 implies this statement.
>
> [1] Belkin, Mikhail. "Fit without fear: remarkable mathematical phenomena of deep learning through the prism of interpolation." Acta Numerica 30, 2021.

---

> > ### Author Response · Authors · 2023-11-13
> >
> > Thank you for your prompt response!
> >
> > We understand that our definitions might be non-standard, but they align well with your intuition. And since there is no scientific consensus, we believe our definition is a valid candidate. We are happy to engage in discussion and to reach a common understanding!
> >
> > > But I do believe that there is an agreed-upon intuition that an over-parameterized function class would have enough capacity to perfectly fit to "arbitrary data."
> >
> >
> > This fits our Definition 2. The arbitrary data can be any data coming from an agreed upon set of distributions $\mathbb D$. The algorithms we consider are ERMs with respect to these distributions, so they fit any data from these distributions. So, for example,  you can pick $\mathbb D$ to be all possible distributions or all possible distributions realizable by some neural network architecture. Both choices are aligned with your intuition above while being more general.
> >
> > >  Some papers also require a "large number" of solutions that perfectly fit any particular dataset.
> >
> > A large number of solutions immediately implies that any interpolating algorithm (or ERM) will have a large error over all other distributions that represent the functions the algorithm did not pick. So no algorithm can perform well on all such distributions. This again aligns well with our Definition 2.
> >
> >
> > Makes sense?

---

> > > ### Comment · Reviewer_Hb6n · 2023-11-13
> > >
> > > Thank you for the response.
> > >
> > > I am not saying that your definition is invalid. But I do believe that it is currently difficult to parse your definition of over-parameterization. Section 4 in its current form is *nowhere* enough to guide the reader through the transition between the overviews in Section 2 and the formal statements in Section 5.
> > >
> > > When I first read your paper, I was confused about Definition 2 until I saw Appendix E. So, I want to see some additional discussion about the implications of Definition 2, e.g. the example with VC dimensions in Theorem 2, as you introduce this definition. In other words, I think it is fine from the technical side, but the presentation is lacking.
> > >
> > > > This fits our Definition 2. The arbitrary data can be any data coming from an agreed upon set of distributions . The algorithms we consider are ERMs with respect to these distributions, so they fit any data from these distributions. So, for example, you can pick to be all possible distributions or all possible distributions realizable by some neural network architecture. Both choices are aligned with your intuition above while being more general.
> > >
> > > I am actually very confused by this. Where does your usage of PAC-learnability plays a role here? And over-parameterization should be the property of the function class, not the learning algorithm.

---

> ### Comment · Reviewer_bL1w · 2023-11-14
> **Overparametrization**
>
> I agree that there is no agreed-upon definition of overparametrization, except perhaps having a larger number of parameters than samples. It is legitimate that you want to introduce your own definition, and the definition you introduce is somewhat natural.
>
> However, this **doesn't change** anything to the fact that your argument that "all existing results must be vacuous since they are typically applied to the overparamterized setting" is **incorrect**. I am still not at all convinced you have invalidated nearly any of the existing literature. The "universally accepted fact" that the settings currently studied are "in the overparametrized regime" is based on a vague understanding of the concept (probably mostly about counting parameters).
>
> As I already tried to explain in my review, a lot of existing research provides bounds in a way that constructs custom function classes which indirectly take properties of the sampling distribution into account.
>  For instance, if a bound depends on the classification margin, then to show that it is "in the overparametrized setting'", you would need to show that the set of all functions which can achieve such a high margin is "large enough'' to qualify as "overparametrized"'. This is especially hard because the concept of the margin and the corresponding function class have been defined to further the opposite cause.
>
>
> I will answer the other points in more detail in the thread of my review.

---

> ### Comment · Reviewer_iLzH · 2023-11-14
> **About Definition 2**
>
> Like my peers, I have no clear definition of what the 'overparametrised setting' should be. However, please note that in my review, I never went against this notion as I mainly asked for more explanation.
>
> Like Rev. Hb6n, I would like expand on Definition 2.
>
> First of all you said in you previous answer that it is possible to take as $\mathbb{D}$ the class of all possible distribution realisable by a neural network architecture. However your definition holds only for finite families and is justified by the fact you draw uniformly over all distribution over $\mathbb{D}$. How do you generalise your definition to infinite (possibly uncountable) classes of distribution?
>
> Furthermore, I did not grasp the link between the intuitive notion of overparametrisation (number of weights far greater than the dataset size) and Def. 2.
>
> Indeed, let us take a neural network with a single neuron aiming to solve a classification task. Assume that it is possible, as you claim, to take $\mathbb{D}$ to be the set of all possible distributions related to this classification task. Then, I would say intuitively (please correct me if I am wrong) that for $\alpha$ small enough and $\beta$ large enough, we cannot hope, even for a large $n$, for a single neuron to generalise well on any distribution in $\mathbb{D}$. Therefore this single neuron architecture would be $(\alpha,\beta,n)$ overparametrised with respect to $\mathbb{D}$, which goes against the intuitive notion mentioned above.
>
> Did I miss something?

---

> > ### Author Response · Authors · 2023-11-15
> >
> > > How do you generalise your definition to infinite (possibly uncountable) classes of distribution?
> >
> > Please note that our negative results only become stronger by showing counterexamples over finite classes. Hence there is no need to generalize our techniques to infinite classes.
> >
> > > [...] and $\beta$ large enough [...]
> >
> > We assume that you meant $\beta$ small enough.
> >
> > > [...] which goes against the intuitive notion mentioned above.
> >
> > Our definition of learnability (and hence overparametrization) assumes that $\mathbb D$ only contains $\mathcal H$-realizable distributions. Hence in your example, $\mathbb D$ can only contain distributions with labelings from  some subset of (all) linear classifiers, and a single neuron can learn over all such $\mathbb D$ with reasonable accuracy and probability, for $n$ reasonably large. Hence one is not situated in the overparametrized regime.

---

### Author Response · Authors · 2023-11-16
**General Answer 1**

## ***General Answer 1: Our Definition of Overparametrization***

A number of reviewers have requested further clarifications on the definition of an "overparameterized setting" proposed in the paper. We address this issue here in detail.

One contribution of this paper is that we offer a formal definition of overparameterization that is at once very close to well-known intuitive notions (as we discuss in a moment) and, at the same time, enables proving insightful mathematical theorems. Finding the correct formalization of a notion is an important step toward fruitful mathematical work.

### Why Use Our Definition?

Consider the following four possible definitions of overparameterization, some of which have been mentioned by the reviewers. We explain why we believe that the definition proposed in this paper is the best definition when considering limitations on generalization bounds.

* **Definition I.** **The number of parameters is larger than the number of samples**. Namely, a hypothesis class $\mathcal{H} = \\{h_w: ~ w \in \mathbb{R}^k\\}$ is overparametrized if $k \gg n$, where $n$ is the size of training set. This is often used as a rule of thumb, e.g., for neural networks.
* **Definition II. The hypothesis class can realize any labeling for any $x_1,\dots,x_m \in \mathcal{X}$ and some large $m$.**
* **Definition III. The size of the training set is smaller than the VC dimension.** E.g., $n \ll d$, where $d$ is the VC dimension of the hypothesis class, and $n$ is the size of training set.
* **Definition IV. The definition we propose** (Definition 2 in the paper).

The first reason to prefer the definition proposed in the paper is that our definition is (typically) implied by the  other definitions. This means that when we prove that a bound cannot be tight in the overparameterized setting according to our definition, that in particular implies that it cannot be tight in any setting that is overparameterized according to the other definitions. **This makes our impossibility results stronger.**

The implications hold as follows:
 * **Definition I** $\implies$ **Definition III**. This implication holds for neural networks. It is well-known that large neural networks have large VC dimension. For instance, Theorem 1 in [Maa94] states that under mild assumptions, any neural network with at least 3 layers has VC dimension linear in the number of edges in the network. [Maa94] showed this for networks with threshold activation functions and binary inputs, and [BMM98] note that "It is easy to show that these results imply similar lower bounds" for ReLU activation. A similar result holds also for networks with 2 layers [Sak93].
 * **Definition II $\implies$ Definition III**. If a class $\mathcal{H}$ can realize any labeling for some  $x_1,\dots,x_m$ then the VC of $\mathcal{H}$ is at least $m$.  Namely, if Definition II holds for some $m \gg n$, then Definition III holds as well.
 * **Definition III $\implies$ Definition IV**. This implication is Item 2 in the proof on page 18 of the paper.

A second reason to prefer our definition is that it generalizes Definition III (and hence the other definitions) in a desirable way. Definition III pertains to a case where the hypothesis class can express _every possible_ labeling of the training set. However, this seems a bit rigid. What if the network or hypothesis class can express every labeling except one? Or it can express most but not all labelings? Intuitively, the network is still very much overparameterized, even if there are some specific choices of labelings that it cannot express. The essence of overparameterization is that the network can express too many of the possible labelings – not necessarily every single one of them. What is “too many”? Our answer, as captured in Definition IV, is that a network is overparameterized (can express “too many” labelings) if it can typically _express many different labelings that are consistent with the training set but disagree on the test set_, leading to poor predictions on the test set.

To conclude, the proposed definition is "natural" (according to jDS8) or at least "somewhat natural" (according to bL1w); impossibility results for the proposed definition are stronger than impossibility results for the other definitions; and the proposed definition generalizes the other definitions in a desirable way.


**Remark.** Additionally, we note that Definition I also has the weakness that it is sometimes possible to parameterize the same hypothesis class with varying numbers of parameters. For instance, a 1-layer linear network and a multi-layer linear network represent the same hypothesis class, but have different numbers of parameters. And, as reviewer Hb6n points out, this definition mostly "makes sense when the function class is linear".

---

> ### Author Response · Authors · 2023-11-16
> **General Answer 1 (Ctd.)**
>
> ### Why Theorems 1 and 2 Apply to Generalization Bounds For Deep Learning
>
> Some reviewers have questioned our view that "most real world problems are in the overparametrized setting" (as Reviewer bL1w15 puts it), or that the limitations we prove in Theorems 1 and 2 on the tightness of generalization bounds apply to actual generalization bounds in the literature. The applicability of Theorems 1 and 2 to deep learning is actually a fairly straightforward and objective mathematical fact, as we explain in this section.
>
> Fix a reasonable sample size $n$, a deep neural network $N$, and a training algorithm for that network (e.g., SGD). Take any generalization bound B. There are two possibilities:
>  1. The bound B explicitly states that it only applies to a specific subset of all possible population distributions or to a specific learning algorithm, and employs these restrictions in its proof. (In this case, we are happy, because these are the types of bounds we are advocating for. [Note that Theorem 3 might still imply some limitations for the bound.])
>  2. The bound B does not explicitly state which population distributions it applies to, and does not rely on the specifics of the training algorithm.
>
> The limitations we prove in Theorems 1 and 2 apply to any bound B that falls in the second category. The literature contains many such bounds, we can provide a long list. The following explains why Theorems 1 and 2 apply to any bound B of the second type.
>
> Because the bound B does not explicitly state any limitations on the set of population distributions that it applies to, _in particular_ this means that B must hold (i.e., be a valid upper bound on the population loss) for all the distributions in $\mathbb{D}$, the collection of all distributions that are realizable by the network N. Because the bound B does not make any specific assumptions on the training algorithm, in particular it must hold (i.e., be a valid upper bound on the population loss) for any of the algorithms $A_h$ and the ERM algorithms mentioned in Theorem 1.
>
> Now, as discussed in the previous sections, it is well known (from [Maa94], [BMM98], [Sak93], etc.) that in a large neural network like N, the VC dimension is roughly linear in the number of edges in the network. In deep learning, the number of edges in the network is much larger than the sample size $n$, and therefore, Definition III of overparameterization (mentioned above) holds. The implication **Definition III $\implies$ Definition IV** above entails that we are in the overparamterized setting as in Definition 2 in the paper (in other words, using a sample of size $n$ it is not possible to learn all the distributions in $\mathbb{D}$). Hence, the assumptions of Theorem 1 are met, and therefore Theorem 1 implies that the bound B is not uniformly tight.
>
> To recap, we have used only two assumptions:
>  * The bound B falls in category 2 above.
>  * This is a deep learning setting, so in particular the number of edges in the network N is much larger than the sample size $n$.
>
> These two assumptions are straightforward and easy to verify. Hence, Theorem 1 applies to many bounds in the deep learning literature. (A similar reasoning applies also to Theorem 2.)
>
> $~$
>
> $~$
>
> _______
>
> * [Maa94] W. Maass. Neural nets with superlinear VC-dimension. Neural Computation, 6(5):877- 884, 1994.
> * [BMM98] Peter Bartlett, Vitaly Maiorov, and Ron Meir. Almost linear VC-dimension bounds for piecewise polynomial networks. Neural Computation, 10(8):2159–2173, Nov 1998.
> * [Sak93] A. Sakurai. Tighter bounds on the VC-dimension of three-layer networks. In World Congress on Neural Networks, volume 3, pages 540- 543, Hillsdale, NJ, 1993. Erlbaum.

---

> > ### Comment · Reviewer_bL1w · 2023-11-19
> > **Distribution dependency**
> >
> > Dear authors,
> >
> > Thanks for your comment. Whilst it does clarify a few things, I believe **severe issues remain in terms of how a reader of your paper might interpret the informal descriptions of your results**.
> >
> > As I explained before, typical generalization bounds indirectly take the sampling distribution into account in the following way: the generalization bound involves a parameter (e.g. the margin, the margins, the intrinsic dimension of the data in some feature space etc.) which will make the bound much tighter if the (empirical!) sampling distribution happens to exhibit "nice" properties. Consider the bound in [7], which essentially scales like $$\widetilde{O}(R\sqrt{\frac{1}{\gamma^2 n}}+\sqrt{\frac{\log(1/\delta)}{n}})$$ where $\gamma$ is the margin and $R$ is some norm-like function of the weights. By a union bound, the result can be made **post-hoc**, which means that (as I explained below), the values of $R,\gamma$ can be optimized **after training**.
> >
> > In particular, the answers to your points above are:
> >
> > Yes, taken at face value, this sort of result technically falls under the category of "The bound B does not explicitly state which population distributions it applies to, and does not rely on the specifics of the training algorithm", **strictly speaking**.  Yes, this means that theoretically, **if we ignore the "overparametrization" issue** (which is difficult, partly because the function class is not well defined when considering the result in its post hoc form), *Theorem 1 technically would apply*.  *However, it doesn't tell us as much as your informal statements in the paper make out*: it just says that for many distributions, the bound is vacuous. However, those **distributions correspond to "bad" configurations** of the parameters of the bound which are sensitive to the empirical distribution.
> >
> > The parameters of the bounds (such as the margins or intrinsic dimension of the data) are chosen specifically to ensure that most "natural" datasets will make the bound small in practice. For instance, the results in [4] improve if the labels align with a principal direction of the data in NTK feature space.
> >
> > This type of well-established practice which you dismiss in your paper is **not actually very different from explicitly assuming something about the distribution**, which is what you propose.
> >
> > **Consider again the idealized setting of a linear classifier** ($w\in \mathbb{R}^d$) in a very high dimensional space, and consider the following algorithm:
> > "Minimize the quantity $|w|_0$ (the number of non zero entries of $w$) amongst all the values of $w$ which achieve optimal empirical error."
> >
> >
> >
> > It's reasonable to show a bound for this type of linear classifier which scales like $\widetilde{O}(\sqrt{\frac{(1+|w|_0)}{n}}+\sqrt{\frac{\log(1/\delta)}{n}})$ where $\widetilde{O}$ notation hides logarithmic factors, including factors of the form $\log(d)$, and where the bound holds for any value of $|w|_0$ (which is minimized by the algorithm). If we apply your theorem, we learn that "for many distributions (i.e., the ones where it is not possible for classify the data correctly with a sparse vector), the bound is vacuous". However,
> >
> > (1) that doesn't make the result "lacking": it doesn't even cast serious doubt on the validity of the approach: in the example above, the quantity $|w|_0$ is introduced because the creator of the bound believes that the input distribution has the a favorable property: only a small number of components are relevant to the label. Once this property is observed in the empirical distribution, the hypothesis is confirmed and the bound is not vacuous
> >
> > (2) The distinction between this approach and simply **assuming** that the population version of the sampling distribution has the property that the label is determined by a small number of components is not very important. Furthermore, the **current approach** (based on the empirical distribution) is **arguably superior**.
> >
> >
> > Again, note that this simple example of a linear map is an illustration of my points, but those points apply to most recent works on neural networks as well: most of recent research can be understood as modifying the problem so that we can, as it were.
> >  "identify the high dimensional representation of the input and the notion of "sparsity" which makes the above argument hold". Probably one of the most informative examples would be the NTK literature [4] (yes, this concerns regression, but this is only a technicality, the results most likely hold for classification as well), where the equivalent of the above condition is that the labels strongly align with the projections of the inputs onto one given direction in NTK feature space. In [3], another ingenious subtle condition is introduced in terms of each layer's margin. Even in more rudimentary results such as those of [7], there is no categorical evidence in your paper or anywhere that the margin parameter cannot play a similar role.

---

> ### Comment · Reviewer_Hb6n · 2023-11-17
> **It's time to pause the philosophical debates and write a revision.**
>
> First, I want to point out that the correct direction should **Definition IV $\Rightarrow$ Definition III** because your definition should cover the existing intuitive notion of over-parameterization.
>
> Back to the important part, now it is clear to me that our (mostly) philosophical debate is hitting diminishing return. I fact, I do not think most of the debate were necessary because many reviewers (e.g. iLzH and me) are not against your line of reasoning in the first place. Personally, I am mainly asking for additional clarification and examples to support your results. In other words, I am more worried that this paper as currently written would not convince other readers, especially those who only take a cursory look at the statements.
>
> I recommend that you incorporate the key points in our discussions into a revision. While you made some good points, they are mostly made without complete contexts and scattered over many threads (whose total length probably exceeds a full paper). Now is time turn our debates into something organized and concrete.
>
> Here are the two key points I want to see in the revision:
>
> 1. A thorough discussion of why your definition of over-parameterization is consistent with the intuitive notions current used in the literature. This needs to include specific example such as the content you had written above. And you need to explain why this new definition is more useful than the existing notions.
>
> 2. As reviewer bL1w noted, the statements of Theorem 1 and 3 are not particularly clear on their exact implications. I personally think the theorems are fine, but evidently they are not clear to some. Therefore, I would like to see a few examples on how your results would apply to some existing generalization bounds in the literature.
>
> 3. I agree with reviewer bL1w that your response are not consistent regarding the formality and preciseness of the concepts such as "over-parameterization" and "distribution-independence". Therefore, in the revision, you must be careful with how exact your definition and explanations are.
>
> I will defer any updates to my assessment of this paper until I read the revised version. And I will make clear that I will *lower* the scores if the revision is not addressing the concerns as I laid-out above.

---

> > ### Author Response · Authors · 2023-11-17
> > ****General Answer 2: Revision & our notion of distribution independence****
> >
> > First, a remark: we agree with reviewer Hb6n in so far that a new version of the manuscript is needed, so that you, the reviewers, can judge more accurately how this work can be digested by a reader that does not have access to the discussions in this forum. We are currently working on a revised version and hope to upload it soon.
> >
> > ================================================================================
> >
> > Reviewer bL1w, and perhaps also Reviewer Hb6n, take issue with our notion of  ‘distribution-independent’ and the type of bounds we deem to fall into this category. We addressed this in our recent response to Reviewer bL1w.
> > But perhaps the core issue is with the terminology? Maybe you will find the term 'distribution-free' better?
> >
> > From https://en.wikipedia.org/wiki/Nonparametric_statistics : "Methods which are distribution-free, which do not rely on assumptions that the data are drawn from a given parametric family of probability distributions."
> >
> > From https://en.wiktionary.org/wiki/distribution-free:
> >
> > "Adjective
> >
> > distribution-free
> >
> > (statistics) Free of specific assumptions about the shape of the probability distribution generating the sample."
> >
> > This terminology fits our framework! we consider only bounds that use the training set and the output hypothesis (or the description of the algorithm). We suggest using the term "distribution-free generalization bound" instead of "distribution-independent generalization bound". Do you think this terminology is better suited?
> >
> > If so, we will replace the term ‘distribution-independent’ with ‘distribution-free’ throughout the manuscript.
> >
> > Please let us know what you think.

---

> ### Comment · Reviewer_iLzH · 2023-11-17
> **About 'Defintion IV should imply Definition III'**
>
> Dear Rev. Hb6n,
>
> Concerning your remark 'I want to point out that the correct direction should Definition IV $\rightarrow$ Definition III because your definition should cover the existing intuitive notion of over-parameterization', I believe that the authors are right on this one.
>
> Indeed, their contributions are negative results, as they prove it is impossible for two assertions to hold simultaneously (with high probability). Thus, a deep net satisfying Definition III is overparametrised in the sense of Definition IV and thus, is also concerned by the negative results of this work.
>
> That being said, it is indeed worth notifying that, as Definition IV does not imply Definition I, the former may not be enough to prove sharp positive results on deep nets satisfying the latter.
>
> Hope this helps!

---

> > ### Comment · Reviewer_Hb6n · 2023-11-17
> >
> > Hi Reviewer iLzH,
> >
> > Thank you for your correction. I was saying that Definition IV is more general than Definition III. And reading the author's comments again, you are right that I have been thinking in the opposite direction.
> >
> > You are right that such arguments still would not lead to Definition I. I am going to wait to see how the authors organize their revision before I pass further judgement. And to your question in your thread, it is indeed possible for the authors to submit a revised paper.

---

> ### Comment · Reviewer_bL1w · 2023-11-19
> **Overparametrization. Many apologies for repeating myself a little but I think there are still misunderstandings. Please read and **upload a revision**.**
>
> I think **there are still severe misunderstandings regarding** the actual nature of most generalization bounds in the **learning theory literature**.
>
> You say: "definition I (more parameters than samples) implies Definition II (Small VC dimension)"
>
> As I have tried to explain in my previous comments, this is **only true** if the function class considered is the **class of all neural networks with a particular architecture**. This **excludes almost all "recent" (after 2015) works, including the ones you mention in your paper**. In other words, your comment only applies in the case of very old results such as those in the three references you cite in your comment [Maa94, BMM98, Sak93]. In most recent results, for instance those in [7,11], there is a restriction on the norms of the weights matrices, which means your definition of overparametrization is now different and much harder to check, and it's questionable whether what people mean when they say that "neural networks are typically overparametrized" implies this.
>
> For the sake of illustration, consider an extremely simplified scenario: the input $x\in\mathbb{R}^d$ for a very large dimensionality $d$. This is a 1 layer equivalent of the bounds in [11]. A basic result says that the generalization of a linear classifier $w\in\mathbb{R}^d$ with $\|w\|\leq R$ for some constant $R$ scales like $$O\left(\sqrt{\frac{R^2}{n}}+\sqrt{\frac{\log(1/\delta)}{n}}\right).$$
> There is **no dependence on the dimension/number of parameters here**. However, for your definition of "overparametrization" to hold, it must be the case that $(\mathcal{H}_R,\mathbb{D})$ is not learnable, where $\mathcal{H}_R$ is the function class defined by the set of weights $w\in\mathbb{R}^d$. This condition is much harder to check.
>
> This is not just a remote example: nearly all works in modern learning theory of neural networks involve and utilize subtle constraints on the norms of the weight matrices, which makes the situation similar. That doesn't mean that your work is not worthy of interest: it is still interesting to say that whether or not norm based bounds are tied is a problem which is fundamentally related to whether the abstract function class defined is small enough to make the problem no longer overparametrized. However, it does make your **paper's current version misleading in its statements**: even for simple examples such as the bounds in [7], you should not be saying things like "alex's analysis is lacking" or "likely to be vacuous for SGD or any specific algorithm of interest" (this is how you describe [7]), because there is no strong evidence that this is true.
>
>
>
>
> References
>
> [1] IAN F. BLAKE AND CHRIS STUDHOLME, PROPERTIES OF RANDOM MATRICES AND APPLICATIONS, 2006
>
> [2] Omer Angel and Yinon Spinka. Pairwise optimal coupling of multiple random variables, 2021. (Arxiv)
>
> [3] C Wei, T Ma, Improved Sample Complexities for Deep Networks and Robust Classification via an All-Layer Margin, ICML 2019.
>
> [4] Sanjeev Arora, Simon S. Du, Wei Hu, Zhiyuan Li, Ruosong Wang, Fine-Grained Analysis of Optimization and Generalization for Overparameterized Two-Layer Neural Networks, ICML 2019.
>
> [5] Vaishnavh Nagarajan, J. Zico Kolter, Uniform convergence may be unable to explain generalization in deep learning, NeurIPS 2019
>
> [6] Philip M. Long, Hanie Sedghi, Generalization bounds for deep convolutional neural networks, ICLR 2020
>
> [7] Peter Bartlett, Dylan J. Foster, Matus Telgarsky. Spectrally-normalized margin bounds for neural networks, NeurIPS 2017.
>
> [8] Wei and Ma, Data-dependent Sample Complexity of Deep Neural Networks via Lipschitz Augmentation, NeurIPS 2019
>
> [9] Vaishnavh Nagarajan and Zico Kolter. Deterministic PAC-bayesian generalization bounds for deep networks via generalizing noise-resilience, ICML 2019.
>
> [10] Antoine Ledent, Waleed Mustafa, Yunwen Lei, Marius Kloft, Norm-based generalisation bounds for multi-class convolutional neural networks, AAAI 2021.
>
> [11] Florian Graf, Sebastian Zeng, Bastian Rieck, Marc Niethammer, Roland Kwitt, On Measuring Excess Capacity in Neural Networks, NeurIPS 2023.
>
> [12] Size-Independent Sample Complexity of Neural Networks. Noah Golowich, Alexander Rakhlin, Ohad Shamir, COLT 2018.
>
> [13] Peter L. Bartlett, Philip M. Long, Gábor Lugosi, Alexander Tsigler, Benign Overfitting in Linear Regression
>
> [14] Shamir, The Implicit Bias of Benign Overfitting, COLT 2022.
>
> [15 ] Tomer Galanti, Mengjia Xu, Liane Galanti, Tomaso Poggio , Norm-based Generalization Bounds for Compositionally Sparse Neural Networks
>
>
> [Maa94] W. Maass. Neural nets with superlinear VC-dimension. Neural Computation, 6(5):877- 884, 1994.
>
> [BMM98] Peter Bartlett, Vitaly Maiorov, and Ron Meir. Almost linear VC-dimension bounds for piecewise polynomial networks. Neural Computation, 10(8):2159–2173, Nov 1998.
>
> [Sak93] A. Sakurai. Tighter bounds on the VC-dimension of three-layer networks. In World Congress on Neural Networks, volume 3, pages 540- 543, Hillsdale, NJ, 1993. Erlbaum.

---

> ### Author Response · Authors · 2023-11-19
>
> Dear Reviewer bL1w,
>
> Thank you for your detailed comment. Just to confirm: did you see **[the response](https://openreview.net/forum?id=NkmJotfL42&noteId=Yi5NO8zIGa)** titled _**Example: A Margin Bound That Is Not Uniformly Tight**_ and _**General Case: Why Margin Bounds Are Not Uniformly Tight**_ that we wrote in your review thread? (We simply want to be completely sure that we are all on the same page before responding further.)

---

> > ### Comment · Reviewer_bL1w · 2023-11-19
> > **Yes**
> >
> > Dear Authors,
> >
> > Thanks for being so responsive!  Yes, I read your response to this comment. I agree with your statement that the bound will not be tight for many distributions. Those distributions are not the ones for which the bound is intended to be tight, and the user can tell that the bound is not tight by observing the margin is not high. That doesn't make the analysis "lacking".
> >
> > Imagine we follow your advice and instead create a "distribution-dependent" bound that (for instance) explicitly assumes that the data is strongly concentrated in two diametrically opposite parts of the sphere. How is that different (or better) than the margin approach? There is more flexibility in the empirical approach.

---

> > > ### Comment · Reviewer_bL1w · 2023-11-19
> > >
> > > I am not asking to reject your paper. I am not saying your results are not interesting. I just think the informal text that discusses their implications can be misleading: in its current there are plenty of casual readers who would read it and come out with the belief that all recent work in learning theory of DNNs is outdated, and that some work with an **explicit** assumption on the distribution would be the only acceptable "correction". After reading your paper and your comments, I still don't think that's close to being true.

---

> ### Author Response · Authors · 2023-11-20
>
> Dear Reviewer bL1w,
>
> $~$
>
> Thanks again for your dedicated and thoughtful engagement with our paper.
>
> You wrote "I think we are getting closer to understanding each other. It seems that we agree on the strict mathematical facts, but not on the implications in terms of how you informally explain your results." We generally agree with this characterization. At least the majority of the differences between our views focus on matters of interpretation.
>
> We are open to revising the interpretation of the results offered in the paper, with the hope of reaching a formulation that everyone feels comfortable with. It is also possible that, in the interest of scientific prudence and precision, the paper should focus on presenting the mathematical facts, and should simply include less interpretational language. This would allow the reader (and the community) to reach their own conclusions, without requiring ICLR to endorse any specific views.
>
> $~$
>
> Before we embark on making substantial changes, we wanted to ask two questions:
>
> $~$
>
>  1. Can you identify which locations in the paper are in your view most in need of revision? We understand that you take issue with Appendices C and D. Are there any additional specific portions of the paper that you would suggest that we focus our attention on?
>
> $~$
>
>  2. We have carefully read the concerns that you and the other reviewers have raised regarding the narrative surrounding our results. We know of various aspects of our interpretation that you find misleading or incorrect. But we are less certain as to what your preferred interpretation of our results would be. In your view, what is the best, most balanced interpretation of these results that you would like to see us expand upon in our revision? Please do not feel obligated to answer this question (as it is of course our duty as authors to explain our work). However, any pointers that summarize what you view as the best interpretation of the results would be very welcome and appreciated.

---

> ### Author Response · Authors · 2023-11-21
>
> ## ***General Answer 3: New Revision of The Paper***
>
> We have uploaded a new revision of the paper that incorporates a lot of the feedback that the reviewers have provided. To accommodate an easy review of the changes, the main changes are highlighted in blue in the new PDF.
>
> The changes include:
>  * **Changed:** Significant change of language concerning the consequences and interpretation of our formal results. The new language is aiming to be more scientifically prudent, and to clearly separate matters of objective mathematical fact from matters that are open for debate, on which the reader might want to form an independent opinion.
>  * **Removed:** The previous revision contained two appendices that some reviewers felt were misleading. These were former appendix C ("On Optimistic Application of Distribution- and Algorithm-independent Bounds"), and former appendix D (the discussion about "Alex"). We have removed these appendices to avoid any possibility of misleading the reader.
>  * **Added:** New Appendix C.1, containing a discussion of Margin Bounds.
>  * **Added:** New Appendix D, containing a discussion of our definition of overparameterization, including a comparison to alternative definitions.
>
> Additionally, we have reorganized some parts of the text, and have improved some of the proofs.
>
> We are looking forward to hearing your feedback on the this revision.

---

> > ### Author Response · Authors · 2023-11-22
> > **Friendly Reminder**
> >
> > Dear Reviewers,
> >
> > Just a friendly reminder:
> >
> > Following your guidance, we have uploaded a new version of the paper. We feel this version addresses the main concerns you have voiced, and we hope you agree it's on the right track.
> >
> > The deadline for the discussion period is near. It would be nice if you could take a look at the new version and tell us what you think. In particular, if you have any remaining major concerns, please let us know so that we can attempt further modifications as needed.

---

### Meta-Review · Area_Chair_Tv43 · 2023-12-05

**Metareview:**

This paper addresses the question of why some algorithms generalize well (i.e., achieve small risk) in the _overparameterized_ setting, while certain generalization bounds cannot provide tight bounds for them (as demonstrated empirically by Jiang et al. (2020)). Starting from a custom definition of overparameterization (which is in some cases loosely equivalent to $n << d$), the paper proves two central results:
1. Bounds that _do not_ consider properties of the learning algorithm or data distribution cannot be tight in the overparameterized setting.
2. Bounds that _do_ consider properties of the learning algorithm cannot be tight for all data distributions in the overparameterized setting. If the bound is agnostic to the data distribution, then there are distributions for which it will fail to explain a well-generalizing algorithm. (Perhaps this summary is too simplistic; see `bL1w`'s review for an excellent detailed summary.)

Let me cut to the chase: everyone (reviewers and myself) agrees that this is a good paper, and it deserves to be accepted. The contributions are (more or less) novel and very compelling; they could potentially shift the direction of research in generalization theory. Moreover, the results are sound, and the writing is excellent.

The main weaknesses highlighted by the reviewers are:
1. The definition of overparameterization used in the paper could use more elaboration and motivation.
2. The paper's tone is at times too sensational or provocative. For instance, claiming that a bound "fails," when in fact what is meant is that it is vacuous but, nonetheless, a valid inequality. Perhaps this is just a matter of oversimplification in the interest of brevity.
3. The discussion of related work is a bit problematic. In trying to fit prior work into their narrative, the authors may have misinterpreted some (e.g., Dziugate & Roy, 2017), or made sweeping generalizations/oversimplifications that do not given proper credit (e.g., dismissing all "Rademacher bounds" as distribution-independent, when margin bounds are implicitly distribution-dependent). And, as pointed out by `bL1w`, the NTK literature is not even discussed. When prior work is properly recognized, the paper's conclusions may not seem all that revelatory.

To their credit, the authors did upload a revision that partially addresses these critiques. Nonetheless, even after the revision, `bL1w` felt that some of the paper's bold statements could be misleading.

In my humble opinion, I agree with `bL1w` that uniform tightness (over data distributions) is not _all that_ important, given that we have implicitly distribution-dependent bounds that are empirically verifiable. But that is just my subjective take; and I agree with the authors that this topic can, and should, be debated. To that extent, I support giving this paper at least a spotlight talk, to call attention to it and spark the debate.

**Justification For Why Not Higher Score:**

While the authors have indeed toned down their claims, especially w.r.t. prior work, I share `bL1w`'s concern that some statements are still misleading or sensational.

**Justification For Why Not Lower Score:**

As already stated, this work is well executed and provocative, and its conclusions deserve to be highlighted at the conference, to spark debate. I think it deserves at least a spotlight.

---

### Decision · Program_Chairs · 2024-01-16

Accept (poster)